# Contribution of amygdala to dynamic model arbitration under uncertainty

Jae Hyung Woo[1], Vincent D. Costa [2], Craig A. Taswell [3], Kathryn M. Rothenhoefer[2], Bruno B. Averbeck [3] & Alireza Soltani [1] ✉

Intrinsic uncertainty in the reward environment requires the brain to run multiple models simultaneously to predict outcomes from preceding cues or actions. For example, reward outcomes may be linked to specific stimuli and actions, corresponding to stimulus- and action-based learning. But how does the brain arbitrate between such models? Here, we combined multiple computational approaches to quantify concurrent learning in male monkeys performing tasks with different levels of uncertainty about the model of the environment. By comparing behavior in control monkeys and monkeys with bilateral lesions to the amygdala or ventral striatum, we found evidence for a dynamic, competitive interaction between stimulus-based and action-based learning, and for a distinct role of the amygdala in model arbitration. We demonstrated that the amygdala adjusts the initial balance between the two learning systems and is essential for updating arbitration according to the correct model, which in turn alters the interaction between arbitration and learning that governs the time course of learning and choice behavior. In contrast, VS lesions lead to an overall reduction in stimulus-value signals. This role of the amygdala reconciles existing contradictory observations and provides testable predictions for future studies into circuit-level mechanisms of flexible learning and choice under uncertainty.

One of the challenging aspects of learning in naturalistic settings is that it is inherently unclear what features or attributes of choices are predictive of subsequent reward outcomes. Imagine successfully operating a new coffee machine after switching it on and pressing a flashing button on the left side of its screen. What should you press the next time you want to get coffee: the same button or any button that is flashing? In this specific example, reward outcomes can be equally attributed to either the identity of a choice option (e.g., flashing button) or the action needed to obtain that option (pressing the left button), corresponding to uncertainty about the correct model of the environment (stimulus-based vs. action-based). More generally and during most naturalistic settings, reward outcomes could be linked to any combination of features or attributes of a selected option or

chosen action. It has been suggested that the brain tackles such uncertainty by running multiple predictive models of the environment, with each model predicting outcomes based on different attributes of choice options, and using the reliability of these predictions to select the appropriate model to inform choice behavior[1–3].

Although many conceptual and algorithmic solutions to model arbitration exist[2–4], confirming implementation level details in terms of the operation of neural circuits has remained a challenge due to several factors. First, most experimental paradigms involve manipulating uncertainty in one of two ways. In some paradigms, there is uncertainty about which of multiple choice options is more rewarding through probabilistic reward contingencies and contingency reversals[5–7]. In other paradigms, the correct option is deterministically linked to

[1]Department of Psychological and Brain Sciences, Dartmouth College, Hanover, NH, USA. [2]Division of Developmental and Cognitive Neuroscience, Emory National Primate Research Center, Atlanta, GA, USA. [3]Laboratory of Neuropsychology, National Institute of Mental Health, National Institutes of Health, Bethesda, MD, USA. ✉e-mail: alireza.soltani@dartmouth.edu

reward outcomes, but there is uncertainty about the correct model of the environment and when to choose that option[8–12]. Few, if any, studies have manipulated expected and unexpected uncertainty[13] related to stimulus-action-outcome relationships together with uncertainty about which model of the reward environment is relevant at the time. Critically, in reward environments where the correct model of the environment does not change frequently, the reliability of different models can reach their asymptotes very quickly, concealing the contributions of circuits involved in dynamic arbitration and model selection. Second, it is intrinsically difficult to measure and track the contributions of multiple learning systems and their interactions because different learning systems can drive choice behavior at any given moment. Third, because computations required for learning and arbitration under uncertainty must interact with each other, many cortical and subcortical areas may appear to be similarly involved in different processes (lack of specialization). For example, the amygdala has been shown to contribute to reward learning under uncertainty by both improving and impairing learning performance[14–20] and by its contribution being associated with different types of uncertainty[13,21].

To overcome these challenges and reveal the circuit and neural mechanisms underlying arbitration, we applied multiple computational approaches to examine choice behavior in three groups of monkeys (a control group and two groups with bilateral lesions of either the amygdala or ventral striatum) performing a probabilistic learning task that involved multiple forms of uncertainty. These included uncertainty about the better option on a given trial (expected uncertainty), uncertainty about the correct model of the environment, and uncertainty about when reward associations change (unexpected uncertainty), thus creating a challenging task that could reveal the role of the amygdala and ventral striatum (VS) in all three of these processes. To track the simultaneous contributions of multiple learning systems and their interaction, we extended metrics based on information theory[22,23] to quantify consistency in choice and learning based on stimulus- and action-based learning over time. Additionally, we developed several reinforcement learning (RL) models that, along with previous models, were used to fit choice behavior on a trial-by-trial basis. These models extended the previous ones by incorporating static or dynamic arbitration among alternative learning systems, based on different signals. We then examined the best models and their estimated parameters, particularly those related to the arbitration process, to pinpoint the roles of the amygdala and VS in reward learning and arbitration. Moreover, by modulating the key parameters of the model, we simulated and qualitatively replicated the distinct behavioral signatures of amygdala or VS lesions. Together, by utilizing the above methods, we provide evidence for interactions between stimulus-based and action-based learning under uncertainty, uncover mechanisms underlying arbitration between the two systems, explore how arbitration and learning processes interact, and determine the amygdala's contributions to arbitration and overall behavior.

## Results

### Behavioral paradigm with multiple forms of uncertainty

We examined monkeys' choice behavior when performing a variant of a probabilistic learning paradigm that involves multiple forms of uncertainty. In this paradigm, during each block of 80 trials, monkeys selected between two novel visual stimuli that were randomly presented on the opposite sides of the screen (Fig. 1a; see *Experimental paradigm* in "Methods"). Selection of each option was rewarded with a certain probability (80:20, 70:30, or 60:40), but the probabilities for better and worse options reversed at a random point within the block without any signal to the monkey. Critically, at the start of a block of trials the monkeys were unaware if the assigned reward probabilities for a particular block were linked to the selection of specific stimuli or locations (Fig. 1b). This created uncertainty about the correct model of the environment (stimulus-based vs. action-based). In one task,

rewards were exclusively based on stimulus-outcome associations (What-only task; Fig. 1c). In the second task, rewards in each block of trials were determined by either stimulus-outcome or action-outcome associations (What/Where task; Fig. 1d). Together, these components resulted in three types of uncertainty: uncertainty about the correct option on a given trial (expected uncertainty), uncertainty about the correct model of the environment, and uncertainty regarding when reward associations reverse (unexpected uncertainty).

### Evidence for multiple learning systems and their interaction

To examine the presence of multiple learning systems and reveal the effects of different forms of uncertainty, we first compared the performance of control monkeys across different tasks and reward schedules. Overall performance was best during the What-only task, in which only the stimulus identity was predictive of reward, and there was no inherent uncertainty about the model of the environment in terms of objective task structure. Using mixed-effects analysis which accounts for subject variability, we observed that the probability of choosing the more rewarding option, $P(Better)$, during the What-only task was significantly higher than in either What blocks (main effects of block type; $\beta_{\text{What}} = -0.074$, $p = 2.74 \times 10^{-28}$) or Where blocks ($\beta_{\text{Where}} = -0.087$, $p = 2.39 \times 10^{-37}$) of the What/Where task, which involved additional uncertainty about the correct model of the environment. We also found that expected uncertainty affects performance, measured by $P(Better)$: across all block types and tasks, performance improved as it became easier to discriminate between the reward probabilities of the two options (main effect of reward variance; $\beta_{\text{var(P)}} = -1.950$, $p = 4.60 \times 10^{-31}$).

To capture the effect of reward feedback and how it was used to perform the task and adjust the behavior, we utilized information-theoretic metrics to quantify consistency in reward-dependent choice strategy on two attribute dimensions, stimulus identity and stimulus location[22,23]. Specifically, we examined the conditional entropy of reward-dependent strategy (ERDS), defined as the Shannon entropy of stay/switch strategy conditioned on the previous reward feedback, separately for stay/switch based on action or stimulus identity (see Eq. 1 in "Methods"). Lower values of $ERDS_{\text{Stim}}$ suggest that the animals stayed or switched after reward feedback based on stimulus identity (stimulus-based learning), whereas lower values of $ERDS_{\text{Action}}$ indicate that the animals' stay/switch strategy was based on assigning reward to the chosen action (action-based learning).

To quantify the interaction between the two learning systems, we computed the correlation between $ERDS_{\text{Stim}}$ and $ERDS_{\text{Action}}$ computed from each 80-trial block (Supplementary Fig. 5). We found that even in the What-only task for which action-based learning was irrelevant and minimally used, there was a negative correlation between $ERDS_{\text{Stim}}$ and $ERDS_{\text{Action}}$ (Spearman's correlation, $r = -0.123$, $p = 4.29 \times 10^{-7}$), suggesting that more consistency in using one model resulted in less consistency using the other model. For the What/Where task, we observed stronger negative correlations between $ERDS_{\text{Stim}}$ and action $ERDS_{\text{Action}}$ for both block types (What: $r = -0.602$, $p = 2.43 \times 10^{-296}$; Where: $r = -0.578$, $p = 1.32 \times 10^{-261}$). Overall, these results reveal significant interactions between the stimulus- and action-based learning systems.

Considering previous findings on the influence of learning strategy on response time[24,25], we hypothesized that reaction time (RT) is influenced by the dominant learning system at any given time. To test this hypothesis, we categorized trials as either stimulus- or action-dominant by directly comparing $ERDS_{\text{Stim}}$ and $ERDS_{\text{Action}}$ (see *Data analysis and statistical tests* in "Methods" for details). Using this approach, we found that during the What/Where task, responses in stimulus-dominant trials were significantly slower than action-dominant trials in both block types (Supplementary Note 1). This contrast in stimulus vs. action-driven RTs was harder to identify in the What-only task, in which choices were dominated by the

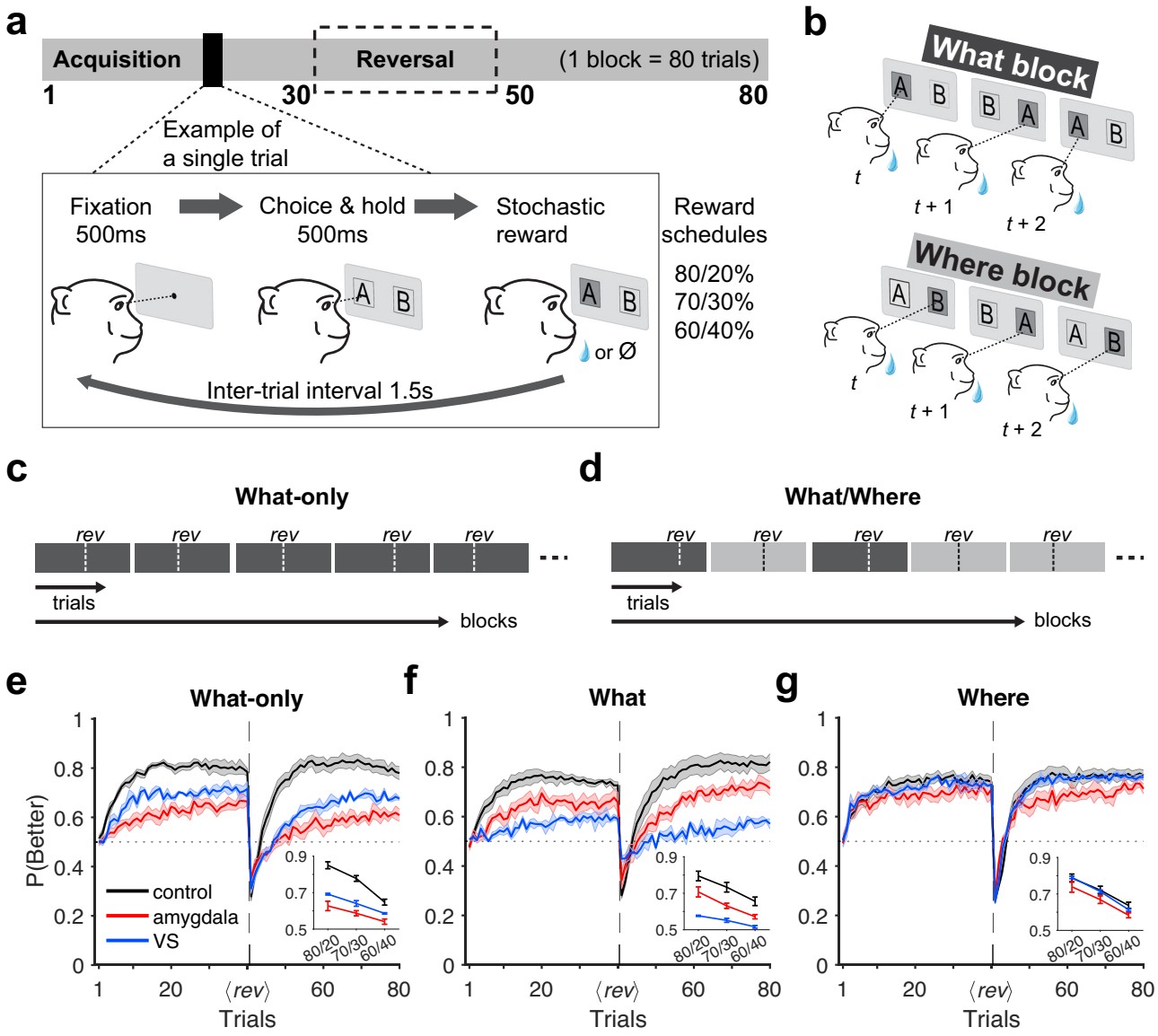

**Fig. 1 | Experimental paradigm, block types, and time course of performance. a** Timeline of each block and a single trial of the experiment. At the beginning of each block, two novel stimuli (abstract visual objects) were introduced. On each trial, animals indicated a choice by making a saccade toward one of the two options on the left and right sides of fixation. The selection of each stimulus was rewarded probabilistically based on three reward schedules. Reward contingencies were reversed between better and worse options on a randomly selected trial (between trials 30 and 50). **b** Different block types. In *What* blocks (top), reward probabilities were assigned based on stimulus identity, with a particular object having a higher reward probability. In *Where* blocks (bottom), reward probabilities were assigned based on the location of the stimuli, with a particular side having a higher reward probability regardless of the object appearing on that side. **c** Outline of the What-only task. Here, only What blocks were used for the entire experiment. Vertical dotted lines, *rev*, indicate a random reversal point within each block. **d** Outline of the What/Where task. In this task, What (black) and Where (gray) blocks were randomly interleaved. **e–g** Time course of performance of the monkeys in each group, measured as the probability of choosing the better option, *P*(*Better*), separately for different tasks and block types. Shaded regions indicate error bars. Insets show the averaged performance by reward schedules. Error bars = SEM across subjects (controls: n = 4 in What-only, n = 6 for What/Where; amygdala: n = 4 for both tasks; VS: n = 3 for both tasks). Source data are provided as a Source data file.

stimulus-based system. Nonetheless, rare action-dominant trials happened when reward value estimates based on the two systems were close to each other, resulting in slower and more erroneous responses. Overall, these results show that entropy-based metrics could be used to identify the adopted model on a given trial and that RT reflected the adopted strategy, with the stimulus-based strategy resulting in longer RT than the action-based strategy.

### Influences of amygdala and ventral striatum lesions on learning and choice behavior

Next, we compared the effects of amygdala and VS lesions on choice behavior to elucidate their contributions to decision-making, learning,

and arbitration. During the What-only task, amygdala-lesioned monkeys exhibited the largest impairment in performance (*P*(*Better*)) across all three reward schedules: $M \pm SD = 0.581 \pm 0.11$; main effect of group in mixed-effects analysis; $\beta_{amyg} = -0.172$, $p = 7.72 \times 10^{-22}$; contrast between lesion groups: $\beta_{amyg} - \beta_{VS} = -0.076$, $p = 1.11 \times 10^{-4}$; Fig. 1e; Supplementary Table 1). VS-lesioned monkeys also showed impairment compared to the control monkeys (main effect of group in mixed-effects analysis; $\beta_{VS} = -0.096$, $p = 1.19 \times 10^{-6}$; Fig. 1e; Supplementary Table 1). Although previous study[17] have reported additional differences in the behavior of amygdala- and VS-lesioned monkeys, the better performance of VS- compared to amygdala-lesioned monkeys (Fig. 1e inset) is surprising, given the established role

of VS for stimulus-based learning[26,27], which is required for the What-only task.

During What blocks of the What/Where task, however, amygdala-lesioned monkeys performed significantly better than VS-lesioned monkeys (mixed-effects analysis, contrast between lesion groups: $\beta_{amyg} - \beta_{VS} = 0.104$, $p = 0.0152$; Fig. 1f inset; Supplementary Table 2), while both lesioned groups were impaired relative to control monkeys (amygdala: $\beta_{amyg} = -0.105$, $p = 0.00334$; VS: $\beta_{VS} = -0.209$, $p = 1.05 \times 10^{-7}$). In Where blocks that did not require stimulus-based learning, only amygdala-lesioned monkeys showed impairments in performance relative to controls (contrast from controls = −0.070, $p = 0.0173$; Fig. 1g inset; Supplementary Table 2). VS-lesioned performance was comparable to that of controls (contrast from controls = −0.037, $p = 0.242$) yet not significantly better than that of amygdala-lesioned monkeys (contrast between lesion groups = −0.033, $p = 0.351$).

Overall, these results demonstrate that in the absence of intrinsic uncertainty about the correct model of the environment and when this model was stimulus-based, VS-lesioned monkeys (with intact amygdala) were able to partially overcome the deficit in stimulus-based learning to a significantly larger degree than amygdala-lesioned monkeys. Under additional uncertainty about the correct model of the environment (the What/Where task), however, VS lesions caused significant impairment in What blocks only, consistent with the role of VS in stimulus-based learning. In contrast, amygdala-lesioned monkeys exhibited impaired performance in both What and Where blocks (but more strongly in What blocks) despite no clear evidence for the significant contribution of the amygdala to action-based learning (but see ref. 20), whereas there is action encoding in the amygdala and VS[28,29]. As noted in the previous work[18], the similar impairments during What and Where blocks observed in amygdala-lesioned monkeys cannot be explained by the amygdala's currently assumed role in stimulus-based learning.

These results are also puzzling because the higher performance of VS-lesioned compared to amygdala-lesioned monkeys in the What-only task suggests a stronger contribution of the amygdala to stimulus-based learning. However, the higher performance of amygdala-lesioned compared to VS-lesioned monkeys in What blocks of the What/Where task contradicts this idea. These findings hint at a potential role of the amygdala in arbitration between stimulus- and action-based learning, in addition to its known role in stimulus-based learning.

To study the relative adoption of the two learning strategies according to uncertainty of the reward environment, we examined the difference between ERDS$_{Stim}$ and ERDS$_{Action}$ (ΔERDS) by block types and reward schedules (Supplementary Fig. 6). We found that the reward uncertainty (measured as variance[13]) is predictive of the relative degree of adoption between stimulus- and action-based strategies. More specifically, in the What-only and What blocks of the What/Where task, animals' strategies became relatively more biased toward action-based (increasing ΔERDS) as the uncertainty of the reward schedule increased (Supplementary Fig. 6d, e). Consistently, in Where blocks, they tended to become relatively more stimulus-based under more uncertainty (decreasing ΔERDS; Supplementary Fig. 6f). These observations demonstrate that both control and lesioned monkeys adjust to reward uncertainty by exploring the incorrect model of the environment, even though they start from different baselines. Critically, amygdala-lesioned monkeys exhibited the smallest distinction between the two types of learning strategies (Supplementary Fig. 6d–f).

Finally, we also examined RT in the two lesioned groups and found consistent results to those of the control animals (Supplementary Note 1). Together, our findings suggest that VS lesions biased behavior toward action-based learning by impairing stimulus-based learning. In contrast, amygdala lesions resulted in more nuanced impairment of both stimulus- and action-based learning, as well as their coordination. To reveal the underlying mechanisms, we developed multiple computational models to fit the choice behavior of both control and lesioned monkeys.

## Mechanisms of arbitration between stimulus- and action-based learning systems

To uncover mechanisms underlying the interaction between the two systems, we developed several hybrid RL models to fit the choice behavior of control monkeys on a trial-by-trial basis (see Supplementary Table 15 for the list of all models). In the simplest model, signals from distinct action-based and stimulus-based learning systems were combined linearly using a fixed weight to control choice behavior. We also tested models with dynamic arbitration in which the relative weighting of the two systems, $\omega$, was updated on each trial based on the reliability of the two systems. Drawing on previous literature, we compared multiple methods for computing reliability: (1) the magnitude of the reward prediction error (|RPE|), (2) the value of the chosen option ($V_{cho}$), (3) discernibility between two competing options (|$\Delta V$|), and (4) the sum of value estimates within each system ($\Sigma V$) (see Eqs. 13–16 in "Methods"). Additionally, we considered a more general model in which the baseline (time-independent) ratio of value signals from the two learning systems (quantified by parameter, $\rho$) could be adjusted independently of $\omega$ (Fig. 2a; see Eq. 10 in "Methods"). As a result, this (Dynamic $\omega$-$\rho$) model relaxes the assumption that an increase in signal strength from one system (or equivalently, the sensitivity of decision-making to those signals) is matched by an equal decrease in signal strength from the other system, and vice versa. To determine the best model, we computed the goodness-of-fit using five-fold cross-validation (see *K-fold cross-validation of model performance* in "Methods").

Comparing the single-system models with the simplest two-system model, which assumes a fixed relative weighting for the two systems (RL$_{Stim+Action}$ + Static $\omega$), we found that the latter provided a better fit. Interestingly, this model improved the goodness-of-fit even in the What-only task, in which action learning was not predictive of reward. Overall, however, all the dynamic models provided a better fit than the model with fixed weighting. Ultimately, the Dynamic $\omega$-$\rho$ model, which uses the value of the chosen option ($V_{cho}$) to estimate reliability and incorporates a baseline weighting of the two systems quantified by $\rho$, provided the best fit across all tasks and for each monkey (Fig. 2b; Supplementary Table 16).

To gain more insight into how dynamic arbitration improves the fit of choice behavior, we next examined the behavior of the Dynamic $\omega$-$\rho$ model and its arbitration weights over time. To that end, we computed the effective arbitration weight ("effective" $\omega$ denoted by $\Omega$) to measure the overall relative weighting between two systems considering the parameter $\rho$ (see Eq. 12 in "Methods"). Both the example block and the averaged trajectories of trial-by-trial $\Omega$ from the best model (Fig. 2c, d) showed dependence on the block type and uncertainty in the reward schedule, especially during the What/Where task that required arbitration between competing models of the environment. These results demonstrate that $\Omega$ can capture behavioral adjustments to uncertainty over time.

As shown above, stimulus-based choices lead to slower RTs (Supplementary Note 1). Motivated by this finding, we tested whether the Dynamic $\omega$-$\rho$ model could capture the differences in RT according to the dominant learning system. To that end, we computed the correlation between the median RTs of the block and the average estimated values of $\Omega$, which measures the overall relative weighting of the stimulus-based to action-based system. For the What-only task, we found a small yet significant correlation between the effective arbitration weight and RT (Spearman's $r = 0.094$, $p = 1.24 \times 10^{-4}$). In comparison, $\Omega$ and RT were highly correlated in the What/Where task ($r = 0.414$, $p = 3.78 \times 10^{-245}$; Fig. 2e). However, because these simple

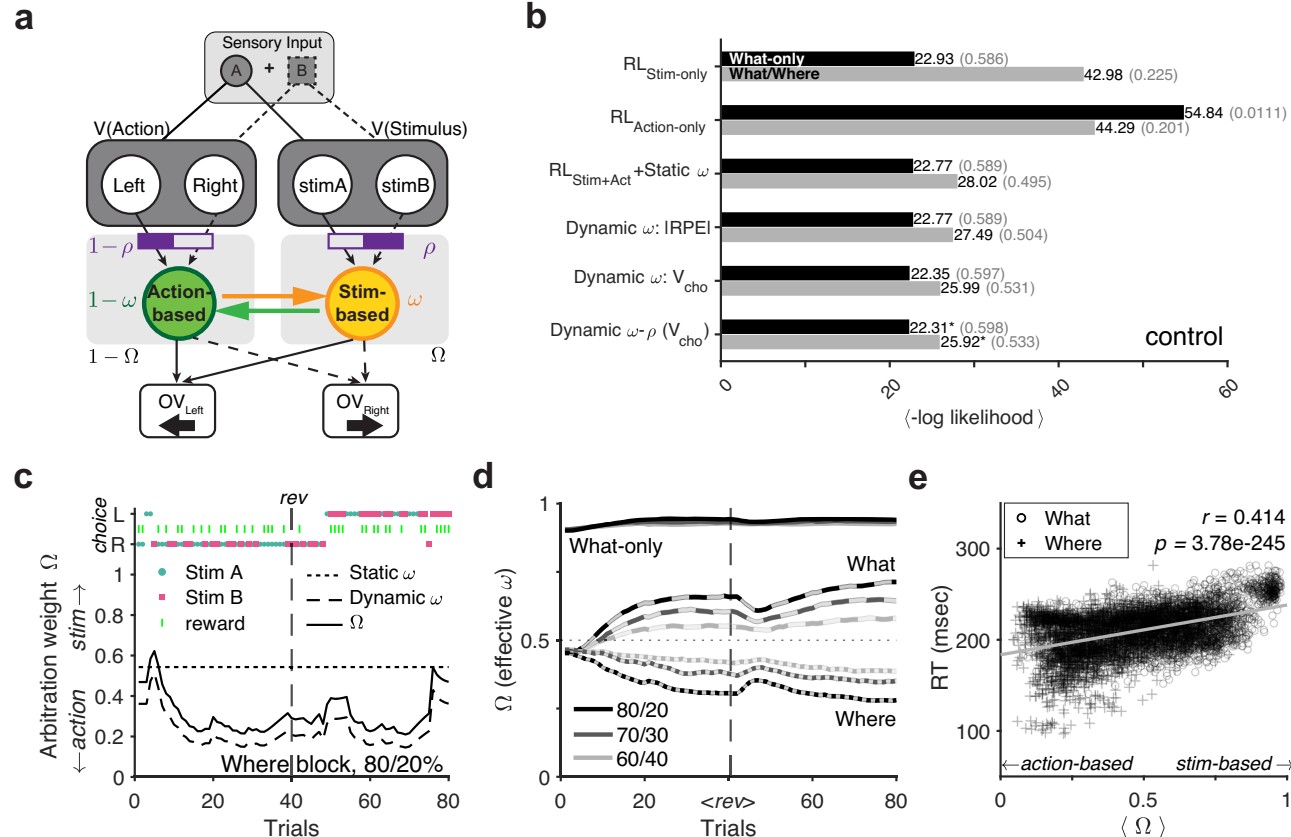

**Fig. 2 | Fit of choice behavior of control monkeys using various RL models.**
**a** Schematic of the RL model with two parallel learning systems, showing an example trial in which stimulus A appeared on the left side. In the static model, a constant $\omega$ is assumed to be fixed for each block of trials. In the dynamic models, $\omega$ is updated on each trial according to the relative reliability of the two systems. In a more general dynamic model, the fixed parameter $\rho$ (estimated for each subject) adjusts the baseline ratio of two value signals. For $\rho = 0.5$, the Dynamic $\omega$-$\rho$ model reduces to the Dynamic $\omega$ model. The overall value (OV) of a left or right saccade is determined as a weighted combination of action and stimulus values.
**b** Comparison of goodness-of-fit across models. Plotted is the mean negative log-likelihood over all cross-validation instances for each task: What-only (black), What/Where (gray). Numbers in parentheses indicate McFadden $R^2$ (Eq. 20). **c** An example Where block in the What/Where task and estimated arbitration weight from the

Static $\omega$ model (dotted line), and arbitration weights ($\omega$, dashed line) and effective arbitration weights ($\Omega$, solid line) from the best model (Dynamic $\omega$-$\rho$ model). In this example, $\rho = 0.61$, effectively biasing behavior toward a stimulus-based strategy. In this block, rightward action (R) was a better option than leftward action (L) before reversal (rev, horizontal dashed line). **d** Average trajectory of $\Omega$ from the Dynamic $\omega$-$\rho$ model during different tasks and blocks: What-only (solid), What (dashed), and Where (dotted). Different colors correspond to different reward schedules: 80/20 (black), 70/30 (dark gray), 60/40 (light gray). <rev> indicates reversal (horizontal dashed line), with positions normalized across blocks. **e** Relationship between $\Omega$ (block-averaged) and median reaction time for a given block during the What/Where task. Reported are Spearman's correlation coefficient $r$ and its $p$-value (two-sided) for all blocks during the What/Where task. Source data are provided as a Source data file.

correlations do not control for other confounding factors such as choice confidence (measured by value difference), choice accuracy (choosing the better or worse option), and long-term drift in RT, we conducted further regression analyses. These analyses included these factors along with other model-derived measures to predict trial-by-trial RT (Supplementary Note 2). Using these analyses, we found that across all groups and conditions, higher $\Omega$ predicted longer RT (Supplementary Note 2). Together, these results indicate that slower RTs occurred when larger weights were assigned to the stimulus-based system, and faster RTs occurred with larger weights on the action-based system. This is consistent with the previous analysis, which showed that action-dominant trials (determined using ERDS) were accompanied by faster RTs.

As part of our exploration of arbitration mechanisms, we also compared multiple algorithms for estimating the reliability of the two systems, including $V_{cho}$, |RPE|, discernibility between two competing options (|$\Delta V$|), and the sum of value estimates within each system ($\Sigma V$) (see Eqs. 13–16 in "Methods"). We found that among the four reliability measures considered, $V_{cho}$ best explained the control monkeys' choice behavior across all block types (Supplementary Fig. 7a). Specifically,

the Dynamic $\omega$ model based on $V_{cho}$ improved the fit over the Static $\omega$ model in all tasks, whereas the Dynamic $\omega$ model based on |RPE| improved the fit over the Static $\omega$ model only in the What/Where task (Fig. 2b). Through model recovery, we confirmed that our model fitting procedure could effectively discriminate between the alternative one-system and two-system models (Supplementary Fig. 8a–d).

Importantly, $V_{cho}$ and |RPE| are conceptually related as both signals measure the predictiveness of reward values in each system. However, the two signals differ in their sensitivity to negative feedback (i.e., no reward). For example, when the chosen value of the more reliable system (e.g., stimulus-based system in What blocks) is (correctly) estimated to be high, negative RPE (and consequently |RPE|) will also be high, undesirably facilitating the update toward the incorrect system. In comparison, $V_{cho}$ by itself is less sensitive to negative feedback, as the updated value of $V_{cho}$ after omission of reward will still reflect the high value estimates for the more reliable system. As a result, the $V_{cho}$ signal distinguishes the more reliable system better than the |RPE| signal, especially for more uncertain reward schedules (Supplementary Fig. 7b–d). Ultimately, the difference in the $V_{cho}$ drives the arbitration process (Eqs. 8–9), and this difference is equal to the difference in

*signed* RPE. This suggests that the reliability signal could be more connected to signed rather than unsigned RPE.

Finally, to further validate our models, we compared the predictions of different models regarding the observed negative interaction between ERDS$_{Stim}$ and ERDS$_{Action}$ during the What-only task. This is to ensure that this relationship was due to competition between the two learning systems and not due to task structure, as the animals could not stay/switch on the stimuli and location dimension at the same time, while positions of stimuli were pseudo-randomly assigned to either side. To that end, we simulated choice behavior using single-system or two-system models and computed regression weights between ERDS$_{Stim}$ and ERDS$_{Action}$ (see *Model fitting and simulation* in "Methods" for more details). Competition between the two learning systems during the What-only task would suggest that weaker stimulus-based learning corresponds to stronger action-based learning and vice versa. We found that in the single-system model that learned the stimulus-outcome contingencies only (RL$_{Stim-only}$), ERDS$_{Stim}$ was only weakly predictive of ERDS$_{Action}$ (Supplementary Fig. 9b; $\beta = -0.007$, $p = 1.07 \times 10^{-44}$). In contrast, in both the static and dynamic two-system models, ERDS$_{Stim}$ was negatively predictive of ERDS$_{Action}$, thus reproducing the competitive interaction between the two systems (Supplementary Fig. 9c; $\beta = -0.0648$, $p = 1.09 \times 10^{-56}$; Supplementary Fig. 9d; $\beta = -0.0893$, $p = 6.73 \times 10^{-57}$). These simulation results further support the presence of multiple learning systems and their dynamic interaction, even in an environment where one of the two systems was not beneficial for performing the task.

### Deficits in arbitration due to amygdala but not VS lesions

Fit of choice behavior of lesioned monkeys revealed that the Static $\omega$ model explained the choice behavior of both lesioned groups better than models with a single learning system (Fig. 3a). Moreover, incorporating dynamic arbitration as in the Dynamic $\omega$ model further improved the fit beyond what the Static $\omega$ model achieved in both groups. Furthermore, including baseline relative weighting, as in the Dynamic $\omega$-$\rho$ model, resulted in the best overall fit (Fig. 3a). Finally, consistent with the results in controls, for both lesioned groups, the dynamic model that used V$_{cho}$ to estimate reliability accounted for choice behavior better than the model using |RPE| for estimating reliability (Supplementary Fig. 7a).

To determine the mechanisms by which different lesions impact the arbitration process, we examined the estimated parameters in the best model. We first confirmed that the parameters of this model were recovered well (Supplementary Fig. 8e, f). The estimated trajectory of $\Omega$ revealed that, similar to controls, arbitration was modulated by reward uncertainty during the What/Where task in both amygdala-lesioned (Fig. 3b) and VS-lesioned monkeys (Fig. 3c). Nonetheless, $\Omega$ values were overall smaller than in controls, corresponding to a more action-based strategy in lesioned animals (compare Fig. 3b, c and Fig. 2d). Importantly, a key difference between the two lesioned groups demonstrates deficits in the arbitration process due to amygdala lesions. During the What/Where task, the effective arbitration weight ($\Omega$) for amygdala-lesioned monkeys increased over time in both What and Where blocks (dashed and dotted curves in Fig. 3b). In contrast, VS-lesioned monkeys showed an increase in $\Omega$ during What blocks and a decrease in during Where blocks (dashed and dotted curves in Fig. 3c), mirroring the pattern observed in control animals (dashed and dotted curves in Fig. 2d). Meanwhile, during the What-only task, $\Omega$ remained stable but at lower values for amygdala-lesioned (solid curves in Fig. 3b) compared to VS-lesioned monkeys (solid curves in Fig. 3c), and significantly lower than in controls (solid curves in Fig. 2d).

These results demonstrate that VS lesions biased behavior toward action-based learning while keeping the arbitration processes relatively intact, whereas amygdala lesions impaired arbitration in addition to biasing behavior toward action-based learning. These suggest that the deficits observed in amygdala-lesioned monkeys cannot solely be

attributed to impairments in stimulus-based learning; instead, they involve a more complex interaction between stimulus- and action-based signals. Consistent with this interpretation, we found that the winning model with dynamic arbitration (Dynamic $\omega$-$\rho$) more accurately captures the key aspects of behavioral strategy in the two lesioned groups compared to the single-system models (Supplementary Fig. 10).

To further investigate the dynamics of the arbitration weight, we next examined the rate of change in $\Omega$ across the three groups. To that end, we calculated the "effective" arbitration rates by calculating the ratio of the overall change in $\Omega$ toward 1 (favoring stimulus-based system) or 0 (favoring action-based system) relative to its original value (Fig. 3d; see Eqs. 17–19 in "Methods"). This quantity measures the rate of arbitration, analogous to the learning rate for updating value estimates. We found that in control monkeys, the effective arbitration rates toward the stimulus-based ($\psi_+$) or action-based ($\psi_-$) system diverged toward the end of a block, reflecting the adoption of the correct model of the environment. That is, when the stimulus-based system was more reliable, the effective arbitration rates toward the stimulus-based system were larger than those toward the action-based system (mixed-effects model with a single fixed intercept, representing the mean $\Delta\psi$; What-only task: $\beta_0 = 0.0930$, $p = 1.24 \times 10^{-102}$; What blocks of What/Where task: $\beta_0 = 0.0444$, $p = 1.59 \times 10^{-9}$; Fig. 3e, f; Supplementary Tables 3 and 4). Similarly, in Where blocks, where the action-based system was more reliable, the effective arbitration rate toward the action-based system was significantly larger than that toward the stimulus-based system ($\beta_0 = -0.0301$, $p = 0.00968$; Fig. 3g).

In contrast, amygdala-lesioned monkeys showed the minimum differentiation between adjustments toward the more and less reliable (correct and incorrect) learning systems. Notably, during the What/Where task, amygdala-lesioned monkeys exhibited no significant difference between the two arbitration rates during either block type (mixed-effects analysis on $\Delta\psi$ with a single intercept; What: $\beta_0 = 0.0114$, $p = 0.218$; Where: $\beta_0 = -0.00347$, $p = 0.824$; Fig. 3f, g; Supplementary Table 4). In contrast, VS-lesioned monkeys exhibited an overall large bias in arbitration rates toward the action-based system (i.e., higher $\psi_-$) during both block types (mixed-effects analysis on $\Delta\psi$ with a single intercept; What: $\beta_0 = -0.0536$, $p = 3.95 \times 10^{-7}$; Where: $\beta_0 = -0.0739$, $p = 2.03 \times 10^{-5}$; Fig. 3f, g; Supplementary Table 4). Overall, amygdala-lesioned monkeys were characterized by the least amount of differentiation between the two arbitration rates (mixed-effects analysis on |$\Delta\psi$|; contrast from controls = −0.0254, $p = 0.0114$; contrast from VS = −0.0282, $p = 0.0177$; Supplementary Table 5).

In the What-only task, with reduced uncertainty about the model of the environment, the difference between the two arbitration rates in amygdala-lesioned monkeys was positive (contrast on group means = 0.0142, $p = 0.00116$; Fig. 3e; Supplementary Table 3) but much smaller than that of control monkeys (main effect of group in mixed-effects analysis; $\beta_{amyg} = -0.0786$, $p = 1.15 \times 10^{-37}$). The VS-lesioned group also exhibited higher arbitration rates toward the stimulus-based system (contrast on group means = 0.0226, $p = 1.20 \times 10^{-4}$), which aligns with the recovered performance observed in these monkeys.

Together, these results suggest that amygdala lesions impair arbitration between the two learning systems by eliminating differential updates for the correct and incorrect (more and less reliable) systems. This indicates that the amygdala is critical for identifying and/or retaining the correct model of the environment, or biasing arbitration toward it. In contrast, VS lesions mainly impair stimulus-based learning and increase the overall arbitration bias toward action-based learning.

### Dynamic interaction between learning and arbitration processes and the impact of the initial state

Considering the observed effects of amygdala and VS lesions on arbitration dynamics, we next examined the estimated parameters

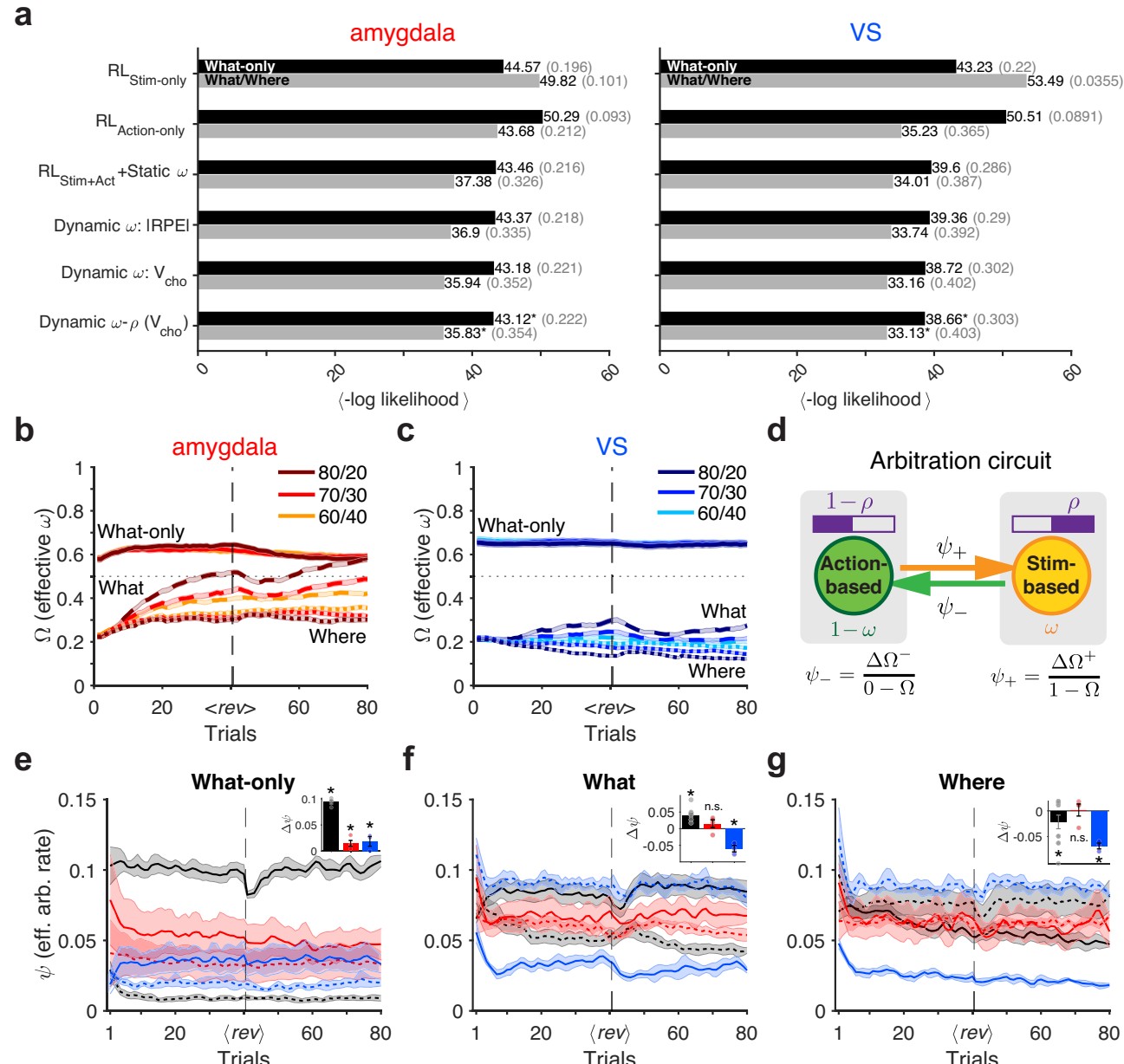

**Fig. 3 | Fit of choice behavior for amygdala- and VS-lesioned monkeys.**
**a** Comparison of the models' goodness-of-fit for the choice behavior of amygdala-lesioned (left) and VS-lesioned (right) monkeys. Plotted is the mean negative log-likelihood over all cross-validation instances for each task: What-only (black), What/Where (gray). Numbers in parenthesis indicate McFadden $R^2$. Averaged trajectory of estimated $\Omega$ (effective $\omega$) in amygdala-lesioned (**b**) and VS-lesioned (**c**) monkeys, separately for each block type and reward schedule. Solid, dashed, and dotted curves indicate What-only, What, and Where blocks, respectively. <rev> indicates reversal (horizontal dashed line), normalized across blocks. Colors indicate different reward schedules: 80/20 (brown and navy), 70/30 (red and blue), 60/40 (orange and cyan). **d** Schematic of effective arbitration rates. $\psi_+$ and $\psi_-$ represent the rate of update toward the stimulus-based (increase in $\Omega$) and action-based system (decrease in $\Omega$). **e** Plotted are the time courses of effective arbitration rates toward the stimulus-based or action-based system during the What-only task for each group of monkeys (controls: black; VS-lesioned: blue; amygdala-lesioned: red). Solid and dotted lines indicate effective arbitration rates toward stimulus-

based ($\psi_+$) and action-based systems ($\psi_-$), respectively. Shaded regions indicate error bars. Insets show the mean paired differences between the two arbitration rates within each block after reversal ($\Delta\psi = \psi_+ - \psi_-$). Asterisks indicate significant difference from zero within each group as determined by mixed-effects analysis ($p < 0.05$, two-sided, corrected for multiple comparisons using Benjamini–Hochberg procedure; control: $p = 1.24 \times 10^{-102}$; amygdala: $p = 0.00116$; VS: $p = 1.20 \times 10^{-4}$; see Supplementary Table 3 for the full statistics). Individual data points represent the mean of each monkey. Error bars = SEM across subjects (control: $n = 4$; amygdala: $n = 4$; VS: $n = 3$). **f** Same plot as in (**e**) but for What blocks of the What/Where task (control: $p = 1.59 \times 10^{-9}$; amygdala: $p = 0.218$; VS: $p = 3.95 \times 10^{-7}$; see Supplementary Table 4 for the full statistics). **g** Same plot as in (**e**) but for Where blocks of the What/Where task (control: $p = 0.00968$; amygdala: $p = 0.824$; VS: $p = 2.03 \times 10^{-5}$; see Supplementary Table 4 for the full statistics). Error bars = SEM across subjects (control: $n = 6$; amygdala: $n = 4$; VS: $n = 3$). Source data are provided as a Source data file.

from the best-fit model (Dynamic $\omega$-$\rho$). In this model, $\rho$ captures whether there is an overall reduction in baseline value signals from the stimulus-based system relative to the action-based system. Consistent with the hypothesized role of VS in stimulus learning, the estimated

values of $\rho$ were on average smaller in VS-lesioned monkeys compared to controls (permutation test for difference in group mean; $p = 0.0044$), indicating a larger baseline reduction in stimulus-value signals relative to action-value signals in VS-lesioned monkeys (Fig. 4a).

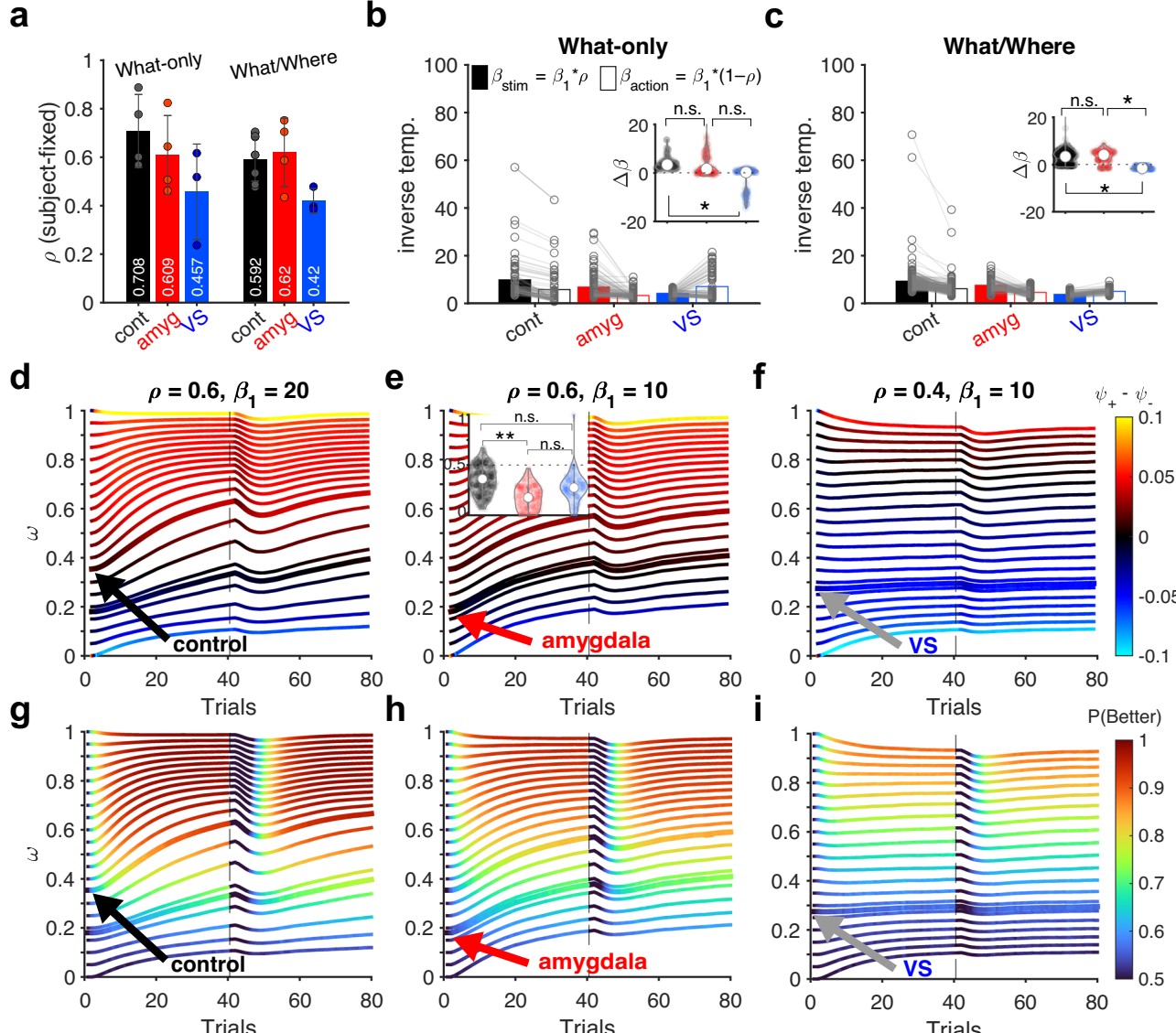

**Fig. 4 | Comparison of relative weighting and sensitivity of choice to stimulus- and action-based signals across control and lesion groups, and their simulations. a** Plots show the mean and individual values (each point represents a monkey) of the relative strength of two systems on choice ($\rho$), separately for each group and task. Error bars = SEM across subjects (controls: $n = 4$ in What-only, $n = 6$ for What/Where; amygdala: $n = 4$ for both tasks; VS: $n = 3$ for both tasks). **b** Comparison of the sensitivity of choice to signals in the two systems, separately for each group during the What-only task. $\beta_1$ is the common sensitivity of choice (inverse temperature) estimated for each session, and $\rho$ is the relative strength of the two systems as in (a). Insets show violin plots of the paired differences ($\Delta\beta = \beta_{stim} - \beta_{action}$), and asterisks indicate significant group effects as determined by mixed-effects analysis ($p < 0.05$, two-sided, corrected for multiple comparisons using Benjamini–Hochberg procedure; cont vs. amyg: $p = 0.280$; amyg vs. VS: $p = 0.080$; cont vs. VS: $p = 0.00658$; see Supplementary Table 6 for the full statistics). **c** Same plot as in (**b**) but for the What/Where task (cont vs. amyg: $p = 0.719$;

amyg vs. VS: $p = 0.0232$; cont vs. VS: $p = 0.0333$; see Supplementary Table 7 for the full statistics). Only VS-lesioned monkeys showed larger sensitivity to action- than stimulus-based systems during both tasks, consistent with the role of the VS in stimulus-based learning. **d**–**f** Plots show the simulated trajectory of $\omega$ and the difference in the effective arbitration rates ($\psi_+ - \psi_-$). Each line represents averaged trajectories (10,000 simulated blocks) of $\omega$ with different initial values ($\omega_0$) and specified values of $\rho$ and $\beta_1$ during What blocks. All other parameters are fixed ($\alpha_+ = \alpha_- = 0.5$, $\beta_0 = 0$, $\zeta = 0.3$, $\alpha_\omega = 0.2$, $\zeta_\omega = 0.05$). Black, red, and gray arrows show the trajectory simulated with $\omega_0$ equal to the mean of control, amygdala-lesioned, and VS-lesioned monkeys, showing $\psi_+ > \psi_-$, $\psi_+ \approx \psi_-$, and $\psi_+ < \psi_-$, respectively. Horizontal line (trial 40) indicates reversal. Inset in (e) shows distributions of $\omega_0$ across the three groups during the What/Where task, and asterisks indicate significant group difference (mixed-effects analysis). **g**–**i** Plots show the simulated trajectory of $\omega$ and performance, $P(Better)$. Conventions are the same as in (**d**–**f**). Source data are provided as a Source data file.

In contrast, we found no such evidence for reduction in $\rho$ in amygdala-lesioned monkeys relative to controls (permutation test for difference in group mean; $p = 0.869$). As a result, VS-lesioned monkeys exhibited a smaller difference in choice sensitivity to stimulus- and action-value signals ($\Delta\beta = \beta_{stim} - \beta_{action}$), with a bias toward action-value signals, compared to controls in both the What-only task (mixed-effects analysis on $\Delta\beta$, main effect of group; $\beta_{VS} = -7.48$, $p = 0.00658$; Fig. 4b inset; Supplementary Table 6) and the What/Where task ($\beta_{VS} = -3.53$,

$p = 0.0333$; Fig. 4c inset). This was not the case for amygdala-lesioned monkeys, which exhibited no significant difference compared to controls in either task (What-only: $\beta_{amyg} = -2.69$, $p = 0.280$; Fig. 4b inset; What/Where: $\beta_{amyg} = 0.542$, $p = 0.719$; Fig. 4c inset; Supplementary Tables 6 and 7). These results suggest that, unlike VS lesions, amygdala lesions did not significantly alter the relative baseline strength of stimulus-value vs. action-value signals. Instead, amygdala lesions reduced sensitivity to both systems. Therefore, consistent with

previous observations, the deficit observed in amygdala-lesioned monkeys cannot be solely attributed to impairments in stimulus-based learning. Instead, they suggest deficits in arbitration processes that subsequently affect learning and decision making.

To confirm this point, we examined the initial arbitration weights that determine the weights of the two systems on choice at the beginning of each block, when the monkeys were unaware of the correct model of the environment during the What/Where task. We note that amygdala- and VS-lesioned monkeys did not significantly differ in the initial $\Omega$ (effective $\omega$) values (planned contrast for group difference in $\Omega_0 = 0.0242$, $p = 0.657$; Supplementary Table 8; compare $\Omega$ of the first trial in Fig. 3b, c). However, by examining the initial arbitration weights ($\omega_0$) before scaling by $\rho$, we found that amygdala-lesioned monkeys had significantly smaller $\omega_0$ values compared to controls (mixed-effects analysis on $\Omega_0$; $\beta_{amyg} = -0.194$, $p = 0.00419$) while the VS-lesioned group did not ($\beta_{VS} = -0.119$, $p = 0.108$; Supplementary Table 9). More directly, the amygdala group showed larger changes in $\omega_0$ after scaling by $\rho$ compared to the VS group (group contrast in mixed-effects analysis on $\Omega_0 - \omega_0 = 0.111$, $p = 0.0204$; Supplementary Table 10). This means that larger values of $\rho$ in amygdala-lesioned monkeys were offset by lower $\omega_0$ values to yield $\Omega_0$ comparable to VS-lesioned monkeys. Therefore, deficits due to amygdala lesions can be mainly attributed to the reduction in the initial weight ($\omega_0$) and subsequent interaction between arbitration and learning processes. In contrast, deficits due to VS lesions are largely caused by a reduction in the relative baseline strength of stimulus-value to action-value signals, measured by $\rho$.

To further validate this idea through model simulations, we generated the choice behavior of the Dynamic $\omega$-$\rho$ model by adjusting two key parameters to mimic the effects of brain lesions: baseline ratio of the weights of the stimulus- to action-value signals ($\rho$) and the initial arbitration weight ($\omega_0$). These two parameters reflected the most consistent effect of lesions across the two tasks, with reduced $\rho$ in VS-lesioned monkeys (Fig.4a–c; $\beta_{VS} = -0.195$, $p = 0.00279$; mixed-effects analysis on compiled $\rho$ across all groups/tasks) and reduced $\omega_0$ in amygdala-lesioned monkeys ($\beta_{amyg} = -0.191$, $p = 0.0171$; mixed-effects analysis on compiled $\omega_0$ across groups/tasks). We kept all other parameters constant except for the common inverse temperature, $\beta_1$.

Trajectories of simulated $\omega$ during What blocks revealed that different values of $\rho$ and $\omega_0$ can create different dynamics with respect to arbitration rates (Fig. 4d–f). More specifically, the simulated trajectory of $\omega$ based on mean $\omega_0$ in control monkeys during the What/Where task (Fig. 4d, black arrow) resulted in larger transitions toward the stimulus-based system ($\psi_+ > \psi_-$), whereas mean $\omega_0$ in amygdala-lesioned monkeys (Fig. 4e, red arrow) reduced the distinction between the two arbitration rates ($\psi_+ \approx \psi_-$). In comparison, simulations using mean $\omega_0$ in VS-lesioned monkeys (Fig. 4f, gray arrow) resulted in an overall update bias toward the action-based system ($\psi_+ < \psi_-$). These results qualitatively mimic the pattern of effective arbitration rates in the three groups (Fig. 3e–g).

Finally, we also tested the causal contribution of initial arbitration weight on performance using simulated choice behavior (Fig. 4g–i). Crucially, we found that lower values of $\omega_0$, as observed in amygdala-lesioned monkeys ($\omega_0 = 0.18$; red arrows in Fig. 4h), lead to reduced performance when compared to higher $\omega_0$ values (e.g., $\omega_0 = 0.40$). This effect was reflected in a significant main effect of $\omega_0$ on the simulated performance ($F_{(20,1659)} = 40.6$, $p = 8.38 \times 10^{-128}$). These simulation results demonstrate that reduced flexibility in the arbitration process--reflected by a lower $\omega_0$--could be the main cause of impaired performance, rather than just a secondary consequence.

Although control monkeys were also biased toward the action-based system at the start of the What/Where task (mean ± s.e.m; $\omega_0 = 0.374 \pm 0.058$), lesions to the amygdala resulted in an even larger bias toward the action-based system ($\omega_0 = 0.179 \pm 0.053$), and this consequently led to a lack of differential updates for the two systems.

Therefore, our simulations indicate that amygdala-lesioned monkeys operate within a parameter space that produces smaller differences in arbitration rates favoring the correct system for a given environment, which ultimately reduces performance. Overall, these results suggest that the initial state of the system ($\omega_0$) is crucial for determining the later trajectory and rates of transition in the arbitration process.

In contrast, lesions to VS mainly decreased $\rho$ to bias signals toward the action-based system, while affecting the initial state of arbitration to a lesser degree ($\omega_0 = 0.276 \pm 0.042$). It is worth noting that the simulations using $\rho = 0.4$ (Fig.4f, i), which mimics the reduction in the relative baseline strength of stimulus-value signals due to VS lesions, result in the biased update rates toward the action-based system for many of the $\omega_0$ values (blue lines in Fig.4f), including $\omega_0 \sim 0.37$, which matches the initial values for control monkeys. Therefore, the consistent adoption of an action-based strategy in the What/Where task can be sufficiently accounted for by a reduced $\rho$ value, without the need for additional constraints on $\omega_0$. These results support the notion that the impairments observed in VS-lesioned monkeys during this task can be fully explained by a reduction in stimulus-value signals, with minimal direct impact on arbitration processes.

## Diversity of behavior driven by the dynamic interaction between learning and arbitration processes

To demonstrate the impact of dynamic interaction between the learning and arbitration processes on behavior, we simulated the model within the task by adjusting parameters such as the learning and forgetting rates. These simulations revealed a wide range of dynamics in performance and arbitration weights, highlighting complex interactions between learning, arbitration, and decision-making processes (Fig. 5). Interestingly, we observed that higher initial arbitration weights, which would allow the animals to correctly bias their behavior toward the stimulus-based system during a stimulus-learning task, can both facilitate and impede learning after reversals depending on other parameters of the model.

However, in most cases, a larger initial bias toward the stimulus-based system helps both initial learning of stimuli and their reversals (Fig. 5a, c–f).

However, in scenarios where the positive learning rate significantly exceeds the negative learning rate, a smaller initial arbitration weight—though it may incorrectly bias behavior toward an action-based strategy—can actually facilitate adjustments to reversals in stimulus values (Fig. 5b). This happens because lower values of $\omega_0$, as in the case of amygdala lesions, result in a dependence of choice on both stimulus- and action-based signals and thus, more explorations that greatly benefit response to reversals. These results, based on our best dynamic arbitration model, can thus explain the paradoxical improvements in performance observed following amygdala lesions or inactivation.

## Contribution of the amygdala to long-term adjustments of behavior

Lesions to certain brain areas are often accompanied by adjustments or compensation by other brain areas that result in reducing initial behavioral impairments over the long term. Considering the observed effects of amygdala and VS lesions on learning and decision-making behavior, we investigated long-term adjustments in these behaviors in the absence of task-imposed, objective uncertainty about the correct model of the environment. To that end, we examined ERDS, median RT, and initial arbitration weight across all sessions of the What-only task using the proportion of sessions completed as an independent variable (Methods).

For consistency in stimulus-based strategy, we observed a long-term decrease in ERDS_Stim in VS-lesioned monkeys (planned contrast for the slope of VS group = $-0.626$, $p = 4.94 \times 10^{-324}$; Supplementary Fig. 11a), to a significantly greater extent than control monkeys

 

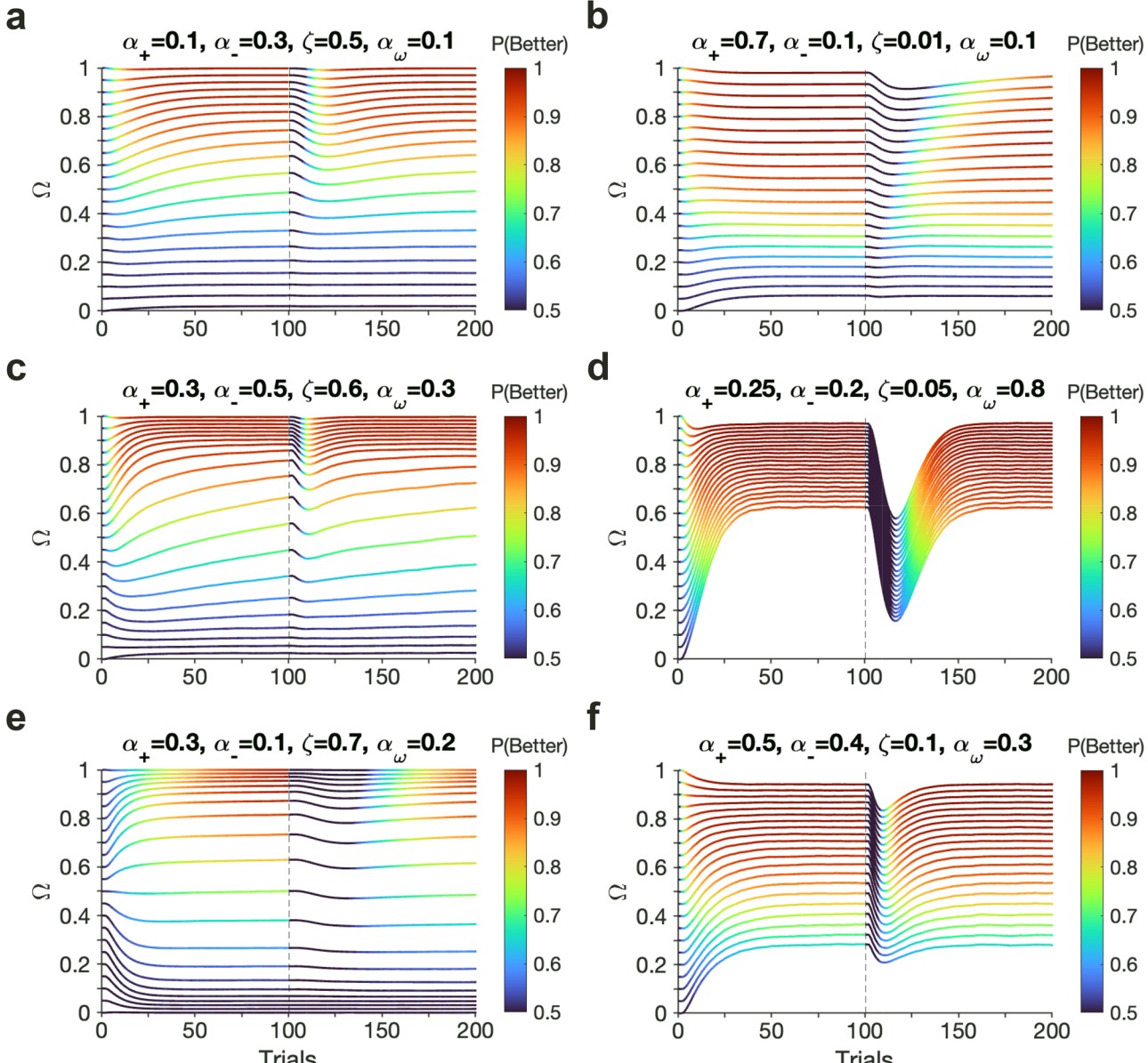

**Fig. 5 | Complex interaction between arbitration and learning gives rise to diverse behavioral patterns.** Each line represents averaged trajectories (10,000 simulated blocks) of $\Omega$ with different initial values during a stimulus-based learning task with reversal at trial 100. All simulations were performed with $\rho = 0.5$, causing $\Omega = \omega$. **a–f** Different behavioral patterns based on simulation of choice behavior using different model parameters, as indicated on the top. All non-specified parameters are fixed across panels at $\beta_1 = 20$, $\beta_0 = 0$, and $\zeta_\omega = 0.05$. **a** Larger initial values ($\omega_0$) facilitate learning after reversal. **b** Larger initial values ($\omega_0$) impede learning after reversal. **c** Initial values $\omega_0 > 0.15$ increase stable points of $\omega$ toward 1, whereas small $\omega_0$ (<0.15) results in low performance. **d** Small decay or forgetting for the unchosen option ($\zeta$) and large transition rate ($\alpha_\omega$) facilitates arbitration toward the correct model. **e** For certain model parameters, bifurcation of trajectories happens around $\omega$ of 0.5. **f** Steady state of arbitration is controlled by the initial value. Source data are provided as a Source data file.

($\beta_{VS:sess\%} = -0.549$, $p = 7.78 \times 10^{-12}$, controls as a reference group; Supplementary Table 11). There was no evidence for such adjustment in control ($\beta_{cont:sess\%} = -0.077$, $p = 0.0881$) or amygdala-lesioned monkeys across time (planned contrast for the slope of amygdala group = −0.033, $p = 0.478$). Specifically, despite their impaired stimulus-based learning, monkeys with VS lesions were able to increase their adoption of stimulus-based strategy over time. Consistently, these monkeys also decreased their adoption of action-based strategy as reflected in the positive slope of ERDS$_{Action}$ over time (planned contrast for the slope of VS group = 0.192, $p = 3.49 \times 10^{-4}$; Supplementary Fig. 11b), which was significantly greater compared to controls ($\beta_{VS:sess\%} = 0.211$, $p = 0.00208$; Supplementary Table 12). There was no evidence of such an effect in control monkeys ($\beta_{cont:sess\%} = -0.019$, $p = 0.652$) or in monkeys with amygdala lesions (planned contrast for the slope of amygdala

group = 0.014, $p = 0.748$). Interestingly, consistent with previous results, the complementary changes in model adoption in VS-lesioned monkeys were also reflected in increased median RT over time in these monkeys (planned contrast for the slope of VS group = 30.7, $p = 1.61 \times 10^{-6}$; Supplementary Fig. 11c; Supplementary Table 13) to greater extent than controls ($\beta_{VS:sess\%} = 24.3$, $p = 0.00236$) or amygdala-lesioned monkeys (planned contrast for group difference in slopes = −29.6, $p = 0.00224$). This was accompanied by a long-term increase in the initial effective arbitration weights $\Omega_0$ (toward stimulus-based system) only in the VS-lesioned monkeys (planned contrast for the slope of VS group = 0.455, $p = 2.88 \times 10^{-4}$; Supplementary Fig. 11d; Supplementary Table 14), which was significantly greater compared to both controls ($\beta_{VS:sess\%} = 0.568$, $p = 5.42 \times 10^{-5}$) and amygdala group (planned contrast for group difference in slopes = −0.431, $p = 0.00183$).

These results provide evidence for adjustments on a long time-scale in VS-lesioned but not amygdala-lesioned monkeys. They suggest that, in the absence of additional uncertainty about the model of the environment, intact amygdala in VS-lesioned monkeys (and not intact VS in amygdala-lesioned monkeys) enabled these animals to slowly improve their performance over time. This amygdala-driven mechanism enabled VS-lesioned monkeys to gradually suppress action-based strategy, resulting in an increase in overall RT and initial effective arbitration weight ($\Omega_0$) over time.

## Discussion

Here, we applied a combination of computational approaches to re-analyze data from control monkeys and those with amygdala and VS lesions[17,18,24] to explore the interaction between stimulus- and action-based learning and to uncover computational and neural mechanisms underlying arbitration processes. Our main goal was to investigate the interaction between stimulus- and action-based learning systems, instead of examining them in isolation as in the original studies. Using multiple behavioral metrics, we found evidence for competitive interaction between the two learning systems. Moreover, by developing various models with arbitration and fitting choice data to these and competing models, we tested the plausibility of various mechanisms for estimating reliability signals that guide arbitration processes. Using this approach, we mapped the distinct effects of two brain lesions onto two key parameters of the model: the initial state of the arbitration ($\omega_0$) for amygdala lesions and the relative baseline strength of stimulus-value to action-value signals ($\rho$) for VS lesions.

For amygdala-lesioned monkeys, the reduced initial arbitration weight was implicated in undifferentiated arbitration rates toward and away from the correct learning system for a given environment. This suggests that the amygdala may have a role in identifying and retaining the correct model of the environment, or in biases model arbitration toward the correct model of the environment. Our model simulations also illustrated that the interaction between learning and arbitration processes generates diverse behaviors with strong dependency on the initial state.

Previous studies using the same dataset have identified deficits in both stimulus-based and action-based learning due to amygdala lesions[17,18], but they considered these deficits independently, as they did not examine the interaction between stimulus- and action-based learning. Using a single-system RL model, they found that amygdala lesions reduce choice consistency (sensitivity to value signals) for stimulus-based learning[17] and increase sensitivity to negative feedback ($\alpha_-$) for action-based learning[18]. We found results using our two-system model with dynamic arbitration (Supplementary Figs. 12b, 13d and 14–16) Moreover, we provided a unified account of the monkeys' choice behavior based on the dynamic interaction between learning and arbitration under uncertainty. Specifically, our simulation results mimicking amygdala lesions (Fig. 4) suggest that a biased initial state strongly favoring action-based learning is the key feature of the deficits observed in amygdala-lesioned monkeys. This strong initial bias altered the interaction between decision-making, learning, and arbitration processes, making the arbitration update rates between the two systems to be less distinguishable from each other compared to the controls. Importantly, VS-lesioned monkeys exhibited a smaller sensitivity to the stimulus-based compared to the action-based signals. As a result, VS lesions led to an overall bias in arbitration update rates toward action-based learning in both What and Where blocks.

More specifically, we found that the difference between the effective weighting of the two systems ($\beta_{stim}$ and $\beta_{action}$) in amygdala-lesioned monkeys was not significantly different from that of controls (Fig. 4b, c inset). This indicates that amygdala lesions reduced sensitivity to stimulus-based and action-based signals to a similar degree, unlike the pattern observed in VS-lesioned monkeys. Instead, the main deficits due to amygdala lesions were captured by a biased initial state of arbitration that favors action-based signals. When coupled with the overall

reduced sensitivity to value signals, this effect diminishes differential effective arbitration rates for correct and incorrect (more reliable and less reliable) models (Fig. 4e). This suggests that in addition to its contribution to stimulus-based learning, amygdala also plays a crucial role in identifying and/or retaining the more reliable model of the environment[16], or mediating the influence of such identification on arbitration processes. This suggests that the amygdala, like the prefrontal cortex, is involved in learning to learn[30] and can explain why amygdala lesions weaken the amount of evidence needed before the animals reverse their choice preference[16].

Interestingly, VS-lesioned monkeys (with intact amygdala) were able to gradually overcome their impaired stimulus-based learning while showing a significantly larger arbitration rate for the stimulus-based system during the What-only task (Fig. 3e). This suggests that a signal to or from the amygdala, but not in the amygdala-to-VS pathway, could bias arbitration toward the more reliable model and lead to slow long-term behavioral adjustments. We found that arbitration was still present in amygdala-lesioned monkeys, suggesting that the amygdala is not required for arbitration per se but has a more nuanced role by setting and/or retaining the initial balance between models and improving the overall sensitivity to value signals. These two effects result in larger arbitration rates for the more reliable model of the environment, thus altering the trajectory of learning and choice behavior.

Moreover, we found that while VS-lesioned monkeys exhibit a bias toward the action-based strategy during the What/Where task, their response times are also shorter than those of control and amygdala-lesioned monkeys. Given the significant involvement of VS in effort exertion[31–34], the shorter RT in VS-lesioned monkeys could be linked to the fact that the stimulus-based strategy requires more cognitive effort. This is reflected in the slightly longer RT in What blocks compared to Where blocks, as well as the positive correlation between RT and arbitration weight. As a result, VS-lesioned monkeys may default to the action-based strategy, allowing them to perform the task faster. The stronger reliance on the action-based strategy in VS-lesioned monkeys can be adequately explained by the reduction in the $\rho$ parameter of our model, without necessarily suggesting an impaired arbitration process.

Arbitration between alternative models has garnered significant interest in cognitive, behavioral, and systems neuroscience. This includes arbitration between model-free vs. model-based RL[35–38], Pavlovian vs. instrumental control[39], habitual vs. goal-directed system[40,41], competing sets of strategies for solving complex stimulus-response mappings[42,43], and during social decision-making[44–46]. Here, we explored more basic arbitration required for any type of decision-making, as any choice option has to be selected by taking an action. Unlike arbitration between different types of learning systems--which often requires distinct reliability signals (e.g., model-free vs. model-based relying on unsigned reward prediction error and unsigned state prediction error[36])--we found that the same reliability signal, based on the value of the chosen option or chosen action ($V_{cho}$), can be used for arbitration between stimulus- and action-based learning. Critically, we found that in both controls and lesioned monkeys, the reliability signal based on $V_{cho}$ captured arbitration better than the reliability signal based on unsigned RPE. Because the difference in chosen values is equal to the difference in signed RPEs (Eq. 14), our results suggest that the reliability of alternative models may be more linked to signed RPE than to unsigned RPE.

Our proposal that the amygdala contributes to model arbitration to identify and reinforce the correct model of the environment is consistent with its postulated role in signaling attentional shifts for relevant control of behavior[47,48]. There are several pathways through which the amygdala could affect model arbitration. One major candidate is prefrontal-amygdala circuits[49,50]. In particular, the orbitofrontal cortex (OFC) receives substantial projections from the amygdala[51,52] and could serve a central role in encoding and monitoring the reliability of multiple actor predictive models[3]. Given that amygdala-to-OFC input has been reported to be significantly involved in value

coding by OFC neurons[53,54], it is possible that this input also carries information for selective arbitration to appropriately bias behavior toward the relevant learning system in a given environment. Conversely, the PFC-to-amygdala pathway could signal an internal state variable (arbitration weight in our model), serving as the necessary input to the amygdala for computing a differential adjustment in model arbitration that is relayed back to the PFC. This may explain why basolateral amygdala lesions could reduce OFC-induced impairment in reversal learning[14]. Thus, strong reciprocal connections between amygdala and PFC, in particular vlPFC, OFC, or ACC, could be crucial for proper arbitration between alternative models of the environment.

While earlier lesion studies have attributed varying degrees of behavioral deficits to amygdala[55,56], its role in instrumental learning has since been a matter of debate due to mixed evidence both in favor of[17–19,57,58] and against[14,15,59–61] its involvement in reward learning. Critically, our framework can account for the amygdala's seemingly inconsistent role. Through simulations of our best-fitting model with dynamic interaction between two systems, we found that in certain situations, the lower initial arbitration weight that biases behavior toward the action-based system can actually facilitate adjustments to reversals during stimulus-based learning, especially if the performance has saturated before reversal (Fig. 5b). This happens because more reliance on the less reliable action-based system allows faster exploration of alternative stimuli and thus faster reversal. This could account for puzzling improvement in performance due to basolateral amygdala lesions in rats[15] or monkeys[16,59], which has been attributed to increased benefits from negative feedback. In these examples, the stimulus-based system would still prefer the previously better option, which is no longer rewarding, but a less reliable action-based system would cause more switching from that option. Other studies that reported null results from reversible amygdala inactivations[60,61] have also utilized object reversals after initial learning of stimulus-reward associations over a long period of time (referred to as discrimination learning).

What these studies have in common is that they all utilize visual discrimination learning over a long period of time before a reversal, which would suppress learning from unrewarded trials (i.e., small $\alpha_-$) and allow the reliability of the stimulus-based system to reach its asymptote, thus slowing down reversal. This is very different from our experimental paradigm in which reversals happened on a short time-scale before the reliability of the stimulus-based system could stabilize. Overall, our study suggests that observed discrepancies in the effects of amygdala lesions are due to a dynamic interaction between arbitration, learning, and decision-making processes.

The dynamic interaction between arbitration and learning processes is particularly relevant in interpreting results using behavioral paradigms that were intended to parse the contributions of one learning system, but where competing learning systems could have strong unintended effects on behavior. Our results indicate that interpreting behaviors shaped by various learning systems should be approached with caution. This is particularly important when the manipulations in use might influence the arbitration process and thereby change the interplay among the learning systems. In principle, a multitude of simple learning strategies could underlie the heterogeneity in the so-called decision variables[62–64], and careful examination of neural signals[65–67] is needed to properly identify the neural substrates of corresponding learning systems.

## Methods

### Experimental paradigm
We examined two variants of a probabilistic reversal learning task in which monkeys selected between two visual stimuli to obtain a juice reward. During each block of the What-only task, reward was assigned stochastically according to stimulus identity only while reward probabilities on the two stimuli (selected afresh on each block) switched between trial 30 and 50 of the block without any signal to the monkeys

(Fig. 1a). In the What/Where task, reward was assigned based on either stimulus identity (What blocks) or stimulus location (Where blocks) with reversal similar to the What-only task (more details below). Data were collected from a total of 20 unique monkeys, some of whom received bilateral excitotoxic lesions to either the amygdala or the VS (Supplementary Fig. 17). All experimental procedures for all monkeys were performed in accordance with the *Guide for the Care and Use of Laboratory Animals* and were approved by the National Institute of Mental Health Animal Care and Use Committee. We describe each experimental setup in more detail below.

### What-only task
Data from this task[17] were collected in eleven male rhesus macaques weighing 6.5–10.5 kg (controls: $n = 4$; amygdala-lesioned: $n = 4$; VS-lesioned: $n = 3$). The monkeys completed an average of 26.73 sessions (SD = 5.98) and an average of 16.81 (SD = 6.72) blocks per session. In total, monkeys completed on average 372.9 blocks (SD = 89.6). Each block consisted of 80 trials and involved a single reversal of stimulus-outcome contingencies on a randomly selected trial between trials 30 and 50 from a uniform distribution (Fig. 1a). On each trial, monkeys were trained to fixate on a central point on a screen (500–750 ms) to initiate the trial. After fixation, two stimuli, a square and a circle of random colors, were assigned pseudo-randomly to the left and right of the fixation point (6° visual angle). Monkeys indicated their choice by making a saccade to the target stimulus and fixating for 500 ms. Reward (0.085 ml juice) was delivered according to the assigned reward schedule for a given block. Each trial was followed by a fixed 1.5 s inter-trial interval. Trials in which monkeys failed to fixate within 5 s or make a choice within 1 s were aborted and then repeated.

The reward schedule was determined by the probabilities of reward on two choice options selected from four possible values: 100/0, 80/20, 70/30, and 60/40. The reward schedule was randomly selected at the start of each block and remained constant within that block. Monkeys performed the deterministic task (100/0 reward schedule) after the data collection for the stochastic task had been completed. Here, we focus on our analyses of the task's stochastic variant to match the reward schedules used in the What/Where task (which only contained stochastic schedules, as described below), and therefore, we have excluded the deterministic portion of the data from our analyses. All monkeys that received lesions were trained and tested following their recovery from surgery. For more detailed surgical information, see the Supplemental Experimental Procedures in the original study[17]. This experimental setup and some analyses of the data have been previously reported[17].

### What/Where task
Unlike the What-only task, the What/Where task involved both stimulus-based and action-based learning, and this feature introduced additional uncertainty about the correct model of the environment. The effects of lesions to the VS and amygdala during the What/Where task were examined in two different studies with separate sets of controls for each. The first study investigating the effect of VS lesions[24] had a total of eight subjects weighing 6.5–11 kg (controls: $n = 5$; VS-lesioned: $n = 3$). One of the five control monkeys and all three of the VS-lesioned monkeys were the same monkeys used in the What-only task[17]. The second study investigating the effect of amygdala lesions[18] had a total of 10 subjects weighing 6–11 kg (controls: $n = 6$; amygdala-lesioned: $n = 4$). One of the six unoperated controls was the same monkey used in the What-only task[17] and the What/Where task involving VS lesion[24]. One additional control monkey was used as an unoperated control for the earlier What/Where task only[24]. The remaining control and the four amygdala-lesioned monkeys were additionally trained for the subsequent study using the What/Where task[18]. See Supplementary Fig. 17a for a summary diagram. Any monkeys that have participated in both the What-only and What/Where

tasks (i.e., one control and three VS-lesioned monkeys) first completed the What-only task and then later completed the What/Where task. Notably, all newly trained monkeys that performed the What/Where task were first trained on a simple two-armed bandit stimulus-based reward associations ("What" condition). After learning about this task, they were then trained with a deterministic version (100/0) of the What/Where task and gradually transitioned into the probabilistic outcomes used for this experiment (80/20, 70/30, 60/40). As such, all monkeys in the study shared the same prior training experience, specifically learning the "What" task first.

The monkeys in this task completed an average of 29.56 sessions (SD = 5.36), with an average of 18.41 (SD = 4.26) blocks per session. In total, monkeys completed on average 559.1 blocks (SD = 141.6). Each block consisted of 80 trials and involved a single reversal of the stimulus-based or action-based contingencies between trials 30 and 50. A given block was randomly assigned as a What or a Where block and remained constant within that block (Fig. 1b). In What blocks, reward probabilities were assigned based on stimulus identity, with a particular object having a higher reward probability. In Where blocks, reward probabilities were assigned based on location, with a particular side having a higher reward probability regardless of stimulus identity. What and Where blocks were randomly interleaved throughout the session, and the block type was not indicated to the monkey. The reward schedule was randomly selected from three schedules (80/20, 70/30, 60/40) at the start of each block and remained constant within that block.

On each trial, monkeys were trained to fixate on a central point on a screen (400–600 ms) to initiate the trial. After fixation, two visual objects were assigned pseudo-randomly to the left and right of the fixation point (6° visual angle). Each block used two novel images that the animal had never seen before. Monkeys indicated their choice by making a saccade to the target stimulus and fixating for 500 ms. Reward was delivered probabilistically according to the assigned reward schedule for a given block. Each trial was followed by a fixed 1.5 s inter-trial interval. Trials in which monkeys failed to fixate within 5 s or make a choice within 1 s were aborted and then repeated. This experimental setup, surgical information, and some analyses of the data have been previously reported[18,24].

## Quantification and statistical analysis

**Entropy-based metrics.** Here, we utilized information-theoretic metrics to quantify learning and choice behavior[22,23]. Specifically, we focused on the conditional entropy of reward-dependent strategy (ERDS) to measure how monkeys associated reward feedback with stimulus identity or action.

Generally, ERDS is calculated as follows:

$$
\begin{aligned}
ERDS = H(str|rew) = - \Big\{ &P(stay,win) \times \log_2 \frac{P(stay,win)}{P(win)} + P(switch,win) \\
&\times \log_2 \frac{P(switch,win)}{P(win)} + P(stay,lose) \times \log_2 \frac{P(stay,lose)}{P(lose)} \\
&+ P(switch,lose) \times \log_2 \frac{P(switch,lose)}{P(lose)} \Big\}
\end{aligned}
\tag{1}
$$

where str is the adopted strategy coded as stay (1) or switch (0), rew is the previous reward outcome coded as reward (1) or no reward (0). Noting that $\frac{P(stay,win)}{P(win)}$ and $\frac{P(switch,lose)}{P(lose)}$ measure tendencies for win-stay and lose-switch strategies, one can see the ERDS combines these tendencies into a single quantity:

$$
\begin{aligned}
ERDS = -\{ &P(stay,win) \times \log_2(WinStay) + P(switch,win) \\
&\times \log_2(1 - WinStay) + P(stay,lose) \times \log_2(1 - LoseSwitch) \\
&+ P(switch,lose) \times \log_2(LoseSwitch))\}.
\end{aligned}
\tag{2}
$$

As the equations above suggest, ERDS measures the consistency in response to reward feedback (win or lose). Lower ERDS values correspond to decreased randomness in the variable and thus more consistency in the utilized strategy, which could be stimulus-based and/or action-based. To detect these strategies, we defined two types of ERDS by considering choice and reward feedback in terms of stimulus identity or action, corresponding to ERDS$_{Stim}$ and ERDS$_{Action}$, respectively.

Therefore, lower values of ERDS$_{Stim}$ suggest that the animals stay or switch consistently based on stimulus identity according to reward feedback, indicating the stronger adoption of the stimulus-based strategy. Conversely, lower values of ERDS$_{Action}$ indicate that the animals adopted the action-based strategy more strongly. Overall, comparison of ERDS$_{Stim}$ and ERDS$_{Action}$ enables us to quantify the adopted strategy on a trial-by-trial basis, either by computing the average values across a block of trials or by aligning all trials relative to the beginning, reversal point, and the end of each block.

## Computational models

**Single-system reinforcement learning (RL) models.** We first used two standard RL models that learn about one type of reward contingencies to fit monkeys' choice data. Specifically, the RL$_{Stim-only}$ and RL$_{Action-only}$ models associate reward outcomes to choice options either in terms of stimulus identity or chosen action in order to estimate stimulus and action values, respectively. These values were used to determine choice on each trial and were updated based on reward outcome at the end of trial, as described below.

More specifically, the value of the chosen option ($V_C$) is updated using reward prediction error (RPE) and two separate learning rates for rewarded and unrewarded trials ($\alpha_+$ and $\alpha_-$, respectively) while the value of the unchosen option ($V_U$) decays to zero:

$$
V_C(t+1) = \begin{cases} V_C(t) + \alpha_+ \big( R(t) - V_C(t) \big) \text{ if } R(t) = 1, \\ V_C(t) + \alpha_- \big( R(t) - V_C(t) \big) \text{ if } R(t) = 0, \end{cases}
\tag{3}
$$

$$
V_U(t+1) = (1 - \zeta) V_U(t),
\tag{4}
$$

where $R(t)$ is the reward feedback on trial $t$ and $\zeta$ is the decay or forgetting rate for the unchosen option. Chosen and unchosen options are coded as {stimulus A, stimulus B} in the RL$_{Stim-only}$ model and as {Left, Right} in the RL$_{Action-only}$ model. In these and other models, the probability of choosing the option on the right, $P_{Right}(t)$, was computed using a softmax function:

$$
P_{Right}(t) = \frac{1}{1 + \exp(-\beta_1 \big( OV_{Right}(t) - OV_{Left}(t) \big) - \beta_0)},
\tag{5}
$$

where $OV_{Left}$ and $OV_{Right}$ denote the overall reward value of options on the left and right, $\beta_1$ controls the steepness of the sigmoid function (inverse temperature) measuring the baseline sensitivity of choice to difference in value signals, and $\beta_0$ is the side bias with positive values corresponding to a bias toward choosing right. For the RL$_{Stim-only}$ model, $OV_{Left}$ and $OV_{Right}$ were assigned based on the stimulus identity appearing on the respective side for a given trial. For example, if stimulus A appeared on the left of fixation, then $OV_{Left} = V_{StimA}$ and $OV_{Right} = V_{StimB}$. For the RL$_{Action-only}$ model, the overall reward values correspond to action values; i.e., $OV_{Left} = V_{Left}$ and $OV_{Right} = V_{Right}$.

**Two-system model with static weighting of stimulus- and action-based learning.** As an extension of the above RL models, we considered hybrid RL models that constituted two value functions, $V_{Stim}$ and $V_{Action}$, to simultaneously track the reward value for alternative stimuli and actions and made choices based on a weighted sum of value

signals from the two systems with a weight that was fixed on each block of the experiment (RL$_{\text{Stim+Action}}$+Static $\omega$ or Static $\omega$ model for short). Specifically, the value functions were updated in parallel using Eqs. 3 and 4.

Therefore, the overall values in this model are computed as follows:

$$OV_i = V_{Stim(i)}\omega + V_{Action(i)}(1-\omega), \qquad (6)$$

where $\omega$ represents the relative weight of the stimulus-based system compared to the action-based system, $i \in \{\text{Left, Right}\}$, and $V_{\text{Stim}(i)}$ indicates the stimulus value for the option appearing on the side $i$. For example, if the stimulus A appeared on the left side, then $OV_{\text{Left}} = V_{\text{StimA}}\omega + V_{\text{Left}}(1-\omega)$ and $OV_{\text{Right}} = V_{\text{StimB}}\omega + V_{\text{Right}}(1-\omega)$. Similar to the learning rates and other parameters, a single value of $\omega$ was estimated for each block of trials. In the special case where $\omega = 0.5$, the stimulus-value and action-value exert equal influence on choice. We note that the overall reward value in our model is used mainly as a convenience for presenting the model and does not require stimulus and action values to be integrated. Rather, each system can first compare its own values (stimulus values against other stimulus values, and action values against other action values), and the results of these within-system comparisons are then combined, with different weights, to determine the choice (see Eq. 7).

Using the above OVs (Eq. 6), the decision rule in Eq. 5 can be rewritten as follows (omitting the trial index t for simplicity):

$$
\begin{aligned}
\text{logit}\left(P_{Right}\right) &= \beta_0 + \beta_1\left(OV_{Right} - OV_{Left}\right) \\
&= \beta_0 + \beta_1\Big\{ V_{Stim(Right)}\omega + V_{Right}(1-\omega) \\
&\quad - \left(V_{Stim(Left)}\omega + V_{Left}(1-\omega)\right)\Big\} \\
&= \beta_0 + \beta_1\Big\{ \left(V_{Stim(Right)} - V_{Stim(Left)}\right)\omega \\
&\quad + \left(V_{Right} - V_{Left}\right)(1-\omega)\Big\} \\
&= \beta_0 + \beta_1\big\{\Delta V_{Stim}\omega + \Delta V_{Action}(1-\omega)\big\} \\
&= \beta_0 + \beta_1\omega(\Delta V_{Stim}) + \beta_1(1-\omega)(\Delta V_{Action}),
\end{aligned}
\qquad (7)
$$

where $\Delta V_{\text{Stim}} = V_{\text{Stim(Right)}} - V_{\text{Stim(Left)}}$ and $\Delta V_{\text{Action}} = V_{\text{Right}} - V_{\text{Left}}$. That is, $\beta_1\omega$ and $\beta_1(1-\omega)$ represent the sensitivity of choice to value signals from the stimulus- and action-based systems, respectively. Therefore, $\omega$ controls the relative sensitivity of choice to the two competing systems, with stronger $\omega$ corresponding to a stronger influence of the stimulus-based system, and vice versa.

**Two-system models with dynamic weighting of stimulus- and action-based learning.** To allow for dynamic arbitration, we constructed hybrid models in which $\omega$ was updated on a trial-by-trial basis using the relative reliability of the two systems. In this model (Dynamic $\omega$), the difference in reliability of the two systems is computed at the end of each trial to update the value of $\omega$ toward the more reliable system. More specifically, the relative reliability, $\Delta$Rel, between two systems at the end of trial $t$ is computed as follows:

$$\Delta Rel(t) = \Delta V_{cho}(t) = V_{C,Stim}(t) - V_{C,Action}(t), \qquad (8)$$

where $V_{\text{C,Stim}}$ and $V_{\text{C,Action}}$ correspond to the value of the chosen option in the stimulus- and action-based system, respectively, and $\Delta V_{\text{cho}}$ denotes the (signed) difference between the two. $\Delta$Rel ranges $[-1, 1]$, with positive values indicating a more reliable stimulus-based system. Intuitively, $\Delta V_{\text{cho}}$ signals the system that gives an overall larger value for the given choice and thus, is more reliable in predicting reward.

Subsequently, the relative sensitivity of choice to the two systems, $\omega$, is updated as follows:

$$
\omega(1) = \omega_0,
$$
$$
\omega(t+1) = \begin{cases} \omega(t) + \alpha_\omega \Delta Rel(t)(1-\omega(t)) + \zeta_\omega(\omega_0 - \omega(t)) \text{ if } \Delta Rel(t) > 0, \\ \omega(t) + \alpha_\omega |\Delta Rel(t)|(0-\omega(t)) + \zeta_\omega(\omega_0 - \omega(t)) \text{ if } \Delta Rel(t) < 0, \end{cases}
$$
$$(9)$$

where $\omega_0$ is the initial $\omega$ on the first trial (onset of each block), $\alpha_\omega$ is the baseline model arbitration rate (distinct from the learning rates in in Eq. 3), and $\zeta_\omega$ is the passive decay rate that pulls $\omega$ toward its initial value $\omega_0$ (distinct from the decay rate in Eq. 4). This additional decay mechanism assumes that $\omega$ defaults back to its initial bias in the absence of exogenous input signaling the reliability difference ($\Delta$Rel). We focus on this model with passive decay for all analyses, as it fits better than the variants without the passive decay term (Supplementary Fig. 18a). Importantly, the arbitration rates $\alpha_\omega$ tend to be larger than the decay rates $\zeta_\omega$ across groups and tasks (Supplementary Fig. 18b, c).

**Two-system models with dynamic weighting and separate baseline signals for stimulus- and action-based learning.** To rule out the possibility that the observed effects in amygdala-lesioned monkeys are solely due to impairment in learning stimulus values (e.g., by reducing the baseline strength of stimulus-value signals) and without any changes to arbitration processes, we included an additional parameter to separate these two types of changes. In the Dynamic $\omega$ model, an increase in the sensitivity to the stimulus-based system, $\beta_1\omega(t)$, is strictly tied to a decrease in the sensitivity to the action-based system, $\beta_1(1-\omega(t))$, and vice versa. This constraint can be removed by introducing a constant factor that further scales the value of a given model before combining with the value from the other model. In this new model referred to as Dynamic $\omega$-$\rho$, the overall values are equal to:

$$OV_i = V_{Stim(i)}\rho\omega(t) + V_{Action(i)}(1-\rho)(1-\omega(t)), \qquad (10)$$

where $\rho$ is a constant that measures the baseline ratio of signal strength from the stimulus-based system relative to the action-based system (estimated for each monkey), independent of the time-dependent arbitration weight ($\omega$). The update for $\omega$ is the same as in the Dynamic $\omega$ model (Eq. 9). Therefore, similar to Eq. 7, the decision rule for the Dynamic $\omega$-$\rho$ model can be simplified as follows:

$$
\begin{aligned}
\text{logit}\left(P_{Right}(t)\right) &= \beta_0 + \beta_1\big\{OV_{Right}(t) - OV_{Left}(t)\big\} \\
&= \beta_0 + \beta_1\rho\omega(t)\Delta V_{Stim}(t) + \beta_1(1-\rho)(1-\omega(t))\Delta V_{Action}(t).
\end{aligned}
$$
$$(11)$$

This shows that $\beta_1\rho$ and $\beta_1(1-\rho)$ can be interpreted as the baseline strength of the stimulus- and action-based signals on choice, respectively, prior to modulation by the time-dependent arbitration parameter $\omega$. Therefore, including $\rho$ allows us to capture baseline (time-independent) differences in the strength of signals from the two learning systems. Therefore, $\rho < 0.5$ ($\rho > 0.5$) corresponds to lower baseline activity or impairment in the stimulus-based (respectively, action-based) system. When $\rho = 0.5$, the Dynamic $\omega$-$\rho$ model reduces to the Dynamic $\omega$ model. Because $\rho$ is assumed to capture the baseline activity, we estimated a single value of $\rho$ per each monkey for the entire duration of the experiment (see *Model fitting and simulation* for more details).

Because in this model, the arbitration weight is further weighted by $\rho$ (or $1-\rho$), we defined an "effective" arbitration weight to measure the overall relative weighting between the stimulus-based ($\beta_{\text{Stim}} = \beta_1\rho\omega$) and action-based ($\beta_{\text{Action}} = \beta_1(1-\rho)(1-\omega)$) systems as

follows:

$$\Omega(t) = \frac{\beta_{Stim}(t)}{\beta_{Stim}(t) + \beta_{Action}(t)} = \frac{\rho\omega(t)}{\rho\omega(t) + (1-\rho)(1-\omega(t))}, \quad (12)$$

where $\Omega$ is the relative weight of the stimulus-based system with respect to the total weight of the two systems on choice. It is worth noting that the common baseline inverse temperature $\beta_1$ is independent of $\Omega$, which is determined only by $\rho$ and $\omega$, and that $\Omega$ reduces to $\omega$ when $\rho = 0.5$.

**Alternative signals for estimating the reliability of learning systems.** In addition to $V_{cho}$ as the reliability signal for updating $\omega$ in the dynamic models, we also considered several other quantities to estimate reliability. As the first alternative to $\Delta V_{cho}$ for the relative reliability used to update the relative weight (Eq. 8), we considered the difference in magnitudes of RPE (|RPE|) of the action- and stimulus-based systems:

$$\Delta Rel(t) = |RPE_{Action}(t)| - |RPE_{Stim}(t)|. \quad (13)$$

Conceptually, a system that yields better prediction of reward on a given trial has a lower magnitude of RPE and thus is more reliable. Note that the difference in chosen stimulus and action values, $\Delta V_{cho}$, can be also rewritten as:

$$\begin{aligned} \Delta V_{cho}(t) &= V_{C,Stim}(t) - V_{C,Action}(t) \\ &= \big(R(t) - V_{C,Action}(t)\big) - \big(R(t) - V_{C,Stim}(t)\big) \\ &= RPE_{Action}(t) - RPE_{Stim}(t). \end{aligned} \quad (14)$$

This demonstrates that using $\Delta V_{cho}$ for the relative reliability corresponds to the signed RPE instead of the unsigned RPE.

We also considered the difference between the value of chosen and unchosen options within each system (|$\Delta V$|) as a measure of the reliability of that system. Intuitively, this reliability signal is larger for a system that yields a better discernibility between the two competing options on a given trial. In this case, the relative reliability can be written as:

$$\begin{aligned} \Delta Rel(t) &= \big|\Delta V_{Stim}(t)\big| - \big|\Delta V_{Action}(t)\big| \\ &= \big|V_{StimA}(t) - V_{StimB}(t)\big| - \big|V_{Left}(t) - V_{Right}(t)\big|. \end{aligned} \quad (15)$$

Finally, we also considered the total sum of the two value estimates (|$\Sigma V$|) as a signal for estimating the reliability of a given system. For this measure, reliability is larger for a system that gives overall larger combined values from the two options. In this case, the relative reliability is equal to:

$$\begin{aligned} \Delta Rel(t) &= \Sigma V_{Stim}(t) - \Sigma V_{Action}(t) = \big(V_{StimA}(t) + V_{StimB}(t)\big) \\ &\quad - \big(V_{Left}(t) + V_{Right}(t)\big). \end{aligned} \quad (16)$$

For all these different versions of $\Delta Rel$, we used the same equation for updating $\omega$ (Eq. 9).

**Effective arbitration rates.** In the above formulation of arbitration mechanism (Eqs. 8–12), the rate of update for $\omega$ depends on several factors including the baseline ratio parameter $\rho$, baseline model arbitration rate $\alpha_\omega$, the trial-by-trial difference in reliability ($\Delta Rel$), and the passive decay mechanism ($\zeta_\omega$). To capture the overall update rates in arbitration weight $\Omega$, we defined the "effective" arbitration rates that quantify the overall rate of change toward either the stimulus- or action-based system (Fig. 3d). Analogous to the update rule in the valuation system (Eq. 3), an update rule for the effective arbitration weight can be

written as follows:

$$\Omega(t+1) = \begin{cases} \Omega(t) + \psi_+(t)(1-\Omega(t)) & \text{if } \Delta\Omega(t) > 0, \\ \Omega(t) + \psi_-(t)(0-\Omega(t)) & \text{if } \Delta\Omega(t) < 0. \end{cases} \quad (17)$$

where $\psi_+$ and $\psi_-$ represent the effective arbitration rate toward the stimulus-based and action-based systems, respectively ($\Delta\Omega = 0$ corresponds to no change in arbitration). Unlike the learning rates ($\alpha_+$ and $\alpha_-$) that are fitted to each session, the effective arbitration rates ($\psi$) are estimated for each trial (Fig. 3e–g). More specifically, the effective arbitration rate on trial $t$ when $\Omega$ increases, shifting choice toward the stimulus-based system, is defined as follows:

$$\psi_+(t) = \frac{[\Delta\Omega(t)]^+}{1-\Omega(t)}, \quad (18)$$

whereas the effective arbitration rate when $\Omega$ decreases, biasing choice toward the action-based system, is defined as follows:

$$\psi_-(t) = \frac{[\Delta\Omega(t)]^-}{0-\Omega(t)}, \quad (19)$$

where []$^+$ and []$^-$ indicate positive and negative changes in $\Omega$, respectively. Similar to the learning rates for rewarded and unrewarded trials ($\alpha_+$ and $\alpha_-$), which capture differential updates based on different reward outcomes, the effective arbitration rates capture differential update rates toward the correct and incorrect learning systems based on the difference in reliability. Specifically, we tested whether the effective arbitration rate of the correct system in a given block type (i.e., $\psi_+$ in What blocks and $\psi_-$ in Where blocks) is distinct from that of the incorrect system, as larger effective rates for the correct system would reduce noise in incorporating feedback, enabling more efficient arbitration and, ultimately, leading to improved performance.

**Model fitting and simulation**
We used the standard maximum likelihood estimation method to fit choice data and estimate the best-fit parameters for the described models. One set of model parameters was fitted to each session (consisting of ~20 blocks) of monkeys' choice data. Fitting was performed using the MATLAB optimization function *fmincon*, repeating the search for 100 sets of random initial parameter values to ensure global minima. See Supplementary Table 15 for the list of models and the ranges of parameters used. We report the mean Akaike Information Criterion (AIC) values of each model in Supplementary Fig. 14.

For the Dynamic $\omega$-$\rho$ model (Eqs. 10–12), we assumed that the value of $\rho$ is fixed for the entirety of the experiment and does not vary across sessions, as it aims to capture the relative strength of one learning system relative to the other. Accordingly, to estimate a single value of $\rho$ for each monkey, we fitted the entire dataset of a given monkey and obtained a single set of best-fit parameters. From this set, we only kept the value of $\rho$ and fit the choice data again for the remaining parameters by allowing different values across sessions. The distributions of the other fitted parameters of this model are reported in Supplementary Figs. 12–13.

To test whether the observed relationship between stimulus- and action-based learning indeed required competition between the two learning systems and was not due to task structure, we simulated choice behavior using single-system or two-system models and computed model-simulated ERDS$_{Stim}$ and ERDS$_{Action}$ (Eq. 1) (Supplementary Fig. 9). To measure the association between the two measures and isolate the within-subject effect, we mean-centered the predictor (ERDS$_{Stim}$) and fitted the mixed-effects model as ERDS$_{Action}$ ~ ERDS$_{Stim}$ + (1|subject) to account for subject variability. To that end, we used the fitted parameters for each session and simulated

the choice behavior 100 times for each block, using the same random reversal position and reward schedule as the behavioral data. We obtained the final averaged ERDS values for stimulus and action.

## K-fold cross-validation of model performance

To compare the goodness-of-fit and determine the winning model, we used five-fold cross-validation, where each set of training/testing blocks was tested repeatedly with 50 unique instances. For each task, we created the training and testing sets as follows: for each subject, we randomly partitioned all the block data experienced by the monkey into five equal subsamples or "folds," and selected each fold as the test data (20%) while using the remaining four folds as the training data (80%). For each instance, we obtained best-fit parameters from the training set that minimized the negative log-likelihood across all training blocks and used these to calculate the negative log-likelihood from the test blocks. Model performance was tested for each fold, thereby exhaustively testing all blocks data for a given subject. We repeated this procedure 50 times for each subject, each time using a unique combination of partitioning the data into five folds. Final mean negative log-likelihoods (-LL) were obtained by averaging across all tested blocks, reflecting the total sum of -LL during each block on average. We report these values in Figs. 2b and 3a, Supplementary Fig. 7a, and Supplementary Fig. 18a. Model performance for individual monkeys is also reported in Supplementary Table 16.

In these results, we note that even a small improvement in -LL implies a significant improvement in the predictability of the model for the tested blocks (Supplementary Note 3). Along with the -LL results, we also provide McFadden's $R^2$ values[68] as an absolute measure for goodness-of-fit for each model in comparison to a null model with chance-level prediction. This quantity is calculated as follows:

$$McFadden\,R^2 = 1 - \frac{\sum_t LL_{model}(t)}{\sum_t LL_{null}} = 1 - \frac{\sum_t LL_{model}(t)}{80 \times ln(0.5)}, \quad (20)$$

where $t$ indicates trial number within each block of 80 trials.

## Models and parameters recovery

To perform model recovery (Supplementary Fig. 8a–d), we simulated the choice behavior of each model during a randomly created block environment similar to the experiment. Specifically, each session consisted of 30 blocks, each block having 80 trials with a reversal and with a randomly assigned reward schedule (80/20, 70/30, 60/40) and block type (What-only or What/Where). To ensure that choice behavior is simulated within a plausible range of parameters, we randomly sampled each parameter value from a kernel distribution fitted to all observed parameter values for a given model. We then fit all models and determined the best-fit model for each simulated session based on the AIC. We repeated this procedure 1000 times and report the proportion of sessions that a given fitted model best accounts for each simulated model.

For parameter recovery (Supplementary Fig. 8e, f), we simulated the choice behavior of the best model (Dynamic $\omega$-$\rho$) using the estimated parameters from the experimental data and then refit the simulated data with the same model. Each block environment was set up using the same reward schedule and reversal position as the actual experiment. We simulated each block one time to ensure that the total number of simulated trials is the same as that of the experiment, from which the true parameters were estimated. We recovered the parameters from the simulated data using the same fitting procedure based on AIC, repeating the search for 100 initial random parameter values to ensure global minima. For recovering the $\rho$ parameter, we used the entire simulated data for each monkey and obtained a single set of best-fit parameters, as was done for the actual data.

## Data analysis and statistical tests

All analyses were carried out using MATLAB (MathWorks, version 2021b). All comparisons were performed using appropriate statistical tests reported throughout the text. For each test, we report the exact $p$-values and effect sizes when appropriate. Unless otherwise noted, all statistical tests used in this study were two-sided. In cases where highly significant results caused the software to return $p$-values of zero due to numerical precision limits, we report the smallest representable floating-point number at machine precision (i.e., $4.94 \times 10^{-324}$).

Because the data were collected across different animals, we primarily utilized linear mixed-effects regression analyses (using MATLAB *fitlme* function) for between-group comparisons, with the group assignment as a fixed effect and subjects as a random effect, to appropriately account for between-subject variance. To maximize the accuracy of models and reduce Type-I error, we considered subject-level intercepts and additional random slopes for long-term adjustment effects, both across the experiment (*proportion of session completed*) and within each session day (*block within a session*). When performing a within-group significance test to compare a paired set of samples (e.g., $\Delta\beta = \beta_{stim} - \beta_{action}$, $\Delta\psi = \psi_+ - \psi_-$), we fitted mixed-effects models with a fixed intercept and subject-level random effects (*data* ~ 1 + (1 + *sess_perc* + *block_in_sess*|*subject*)), where the main intercept represents the mean value of the paired difference. We then tested whether this intercept significantly differed from zero, as indicated by the coefficient $\beta_0$. For comparing block-wise performance within each group, we included random effects of subjects and fixed effects of block types (*What-only, What, Where*) and reward uncertainty (measured as the variance of the outcome[13]) with the following formula: *P(Better)* ~ *variance* + *BlockType* + (1 + *variance+sess_perc+block_in_sess*|*subject*). Specifically, the variance was calculated as $p_B*(1-p_B)$, where $p_B$ represents the reward probability of the better option (i.e., $p_B = \{0.8, 0.7, 0.6\}$). Variance was further mean-centered by subject for better interpretability of other coefficients.

To categorize each trial as either *stimulus-* or *action-dominant*, we directly compared ERDS$_{Stim}$ and ERDS$_{Action}$ (computed from a moving window of 10 trials; Eq. 1). Trials with ERDS$_{Stim}$ < ERDS$_{Action}$ and ERDS$_{Action}$ < ERDS$_{Stim}$ were categorized as stimulus-dominant and action-dominant, respectively. We dropped trials with ERDS$_{Stim}$ = ERDS$_{Action}$, amounting to 11.1% of total trials in the What-only and 11.7% in the What/Where tasks. We used a mixed-effects regression model with subjects as random intercepts to test whether the dominant strategy significantly modulates RT. Fixed effects included the following predictors: dominant strategy, coded as stim-dominant (0) or action-dominant (1), whether the monkey had chosen the better option (1) or not (0), reward schedule or uncertainty (measured as variance), trial number within a block, block number within a session, and session number within the subject. We also included interaction between the dominant strategy and the choice of a better option, as the latter could depend on the adopted strategy. Variables were normalized by each subject to yield comparable standardized regression coefficients.

To study the long-term adjustment in behavior across the time course of the experiment (Supplementary Fig. 11), we calculated ERDS$_{Stim}$, ERDS$_{Action}$ (Eq. 1), and median RT for each block and regressed them on the proportion of the sessions (total block number) completed as the predictor variable. This regressor was further mean-centered by subject for better interpretability of other coefficients. Initial effective arbitration weight ($\Omega_0$), which was estimated for each session, was analyzed at the session level. To further account for the variability in adjustment at the subject level, we considered random slopes and intercepts for the effect of sessions within each subject using mixed-effects models. We included all group data from the What-only task and used planned contrasts to infer slopes for each group and the group differences in the slopes. Full results are reported in Supplementary Tables 11–14. For visualization purposes only, we

plot the simple least-squares lines and the estimated slopes for each group ($\beta_{sess}$) in Supplementary Fig. 11.

To estimate the effective arbitration rate across time (Fig. 3e–g; Eqs. 18 and 19), we computed the mean trajectory for each of two transition rates (toward stimulus- or action-based system) across blocks by aligning all trials relative to the beginning, reversal point, and the end of each block. For the bar plots in the insets, we computed the difference between the mean of the two transition rates within each block. In particular, we focused on the trials after the reversal to avoid potential confounds with initial bias ($\omega_O$) and observed the transition behavior after adjusting to the initial uncertainty of the block.

To plot performance (Fig. 1e–g) and effective arbitration rates over time (Fig. 3e–g; Eqs. 18 and 19), we obtained the trajectories by concatenating 20 trials relative to the beginning, reversal point, and the end of each block, to account for the random reversal positions. These curves were then smoothed with a moving window of five trials, separately within the acquisition and reversal phases.

To identify the model parameters that are significantly modulated in the lesioned group with the consistent direction of effects across task conditions (*What-only* and *What/Where*), we adopted the following mixed-effects model: *parameter ~ group + (group|task) + (1|subject:task)*. Namely, the model included a fixed effect of group (*control, amygdala, VS*) and random effects of subjects and tasks with a random slope of each group in each task. This formulation assumes a single, uniform effect of the group regardless of task condition, thereby identifying parameters that are consistently modulated by brain lesions. Note that subjects were nested within tasks to allow for a separate baseline for each task, as some of the monkeys participated in both the What-only and What/Where tasks. We tested this mixed-effects model on each of the parameters compiled across groups and tasks, with control monkeys as the reference group. For testing the group difference in the $\rho$ parameter (Eqs. 10 and 11) across task conditions, which was fitted for each subject and therefore lacks adequate sample size, we utilized two-sided permutation tests. Specifically, we conducted permutation tests on the $\rho$ values of each subject from a given pair of tested groups across both tasks and generated a null distribution of the test statistic (i.e., mean group difference in $\rho$) by randomly shuffling the group assignment 10,000 times. The *p*-value was calculated as the proportion of permuted test statistics that were as extreme or more extreme than the empirically observed test statistic.

## Reporting summary

Further information on research design is available in the Nature Portfolio Reporting Summary linked to this article.

## Data availability

The data analyzed in this study are available at https://github.com/DartmouthCCNL/woo_etal_amygdala[69]. The data generated in this study are provided in the Source data file. Further information and requests for resources should be directed to and will be fulfilled by the Lead Contact, Dr. Alireza Soltani (alireza.soltani@dartmouth.edu). Source data are provided with this paper.

## Code availability

Custom analysis codes are available at https://github.com/DartmouthCCNL/woo_etal_amygdala[69].

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

## Acknowledgements

The authors thank Chanc VanWinkle Orzell and Dmitriy Lisitsyn for their helpful comments on the manuscript. This work is supported by the National Institutes of Health (R01 DA047870 to A.S.) and by the Intramural Research Program of the NIMH (ZIA MH002928). The contributions of the NIH authors were made as part of their official duties as NIH federal employees, are in compliance with agency policy requirements, and are considered Works of the United States Government. However, the findings and conclusions presented in this paper are those of the authors and do not necessarily reflect the views of the NIH or the U.S. Department of Health and Human Services.

## Author contributions

J.H.W. and A.S. designed the study; V.C., C.T., K.R., and B.A. designed the experiments; J.H.W., V.C., C.T., K.R., B.A., and A.S. performed research; J.H.W. and A.S. analyzed data; J.H.W. and A.S. wrote the first draft paper. All authors contributed to the revision of the paper.

## Competing interests

The authors declare no competing interests.
