## [Transparent Peer Review file · Nature Communications]

Contribution of amygdala to dynamic model arbitration under uncertainty

Corresponding Author: Dr Alireza Soltani

Version 0:

Reviewer comments:

Reviewer #1

(Remarks to the Author)

The authors presented a study in which they reanalyzed lesion and behavioral data from non-human primates from three previously published studies. They compared the ability of different reinforcement learning (RL) based computational models to predict the behavior of a control group, a group with bilateral lesions in the ventral striatum (VS), and a group with bilateral lesions in the amygdala. The behavioral tasks consisted of two types of two-armed bandit tasks. In the first (what-only task), subjects had to identify the stimulus associated with a higher probability of reward regardless of the side of presentation, while in the second, they were exposed to a non-stationary environment alternating between bandits where reward probability was linked to stimulus features and bandits where the location of the stimulus (regardless of its perceptual features) was reinforced (what/where task).

The authors demonstrated that the computational model that best captured the behavior of the subjects included a dynamic arbitration mechanism between stimulus-based and action-based learning (updated trial-by-trial) and different baselines for stimulus-based and action-based learning. All lesioned subjects showed a bias towards action-based learning (where), while only the subjects with amygdala lesions exhibited a deficit in the arbitration process, which appeared to converge toward optimality much more slowly than in the controls (comparison in Figs. 2d and 3b). This phenomenon also emerged from the analysis of the update rate in the arbitration process, which was deficient in the amygdala-lesioned subjects for both the “what” and “where” environments. Finally, through the analysis of the parameters optimized on the behavioral data, the authors showed that amygdala lesions resulted in a shift in the initial value of the arbitration parameter between stimulus-based and action-based learning, creating an imbalance in favor of the latter. The same occurred due to VS lesions but to a lesser extent. The parameter analysis also showed that VS lesions caused a reduction in the baseline stimulus-based signals (“what” task).

The presented study is interesting and methodologically robust. The manuscript is clear, although it would benefit from streamlining the presentation of the results to help the reader focus on the main findings. However, there are some major concerns that could mine the soundness and the impact of the findings from this study. Below, I outline some of the most relevant points.

Impact

The study is based on experimental data collected in previous works by the same research group (refs. 14, 15, 22). Although the manuscript highlights a clear advancement in understanding the cognitive and computational functions of the amygdala and VS, many of the phenomena arising from amygdala and VS lesions were already described in previous studies (14, 15, 22). The results obtained in this work align with prior findings and provide a more complete theoretical framework but are not exceptionally surprising.

Effort confound between the two tasks

The “what” task appears to be more difficult than the “where” task since, in the former, the subject has to identify the target they believe most rewarding and then plan a movement to the right or left. This difference in difficulty is also suggested by the longer RTs in the “what” task. This implies that the experimental subjects had to invest more cognitive resources in the “what” task compared to the “where” task. Given the importance of the VS in effort management, this could be a significant confounding factor that the authors should address.

Precision of the lesions

As indicated in previous works (e.g., ref. 15), the amygdala lesions also involved hippocampal and perirhinal structures to a quite large extent. Given that these lesions lead to a deficit in updating (learning) the arbitration process, it is difficult to assess the role of lesions in structures adjacent to the amygdala (e.g., Corbit & Balleine 2000; Cheung & Cardinal 2005; Lee... Chen 2024).

Model evaluation

The authors used k-fold cross-validation to compare the models. It would be more informative to use AIC since the models have different numbers of parameters. Additionally, since cross-validation is used, why not provide readers with a performance metric such as F1 score or accuracy? Based on the manuscript's graphs, if one converts the $-\log$ likelihood into accuracy, it becomes evident that the differences between the various models are very small. For instance, in Fig. 2b, the maximum difference in accuracy between the Two-system model with static omega and the one with dynamic omega is only 1.3%, a negligible value.

(Remarks on code availability)

Reviewer #2

(Remarks to the Author)

The presented study aimed to elucidate the distinct contributions of the ventral striatum and the amygdala to learning from various sources of reward uncertainty, a crucial aspect of understanding how neural circuits coordinate to enable adaptive behavior.

The authors designed a novel task that manipulated reward probabilities and the stimulus feature associated with these probabilities. By examining the learning patterns of subjects with lesions to either the ventral striatum (VS) or the amygdala, they inferred the role of these circuits in learning from different sources of uncertainty. Computational models fitted to the animals' behavior revealed dissociable impairments following VS and amygdala lesions. Based on their results, the authors propose that the amygdala is causally involved in the process of arbitrating between internal models to guide animals' behavior by inferring appropriate reward contingencies.

In their task, where reward probability can be specified as a function of either stimulus identity or location, VS lesions result in a systematic bias toward attributing reward probability to stimulus location, regardless of the true task contingency. In contrast, amygdala lesions lead to an impairment in appropriately attributing reward probabilities to either task contingency. The dynamic arbitration model employed to analyze the data was well-suited to the research question and allowed the authors to reconcile previous conflicting findings regarding the amygdala's role in reward-based learning.

Overall, the results support the authors' conclusion that the amygdala plays a causal role in arbitrating which internal model reward fluctuations are assigned to, and using that to appropriately select the internal model to guide behavior. There is also a very clear bias toward action-based learning in VS lesion animal. In the case of amygdala lesions it appears that neither stimulus identity or location are correctly inferred as determining reward probabilities, whereas in VS lesion case reward probabilities are largely attributed to stimulus location. Could both these deficits can be interpreted as different types of arbitration errors as opposed to the interpretation that the arbitration process is intact after VS lesions?

Minor comments:

- It would be helpful to refer to specific sections and equations in the methods when providing additional details about the analyses.
- Near equations 3 and 4, where the learning rate (α) and decay parameter are first introduced, it would be useful to specify how these parameter values were chosen. Presumably, these are distinct from the learning rate and decay parameters for the dynamic weighting model presented in equation 9.
- In the methods section describing each model, it would be helpful to explicitly specify all the free parameters, which parameters were fit, and which were hyperparameters.
- I found the analyses presented in Fig 3e-g somewhat challenging to follow, particularly in terms of understanding the key insights they were intended to convey beyond the results already presented in Fig 3b-c. Additionally, I had difficulty locating the explanation for how the baseline for these analyses was determined within the methods section.

(Remarks on code availability)

Reviewer #3

(Remarks to the Author)

This study examines the effect of lesions of ventral striatum (VS) and amygdala in monkeys performing a decision task in which there are blocks in which the reward probability is determined by stimulus identity 'what blocks' and blocks where the reward probability is determined by stimulus location on the screen 'where blocks'. The task is interesting because it requires the subjects to dynamically arbitrate between two different possible strategies (stimulus-based vs action based) when assigning credit to guide subsequent choices. On the basis of the pattern of deficits observed in the lesion groups, combined with computational modelling, the authors argue that VS lesions impair stimulus-based learning hence biasing the

animals towards an action-based strategy, whereas amygdala lesions impair both stimulus and action-based learning, but critically also affect the arbitration processes that determine which strategy is used.

The experiment and computational modelling are well motivated and well designed, and the study has potential to be an interesting and valuable contribution to the field. However, there appears to be a major issue with how the statistics have been done which will cause a high risk of false positive findings. The experiment design is inherently cross-subject as the authors are comparing the effect of lesions between different groups of animals. However, the authors appear not to have used subject as the experimental unit for statistical analyses, but rather have used within-subject observations as the statistical unit. I am not 100% sure whether the experimental unit was blocks or trials, but the number of experimental units used for statistical comparisons between groups was much larger than the number of subjects, as seen by the degrees of freedom reported, e.g. line 156: "amygdala-lesioned monkeys performed significantly better than VS-lesioned monkeys (two-sample t-test: $t(1832) = 17.72$, $p = 6.05 \times 10^{-65}$)".

Using repeated within-subject observations as the experimental unit for statistical comparison in a between-subject experimental design massively inflates the risk of false positive findings of difference between the groups. This is because the hypothesis being tested is whether the observations from the two groups come from the same distribution, which they necessarily do not because the groups contain different subjects, and there is inevitably cross-subject variability. I have created an IPython notebook which demonstrates the issue using simulated data, which can be viewed here: <https://notebooksharing.space/view/56f8440250013f8bd6f5ef38aba12054e690b92dadb5419855c44bbca23966a8>. Another way of stating the issue is that statistical tests typically assume that observations are independent and identically distributed (IID), but this assumption is not met when working with repeated observations from multiple subjects, due to cross-subject variability.

It is essential to fix these statistical issues before the scientific content of the paper can be evaluated. Please can the authors:

- Use appropriate statistical tests which take cross-subject variability into account when testing for differences between the lesion groups. The simplest approach would be to average observations within subject to obtain a single number per subject, then test for significance between groups using standard statistical tests with subject as the experimental unit. More power might be obtained by using mixed-effects regression analyses, to enable observation-level data to be compared between groups while appropriately taking between-subject variance into account. Permutation tests where subjects are permuted between groups can also be useful in situations such as testing for a difference in parameters of RL model fits, where it may be necessary to combine data from multiple subjects to accurately fit model parameters.
- When comparing data from different lesion groups in plots, ensure that the error bars or confidence regions use cross-subject variance, and report the experimental unit used in the figure legend (e.g. Error bars = SEM across subjects).
- Also on the subject of stats, if the authors want to argue that there are interactions between lesion group and task condition (i.e. what-only, what, where) then they should directly test for significant interactions using e.g. ANOVA or regression analyses.

Other points:

- Please provide a diagram summarising the flow of subjects through the different stages of the experiment, as some but not all subjects did both 'what-only' and 'what-where' versions of the task and it would be useful to visually summarise the structure of the experiment to help the reader understand what was done in what order.
- Please include a supplementary figure summarising the extent of the lesions. I know this has been included in previous publications but it would be useful for the reader to have it in this manuscript.

(Remarks on code availability)

Reviewer #4

(Remarks to the Author)

In the manuscript entitled 'Contribution of amygdala to dynamic model arbitration under uncertainty', Jae Hyung Woo and colleagues investigate the role of the amygdala in choosing which among two models of the environment (stimulus-based vs. action-based) best predict reward outcomes. They combined analyses of previously published experimental data and computational modeling to quantify concurrent learning in monkeys performing a probabilistic learning task with three forms of uncertainty: uncertainty about the better option, uncertainty about the correct model of the environment, and uncertainty about when reward associations change. The data from different groups was analyzed: bilateral lesions to the amygdala; bilateral lesions of the ventral striatum; control. They used entropy-based metrics to quantify consistency in choice and learning based on stimulus- and action-based learning over time. They found elements suggestive of a dynamic, competitive interaction between stimulus-based and action-based learning, and interpret their results as a distinct role of the amygdala. Specifically, they propose that the amygdala adjusts the initial balance between the two learning systems, thereby altering

the interaction between arbitration and learning that shapes the time course of both learning and choice behaviors.

Overall, I have a mixed feeling after reading this manuscript. Since the experimental data is mostly (if not all, see question below) taken from previously published studies, only the computational analyses are new. Some of these analyses are nice and insightful. But overall there is no strong computational result that comes out of this work, since the modeling work is incomplete (see detailed comments), since many model parameters change at the same time, and since alternative interpretations are possible at multiple stages of the work.

LIMITED ORIGINALITY

First, it is important to mention that this paper presents a novel series of analyses of data previously published in:

- Costa, V. D., Dal Monte, O., Lucas, D. R., Murray, E. A. & Averbeck, B. B. Amygdala and Ventral Striatum Make Distinct Contributions to Reinforcement Learning. *Neuron* 92, 505–517 (2016).
- Rothenhoefer, K. M. et al. Effects of Ventral Striatum Lesions on Stimulus-Based versus Action-Based Reinforcement Learning. *J. Neurosci.* 37, 6902–6914 (2017).
- Taswell, C. A. et al. Effects of Amygdala Lesions on Object-Based Versus Action-Based Learning in Macaques. *Cereb. Cortex* 31, 529–546 (2021).

About the data from the What/Where task, the authors wrote (lines 704-705) that 'The remaining control and any of the amygdala-lesioned monkeys were not used in any previous studies.' However, they have in this paragraph only mentioned studies 14 and 22. Could the authors please double confirm that the control and amygdala-lesioned monkeys were not used in study 15 (Taswell et al. 2021) either? Study 15 indeed states that 'The subjects included 10 male rhesus macaques with weights ranging from 6 to 11 kg [...] with] four of the male monkeys received bilateral excitotoxic lesions of the amygdala [...] Four out of the six unoperated control monkeys were the same monkeys used in a previous study (Costa et al. 2016). Five out of the six unoperated control monkeys were the same monkeys from an additional study (Rothenhoefer et al. 2017). All remaining monkeys were not previously used in the studies mentioned above.'

In the discussion, the authors cannot write 'In addition to replicating these results' when referring to studies 14 and 15 which are the previous studies from the same authors from which the same data has been re-analyzed here. Replicating the results per se would imply that new data is acquired in new animals.

One of the conclusions ('The similar impairments observed in amygdala-lesioned monkeys during What and Where blocks cannot be explained by the amygdala's currently assumed role in stimulus-based learning.') has already been made and further analyzed by some of the same authors in [15] (Taswell et al. 2021)

COMPUTATIONAL MODELING LIMITATIONS

Parts of the computational analyses are nice, but incomplete and not always compelling.

While the authors show good model fitting, with k-fold cross-validation, they then draw conclusions from fixed effects over all monkeys and from tiny likelihood differences. It is important to also perform random effect analyses and show the fit distribution over subjects.

Moreover, I don't find the interpretations about the way the model captures monkey behavior compelling, since all model parameters change. There is no clear result, where a single parameter significantly varies, so as to interpret behavioral properties and deficits as being due to a specific mechanism.

Rather, the computational analyses remain at a descriptive level: posterior predictive analyses are lacking; the study also lacks simulations of ablated versions of the model to show that specific artificial lesions produce the same behavior as the animals.

Moreover, to make a substantial contribution that justifies publishing yet another paper with the same data, the authors should show that the specific behavioral deficits produced by lesioning a specific part of the arbitration model enables to explain why the same authors with the same data have previously interpreted amygdala lesions as 'reduc[ing] choice consistency (sensitivity to value signals) for stimulus-based learning and increas[ing] sensitivity to negative feedback ($\alpha\alpha$ -) for action-based learning'. Similarly, they should produce VS-equivalent lesions on the arbitration model and help explain why they previously interpreted VS-lesions as producing 'deficits in learning to select rewarding images but not rewarding actions', as displaying 'more influence by negative feedback and [...] lower choice consistency than controls', and as 'only affect[ing] learning in the stochastic task' of Costa et al 2016.

The correlations with reaction times is not satisfying either, because error rates constitute a confounding factor.

The arbitration model's ability to account for 'VS lesions hasten[ing] the monkeys' choice reaction times' should also be tested.

Overall, additional analyses are required to verify that the authors are not just interpreting small differences in the identified patterns which could simply be due to noise.

ALTERNATIVE BEHAVIORAL INTERPRETATIONS

A neglected aspect which may have played an important role here is the sequentiality between the tasks, and whether having first performed the What-only task sets priors in favor of the stimulus-based strategy for the following What/Where task. In the Methods, the authors state that 'Four of the five controls and all of the three VS-lesioned monkeys were the same monkeys used in the What-only task'. Could the authors please confirm that those monkeys first performed the What-only task and then the What/Where task? What happens when animals start with the latter? What happens when animals start with a Where-only task and then perform a What/Where task? Could the authors disentangle the role of such priors during the What/Where task?

There is also a problem with the association the authors make between the What-only task and 'the absence of uncertainty about the correct model of the environment'. This is true only from an ideal observer point of view. This was not true for the monkeys, especially during initial trials of each block, and trials following reversals. This view is supported by the authors' observation that the animal's strategy was less than 80% of the trials classified as stimulus-dominant in the What-only task (Supplementary Note 1, Line 1155). Further evidence comes from the finding that arbitration models get a lower log likelihood than the stim-only model in the What-only task (Figure 2B), arguing for the animals' uncertainty about whether the action-only strategy was relevant or not in this task. A different protocol would have been required to investigate the impact of VS- and amygdala-lesions in a case of complete absence of uncertainty about the correct strategy.

In the same lines (167-170), it is not true that 'amygdala-lesioned monkeys' were 'not [...] able to partially overcome the deficit in stimulus-based learning.' They did show a mild increase of performance, both before and after the reversal. It is just that VS-lesioned animals better overcame this deficit. In general, the authors should be careful with their too frequent over-statements about results and interpretations.

Lines 202-213 and in Supplementary Figure S1, the difference between ERDS_Stim and EDS_Action is not sufficient by itself to investigate increases in the adoption of 'competing (incorrect) strategy', (1) because there can be other strategies than these two, e.g. alternation, random, etc.; (2) because plotted variations of the difference could be due either to an increase/decrease in ERDS_Stim only, an increase/decrease of ERDS_Action only, or both. Could the authors also plot ERDS_Stim as a function of uncertainty for What-only and What blocks, as well as ERDS_Action as a function of uncertainty for Where blocks, please?

Lines 627-640 of the Discussion, the argument according to which an 'arbitration weight that biases behavior toward the action-based system' could be an explanation for increased flexibility to reversal in stimulus-based tasks does not stand: The reason is that such a bias predicts a lower performance than controls before the reversal, which is not the case in the cited studies (ex : Izquierdo et al. 2013).

ABUSE OF LANGUAGE

The employed terminology sounds often like an overkill. For instance, why talking about a 'model of the environment', while it is just about the action reference frame (stimulus- vs. space-based)? The authors themselves nuance their terminology in the discussion, stating that a variety of arbitration models exist, and that their model includes a 'more basic arbitration'. I think they should simplify the terminology from the very beginning of the paper, including title, abstract and introduction, so as not to disappoint readers expecting to find an arbitration mechanisms between different internal world models.

The authors are also abusing language when citing the studies of Birrell & Brown, Ragozzino et al., Floresco et al., as if their task made it easy to find the more rewarding option: 'In other paradigms, the more rewarding option is easily ascertained, but there is uncertainty about the correct model of the environment and when to choose that option8–10.'

Overall, the way to present the main results constitutes itself overselling, due to the multiple limitations mentioned above about the computational analyses. I think the authors should either modify their claim that they have an implementation model of amygdal function, and rather refer to it as a descriptive model which can be useful to identify latent variables and describe things that happen in the monkey. Alternatively, if the authors want to capture the function of the amygdala, they should not just compare a set of arbitration models, but rather also test alternative models, including simpler models where a single mechanism produces different adaptations to different environmental changes (e.g., metaplasticity, or Wang, Kurth-Nelson and Botvinick's meta reinforcement learning model), instead of models requiring two mechanisms in parallel.

MINOR POINTS

When looking at reaction times in the What-only task, and observing that 'most action-dominant trials happened when reward value estimates based on the two systems were close to each other, resulting in slower and more erroneous responses', I think it's important to state (both in manuscript and supporting information) that the higher error rate may be a possible confound when interpreting RT difference between stimulus- and action-dominant trials. This is especially true given that the authors themselves wrote in Supplementary Material that 'performance was higher for the correct strategy'.

To further investigate the effect of strategy on RT in the What/Where task, the authors should restrict their analysis to the last third of each block, where performance has reached a plateau, and at this stage see whether there is a different in RT between stimulus-dominant and action-dominant strategy.

There is already an old literature on spatial and object reversal learning, which is not cited here. See for instance Helen Mahut (1971) Spatial and object reversal learning in monkeys with partial temporal lobe ablations. *Neuropsychologia*. The

paper also investigates the effect of amygdala lesions and should thus be discussed here.

(Remarks on code availability)

Version 1:

Reviewer comments:

Reviewer #1

(Remarks to the Author)

About the manuscript:

All my comments were addressed in the revision. I have no further suggestions.

About the code:

I cloned the repo, but the demo does not work, even after including all the dirs in the matlab path.

(Remarks on code availability)

I cloned the repo, but the demo does not work, even after including all the dirs in the matlab path.
error:

Unable to find file or directory

'dataset/preprocessed/all_stats_control.mat'.

Reviewer #2

(Remarks to the Author)

The authors have made substantial updates in response to the comments from the reviewers and my comments have been sufficiently addressed.

(Remarks on code availability)

Reviewer #3

(Remarks to the Author)

The authors have engaged with the concerns raised in my previous review and appear to have made a good faith effort to correctly take cross-subject variability into account in both their statistical analyses and visual presentation of the data.

The updated versions of the plots now showing cross subject SEM error bars are a great improvement, thanks for doing this. The diagram showing the flow of animals through the experiment is also really useful.

The move to using mixed effects models for most statistical tests is sensible, as it provides a principled way to take cross-subject variability into account while maximising statistical power. However, many aspects of how these analyses were setup are not clear from the text, which makes it hard to interpret the reported stats and to tell whether the analyses have been set up appropriately.

To illustrate this, I will provide some examples of reported results:

Line 161: "amygdala-lesioned monkeys exhibited the largest impairment in performance (P(Better) across all three reward schedules: $M \pm SD = 0.581 \pm 0.11$; main effect of group in mixed-effects analysis; $\beta_{VS} = 0.054$, $p = .0061$; $\beta_{control} = 0.173$, $p = 3.25 \times 10^{-21}$; Fig. 1e). VS-lesioned monkeys also showed impairment compared to the control monkeys (main effect of group in mixed-effects analysis; $\beta_{VS} = -0.120$, $p = 1.97 \times 10^{-12}$; Fig. 1e)."

It is unclear from the above how the reported β_{VS} and $\beta_{control}$ demonstrate that amygdala lesioned monkeys exhibited the largest impairment in performance. It is also unclear why are two different values reported for β_{VS} (0.054 and -0.120), for what is apparently the same analyses of the same figure's data (main effect of group in mixed-effects analysis, Fig. 1e). The reporting of beta weights and P values associated with the control group ($\beta_{control}$) appears inconsistent with the statement at line 1138 in the methods stating: "We tested this mixed-effects model on each of the parameters compiled across groups and tasks, with control monkeys as the reference group", because the reference group for a categorical variable in a regression typically does not have a beta associated with it.

My best guess at what the authors have done here is that they have run two different regression analyses of the same data using different groups as the reference category, in order to test for different things. For the first reported stats ($\beta_{VS} = 0.054$, $\beta_{control} = 0.173$) I assume the amygdala group has been chosen as the reference category such that P values for β_{VS} and $\beta_{control}$ test for significant differences between these groups and the amygdala group. For the second reported stats ($\beta_{VS} = -0.120$) I assume the control group have been chosen as the reference category such that the p value for β_{VS}

indicates whether the VS group are significantly different from the controls.

If this is correct, the decision to test for differences between the lesion groups by using the amygdala group as the reference group is not ideal because (a) it is an arbitrary choice – why use the amygdala group rather than the VS group, and (b) it is confusing for the reader to report p values for β_{VS} and $\beta_{control}$ in order to say something about the amygdala group. To test for differences between the two lesion groups it would be preferable to either i) Run a regression with one predictor coding for the difference between control and both lesion groups and a second predictor coding for the difference between the two lesion groups. Or ii) Run a regression with the control group as the reference group and a predictor for each lesion group, then compute the contrast between the lesion groups from the output of this regression (I think the emmeans package in R provides functionality to do this, see e.g. <https://www.r-bloggers.com/2019/03/getting-started-with-emmeans/>).

Another example:

Line 170: “During the What blocks of the What/Where task, however, amygdala-lesioned monkeys performed significantly better than VS-lesioned monkeys (mixed-effects analysis with group \times block type; $\beta_{amyg:What} = 0.0855$, $p = 1.67 \times 10^{-12}$; Fig. 1f inset), while both lesioned groups were impaired relative to control monkeys (amygdala: $\beta_{amyg:What} = -0.0679$, $p = 3.48 \times 10^{-52}$; VS: $\beta_{VS:What} = -0.180$, $p = 7.94 \times 10^{-7}$). In the Where blocks that did not require stimulus-based learning, only amygdala-lesioned monkeys showed impairments in performance relative to controls ($\beta_{amyg:Where} = -0.0212$, $p = 4.48 \times 10^{-6}$; Fig. 1g inset). VS-lesioned performance was comparable to that of controls ($\beta_{VS} = -0.00871$, $p = .811$) and better than that of amygdala-lesioned monkeys ($\beta_{amyg:Where} = -0.0361$, $p = .00298$), suggesting no deficits in action-based learning for the VS lesioned monkeys.”

How do stats for predictor $\beta_{amyg:What}$ test for differences between amygdala and VS groups if the reference groups is the control group as stated in the methods? Why are two different beta reported for the same predictor ($\beta_{amyg:What} = 0.0855$, $\beta_{amyg:What} = -0.0679$)? Were separate regressions run for the different block types or were multiple block types included in the regression with block type as a categorical predictor? If the latter then how was the block type predictor coded? Did the random effects structure account for potential differences between subjects in the effect of block type, or just consider a random intercept?

To clarify how the statistics have been run, please provide a summary of the different regression analyses that have been run on the data, specifying for each regression:

1. What data was included in the regression (which tasks, block types, and lesion groups).
2. What the dependent variable was and what each observation corresponded to (e.g. performance in one block).
3. What predictors were included and how categorical predictors were coded.
4. What the fixed and random effects structure was (provide the formula used to specify it).
5. What hypothesis the regression was designed to test (e.g. differences between which groups or block types), and which predictor are relevant for testing these.

Providing the full results tables for the key regressions along with this information would be really helpful for understanding how the stats reported in the text were run.

A related statistical issue is that in several places the text imply differences between block types or groups, but the stats reported do not appear to correspond to direct comparisons between the groups or conditions that would be needed to demonstrate a significant difference. E.g.:

Lines 411:426: “When the stimulus-based system was more reliable, the effective arbitration rates toward the stimulus-based system were larger in control monkeys” and “In contrast, amygdala-lesioned monkeys showed the minimum differentiation between adjustments toward the more and less reliable (correct and incorrect) systems”. These statements seem to imply differences between groups and/or conditions, but the only P values reported are for intercept predictors β_0 which do not appear suitable for making statements about such differences.

Line 573: “we observed a significant decrease in ERDSSstim in VS-lesioned monkeys (VS: $\beta_{session(\%)} = -0.207$, $p = 1.03 \times 10^{-5}$), whereas there was no evidence for changes in control or amygdala-lesioned monkeys across time (Control: $\beta_{session(\%)} = -0.0592$, $p = .285$; amygdala: $\beta_{session(\%)} = -0.0108$, $p = .802$.” Again, a difference between groups is being suggested in the absence of any statistics associated with a direct comparison between the groups. A difference in significant is not itself evidence of a significant difference (see <https://doi.org/10.1198/000313006X152649>). Please ensure that all statements about differences between groups and/or conditions are supported by appropriate statistical tests.

Please also report how the new permutation tests reported in the results were run, i.e. what was permuted and how the p values were computed.

(Remarks on code availability)

Reviewer #4

(Remarks to the Author)

The authors have admitted that their dataset is not original, and that they here test a linear combination of two already existing models. Thus, the conceptual novelty is very limited.

Nevertheless, I thank the authors for performing a few additional analyses which have improved a bit their manuscript (model comparison + model simulation (Fig.S7)), but which nevertheless confirm that the data does not support the authors' conclusions, as they show that neither model correctly reproduces the animal data.

This manuscript should not be accepted for publication in such a high-impact journal given its strong limitations.

LIMITED ORIGINALITY

- The authors have clarified in the rebuttal that "the data used in our study have been previously published". However, there are still sentences in the manuscript that may mislead readers by making them think some data have been newly collected: "The remaining control and the four amygdala-lesioned monkeys were newly trained for this study". The authors should more clearly state that this paper does not present any new data.
- Regarding conceptual originality, their new model is, as the authors state, consistent with the previous one, which was based on the exact same dataset. Hence, beside having the two models (one for each task) linearly combined into one, there is no conceptual novelty proposed in this paper.
- The best fitting model, a `_dynamic_` combination of both RL models, does not constitute a conceptual breakthrough for multiple reasons:
 - 1) Presented analyses lack compelling evidence that the dynamic part adds on significant explanatory power: the authors insist that the amygdala lesion is specifically associated with a lower ω_0 (ignoring the difference in α - and α_{ω} cf Fig S9, S10), which corresponds to the baseline arbitration. The fact that this affects the arbitration weight dynamically in simulations does not imply that this dynamic mechanism is necessary to explain the behavioral patterns, compared to a static arbitration weight. Adding ω_0 in regression analyses of the RT (figure SN2) and showing simulations of the static model in posterior predictive checks (figure S7) to see if it behaves significantly worse than the dynamic model, could help support the proposed dynamic model as a real breakthrough.
 - 2) The ambiguity surrounding the lack of novelty in the data presented overshadows the explanatory potential of the proposed model. What predictions would this dynamic arbitration model does about neural circuitry and cognition in other tasks? What would be the experimental design to falsify a static model or other types of arbitration? Lines 708-732 the authors discuss known results about the role of amygdala in reversal tasks, and how the experimental data presented at the beginning of the paper fit in. But these data have already been published, and a theoretical paper presenting a conceptual novel and unifying model should present more simulations to support these verbal assumptions.

COMPUTATIONAL ANALYSES

- The authors have improved the model comparison by showing individual cross-validated log-likelihood and performing the cross-validation across all the experimental blocks.
- New simulations of different models compared with the data make an important contribution (figure S7). They do not support the authors' conclusions, however, as they show quite clearly that neither model correctly reproduces the animal data. Furthermore, as mentioned above, the same simulations using the static arbitration model (with a common ω across the blocks) are needed to conclude that the dynamic arbitration aspect is a necessary ingredient.

(Remarks on code availability)

The code is well commented and should help reproducibility.

Version 2:

Reviewer comments:

Reviewer #3

(Remarks to the Author)

The authors have made a thorough and good faith effort to address the issues raised in my previous review and have massively improved the clarity of the stats reporting through the use of contrasts to assess differences between the two lesion groups.

The revised manuscript is an interesting and valuable piece of work which I believe has substantial novelty, through introducing and carefully providing evidence for a model of dynamic arbitration between stimulus and action-based learning, and the effects of VS and amygdala lesions on this.

Minor point: I think there is a typo in supplementary table 3.3 as the model formula does not include group, but there are values detailed for coefficients group_amygdala and group_VS.

(Remarks on code availability)

Reviewer #4

(Remarks to the Author)

I would like to thank again the authors for all the efforts they made to significantly improve their manuscript. I do acknowledge, as I did before, that they have improved their manuscript by performing additional analyses. I acknowledge the further clarifications and analyses that they have provided at this new stage in response to Reviewers #2, #3 and to me.

"However, we were concerned by the accusatory tone in phrases such as "the authors have admitted...". [...] First, regarding the originality of the dataset, we want to emphasize that we have never intended to obscure or downplay the fact that these datasets were previously published, contrary to the reviewer's accusatory comment."

I thank again the authors for having now clarified their manuscript about the data. While I don't pretend that the previous formulations about the experimental data (e.g., "All remaining monkeys were not previously used in the studies mentioned above") had been intentional, I think the authors had not made enough efforts to make it very clear initially that all the data had already been published before. Their answers to my previous and present comments speak for themselves:

Previous stage: "[R4.1] [...] In the process, we also identified an error regarding the monkeys used across experiments: only one of the control monkeys (monkey #2), and not four as previously stated, and all three VS- lesioned monkeys participated in both the What-only and What/Where tasks."

Present stage: "[R4.2.1] We thank the reviewer for catching this. By "this study" we had originally meant to refer to the later What/Where task, not the current manuscript. We have now fixed the given sentence [...]"

This is not a negligible issue since the formulations adopted in an initial submission leave first impressions to reviewers that may contribute to inflating their impression about the originality. This is why I think it is deontologically not acceptable.

Nevertheless, as for computational analyses, I would like to thank the authors for providing characterizations of the interaction between the two components of their computational model. I agree that this brings insights into the dynamic arbitration mechanism that balances the influence of stimulus-based and action-based systems on choice behavior, thus going beyond the static omega model.

The manuscript has now been further improved and can be published. I still consider that the originality is too limited for Nature Communications. But I leave this decision to the editor.

(Remarks on code availability)

Round 1: RESPONSE LETTER TO THE REVIEWERS' COMMENTS AND SUMMARY OF REVISIONS

Title: Contribution of amygdala to dynamic model arbitration under uncertainty

Authors: Woo, Costa, Taswell, Rothenhoefer, Averbeck, & Soltani

We would like to thank the four reviewers who have carefully assessed our manuscript and provided helpful comments in detail. To address the reviewers' concerns, we have performed additional analyses and made substantial changes to our manuscript. Below we provide point-by-point responses to each reviewers' comment. New changes and analyses have been indicated in blue and marked with [R#. #] in the revised manuscript (e.g., [R1.1] indicates changes in response to the point # 1 of the Reviewer #1). We also have provided a cleaned version where these placeholders are omitted. Below is a summary of key revisions.

First, in response to the comments about model-fitting results, we have revised our cross-validation procedure to make the validation procedure more stringent. Specifically, we now cross-validate all blocks for each monkey in each task *without* distinguishing the block types (for the case of the What/Where task) or the reward schedules (for both the What-only and What/Where tasks). This is indeed closer to the actual block progression as experienced by the monkeys, who did not have access to the ground truths about the reward schedule or the block type. Because this new procedure does not estimate separate sets of parameters based on block types or reward schedules, the resulting log-likelihood values of all models have slightly increased overall, making the new method more conservative in terms of model-fitting performance.

Moreover, we have refined the statistical tests to account for the inter-subject variability. As Reviewer #3 pointed out, statistical tests which do not account for cross-subject variability may inflate the risk of finding false positive results. To address this issue, we have primarily used mixed-effects regression analyses, which preserve the statistical power while accounting for the variability among subjects. Although some previously significant results are no longer statistically significant with the revised method, all the main claims of the study remain qualitatively unchanged.

Please see our point-by-point responses to individual reviewers' comments below.

Reviewer #1 (Remarks to the Author):

The authors presented a study in which they reanalyzed lesion and behavioral data from non-human primates from three previously published studies. They compared the ability of different reinforcement learning (RL) based computational models to predict the behavior of a control group, a group with bilateral lesions in the ventral striatum (VS), and a group with bilateral lesions in the amygdala. The behavioral tasks consisted of two types of two-armed bandit tasks. In the first (what-only task), subjects had to identify the

stimulus associated with a higher probability of reward regardless of the side of presentation, while in the second, they were exposed to a non-stationary environment alternating between bandits where reward probability was linked to stimulus features and bandits where the location of the stimulus (regardless of its perceptual features) was reinforced (what/where task).

The authors demonstrated that the computational model that best captured the behavior of the subjects included a dynamic arbitration mechanism between stimulus-based and action-based learning (updated trial-by-trial) and different baselines for stimulus-based and action-based learning. All lesioned subjects showed a bias towards action-based learning (where), while only the subjects with amygdala lesions exhibited a deficit in the arbitration process, which appeared to converge toward optimality much more slowly than in the controls (comparison in Figs. 2d and 3b). This phenomenon also emerged from the analysis of the update rate in the arbitration process, which was deficient in the amygdala-lesioned subjects for both the “what” and “where” environments. Finally, through the analysis of the parameters optimized on the behavioral data, the authors showed that amygdala lesions resulted in a shift in the initial value of the arbitration parameter between stimulus-based and action-based learning, creating an imbalance in favor of the latter. The same occurred due to VS lesions but to a lesser extent. The parameter analysis also showed that VS lesions caused a reduction in the baseline stimulus-based signals (“what” task).

The presented study is interesting and methodologically robust. The manuscript is clear, although it would benefit from streamlining the presentation of the results to help the reader focus on the main findings. However, there are some major concerns that could mine the soundness and the impact of the findings from this study. Below, I outline some of the most relevant points.

We appreciate the reviewer’s positive feedback and valuable suggestions. We hope that our responses below, along with the accompanying revisions to the manuscript, adequately address their concerns.

Impact

The study is based on experimental data collected in previous works by the same research group (refs. 14, 15, 22). Although the manuscript highlights a clear advancement in understanding the cognitive and computational functions of the amygdala and VS, many of the phenomena arising from amygdala and VS lesions were already described in previous studies (14, 15, 22). The results obtained in this work align with prior findings and provide a more complete theoretical framework but are not exceptionally surprising.

[R1.1] We appreciate the reviewer’s concern regarding the novelty of our results in relation to previous findings using the same datasets. First, the main novel contribution of our study is to investigate the interaction between stimulus- and action-based learning systems, instead of examining them in isolation as in the original three studies. To that end, we used a new conceptual framework based on model arbitration to re-analyze the data while also considering the models used in the previous studies (single-system, stimulus-only and action-only models). This allowed us to better account for and understand the behavior of both control and lesioned monkeys. It also helped connect findings from the three original studies and provided explanations for previously reported contradictory effects of amygdala lesions (e.g., the puzzling effect of amygdala lesions on facilitating reversals during stimulus-based task). The goal of our study was thus to provide a common conceptual framework across multiple datasets, which have not been done previously for monkeys or other animals. Critically, our new computational models more accurately capture the choice behavior of monkeys across all groups and all experiments than the models used in the original three studies, while also identifying specific impairments due to amygdala and VS lesions.

In response to the reviewer’s feedback, we have streamlined the main findings to better distinguish them from the results of previous studies using the same datasets. To summarize, the overall flow of our analyses using multiple computational approaches involved the following steps: (i) providing evidence for the presence of multiple learning systems; (ii) fitting various RL models (including those in the original studies) to animals’ choice data and demonstrating the superiority of the dynamic two-system models in accounting for data across all conditions (**Fig. 2b, 3a**); (iii) model and parameter recovery to ensure that the parameters of the winning model are well recovered and performing descriptive analysis of behavior based on the estimated parameters (**Fig. 2c–e, 3b–g, 4a–c**); (iv) validating that the model also captures the key aspects of the data through simulations (e.g., **Fig. S6, S7**); (v) analyzing key parameters related to arbitration (ω , ρ) to identify the specific effects of lesions on the arbitration process (**Fig. 4d–f**); (vi) illustrating how dynamic interactions between learning and arbitration can create diverse behavior, with implications for understanding results of other amygdala-lesion studies (**Fig. 5**).

We have revised the following sections in the Introduction and Discussion to highlight how our approach differs from those of the three original studies and provides additional insights.

(In the last paragraph of Introduction; Line #83)

“Additionally, we developed several new reinforcement learning (RL) models that, along with previous models, were used to fit choice behavior on a trial-by-trial basis. These new models extended the previous ones by incorporating static or dynamic arbitration among alternative learning systems, based on different signals. We then examined the best models and their estimated parameters, particularly those related to the arbitration process, to pinpoint the roles of the amygdala and ventral striatum in reward learning and arbitration. Moreover, by modulating the key parameters of the model, we simulated and replicated the distinct behavioral signatures of amygdala or VS lesions. Together, by utilizing the above methods, we

provide evidence for interactions between stimulus-based and action-based learning under uncertainty, uncover mechanisms underlying arbitration between the two systems, explore how arbitration and learning processes interact, and determine the amygdala's contributions to arbitration and overall behavior.”

(In the first two paragraphs of Discussion; Line #610)

“Here, we applied a combination of computational approaches to reanalyze data from control monkeys and those with amygdala and VS lesions^{17,18,24} to explore the interaction between stimulus- and action-based learning and to uncover computational and neural mechanisms underlying arbitration processes. Our main goal was to investigate the interaction between stimulus- and action-based learning systems, instead of examining them in isolation as in the original studies. Using multiple behavioral metrics, we found evidence for competitive interaction between the two learning systems. Moreover, by developing various models with arbitration and fitting choice data to these and competing models, we tested the plausibility of various mechanisms for estimating reliability signals that guide arbitration processes. Using this approach, we mapped the distinct effects of amygdala and ventral striatum (VS) lesions onto two key parameters of the model: the initial state of the arbitration (ω_0) and the relative baseline strength of stimulus-value to action-value signals (ρ), respectively. For amygdala-lesioned monkeys, the reduced initial arbitration weight was implicated in undifferentiated arbitration rates toward and away from the correct learning system for a given environment, suggestive of its role in identifying and/or retaining the correct model of the environment, or biasing arbitration toward it. Our model simulations also illustrated that the interaction between learning and arbitration processes generates diverse behaviors with strong dependency on the initial state.

Previous studies using the same dataset have identified deficits in both stimulus-based and action-based learning due to amygdala lesions^{17,18}, but they considered these deficits independently, as they did not examine the interaction between stimulus- and action-based learning. Using a single-system RL model, they found that amygdala lesions reduce choice consistency (sensitivity to value signals) for stimulus-based learning¹⁷ and increase sensitivity to negative feedback (α_-) for action-based learning¹⁸. We found consistent results using our new two-system model with dynamic arbitration (**Supplementary Fig. S9b, d**), and moreover, provide a unified account based on the dynamic interaction between learning and arbitration under uncertainty. Specifically, our simulation results mimicking amygdala lesions (**Fig. 4**) suggest that a biased initial state strongly favoring action-based learning is the key feature of the deficits observed in amygdala-lesioned monkeys. This strong initial bias altered the interaction between decision-making, learning, and arbitration processes, making the arbitration update rates between two systems to be less distinguishable from each other compared to the controls. Importantly, VS-lesioned monkeys exhibit a smaller sensitivity to the stimulus-based compared to the action-based signals. As a result, VS lesions led to an overall bias in arbitration update rates toward action-based learning in both What and Where blocks.

Effort confound between the two tasks

The “what” task appears to be more difficult than the “where” task since, in the former, the subject has to identify the target they believe most rewarding and then plan a movement to the right or left. This difference in difficulty is also suggested by the longer RTs in the “what” task. This implies that the experimental subjects had to invest more cognitive resources in the “what” task compared to the “where” task. Given the importance of the

VS in effort management, this could be a significant confounding factor that the authors should address.

[R1.2] We thank the reviewer for raising this concern about the difference in efforts required for performing the What and Where tasks. Even considering the role of the VS in effort management, there are several reasons why the efforts required to perform the two tasks are not significantly different and should not impact our results or conclusions. First, in these tasks, monkeys made decisions using saccadic eye movements, which are not effortful by any means. Second, in terms of cognitive resources, the only additional resource needed for the What task (compared to the Where task) is the transformation of the choice made in stimulus space into left or right saccades based on the stimuli's locations. This additional step is routinely performed by the brain, which likely explains why the overall difference in reaction time between the What and Where blocks is very small across all groups (see **Fig. R1.2** below). Third, results from the What-only task clearly show that monkeys with VS lesions can recover some of their impairments, which is inconsistent with the idea that effort management plays a significant role in the performance of What task.

Nonetheless, to directly assess whether the response times of VS-lesioned monkeys were influenced by task variables differently from those of control and amygdala-lesioned monkeys, and to explain the shorter RT observed in VS-lesioned monkeys, we performed additional regression analyses. As shown in **Fig. SN2** below, we found that all task-relevant variables (including effective arbitration weight Ω , which controls arbitration) had qualitatively similar effects on RT across all groups (control and both lesion groups). This suggests that effort management is not particularly affected in VS-lesioned monkeys. Interestingly, the larger impact of Ω on RT for VS-lesioned animals may partially explain the shorter RT observed in this group.

Finally, the main contribution of our study is to elucidate the role of the amygdala in arbitration, which is not influenced by a potential role of the VS in effort management. We have included **Fig. SN2** below and the accompanying results in **Supplementary Note 2**. Moreover, we have added the following to the Discussion (Line #668), along with new references (refs 31-34 below) to mention the possible contributions of VS to effort management:

“Interestingly, we found that while VS-lesioned monkeys exhibit a bias toward the action-based strategy during the What/Where task, their response times are also shorter than those of control and amygdala-lesioned monkeys. Given the significant involvement of VS in effort exertion³¹⁻³⁴, the shorter RT in VS-lesioned monkeys could be linked to the fact that the stimulus-based strategy requires more cognitive effort. This is reflected in the slightly longer RT in the What task compared to the Where task, as well as the positive correlation between RT and arbitration weight. As a result, VS-lesioned monkeys may default to the action-based strategy, allowing them to perform the task faster. The stronger reliance on the action-based strategy in VS-lesioned monkeys can be adequately explained by the reduction in the ρ parameter of our model, without necessarily suggesting an impaired arbitration process.”

31. Schoupe, N., Demanet, J., Boehler, C. N., Ridderinkhof, K. R. & Notebaert, W. The Role of the Striatum in Effort-Based Decision-Making in the Absence of Reward. *J. Neurosci.* **34**, 2148–2154 (2014).
32. Dobryakova, E., Jessup, R. K. & Tricomi, E. Modulation of ventral striatal activity by cognitive effort. *NeuroImage* **147**, 330–338 (2017).
33. Silveira, M. M., Tremblay, M. & Winstanley, C. A. Dissociable contributions of dorsal and ventral striatal regions on a rodent cost/benefit decision-making task requiring cognitive effort. *Neuropharmacology* **137**, 322–331 (2018).
34. Kim, A. & Chib, V. S. Neural substrates underlying the expectation of rewards resulting from effortful exertion. (2024) doi:10.1101/2024.02.01.578411.

Figure R1.2. Reaction time (RT) by task and block types. (a, b) RT of control monkeys during the What-only (a) and the What/Where task (b). Reported are mean and standard deviation of RT by block types. (c, d) RT of amygdala-lesioned monkeys. (e, f) RT of VS-lesioned monkeys. Overall, there was little difference between RT during the What and Where blocks of the What/Where task.

Figure SN2. Regression of reaction time (RT) on arbitration weight and other task-related variables. Plotted are standardized regression coefficients from a mixed-effects model with subjects as a random effect, for given lesioned groups and tasks. Asterisk indicates significant effect of a given predictor (*: $p < .05$, **: $p < .01$, ***: $p < .001$). *corr. vs. incorr.*: the main effect of choosing the option with higher reward probability (correct option). (a, c, e) Regression coefficients for predicting RT data during What-only task, in controls (a), amygdala- (c), and VS-lesioned (e) monkeys. (b, d, f) Regression coefficients for predicting RT data during What/Where task, in controls (b), amygdala- (d), and VS-lesioned (f) monkeys. Overall, task-related variables have similar impacts on RT in all groups.

Precision of the lesions

As indicated in previous works (e.g., ref. 15), the amygdala lesions also involved hippocampal and perirhinal structures to a quite large extent. Given that these lesions lead to a deficit in updating (learning) the arbitration process, it is difficult to assess the role of lesions in structures adjacent to the amygdala (e.g., Corbit & Balleine 2000; Cheung & Cardinal 2005; Lee... Chen 2024).

[R1.3] We thank the reviewer for raising this point and the relevant literature. As noted in the previous study by Taswell et al. (2021), the amount of damage estimated from the T-2 weighted MRI scans typically overestimate the unintended extra-amygdala damage by a significant amount (Basile et al., 2017 *Frontiers*). Supporting this claim, this study had previously observed no apparent correlation of behavioral measures with the amount of inadvertent damage to the surrounding structures.

Nevertheless, to directly confirm this effect specific to our current findings, we tested whether the amygdala-lesioned monkeys exhibit systematic changes in the behavioral impairment as a function of extra-amygdala damage. In particular, we tested two metrics most relevant to our hypothesis about arbitration between competing learning strategies in the What/Where task: (i) $\Delta ERDS = ERDS_{Stim} - ERDS_{Act}$, measuring the relative adoption of action-based strategy over stimulus-based; and (ii) ω_0 , the initial arbitration weight. For this analysis, we used linear mixed-effects models that included extra-amygdala damages and the session number as fixed effects, and subjects as random effects. The results indicated no significant effects of extra-amygdala damage on either $\Delta ERDS$ ($\beta_{extra} = 0.0017, p = .632$) or ω_0 ($\beta_{extra} = 0.0022, p = .442$). In contrast, we observed that the extent of (intended) amygdala damage, when included as an additional fixed effect in the above model, significantly predicts $\Delta ERDS$ ($\beta_{amyg} = 0.034, p = .0088$) and ω_0 ($\beta_{amyg} = 0.027, p = .0127$), whereas the effects of the extra-amygdala damage extent remain insignificant ($\Delta ERDS: \beta_{extra} = -0.0028, p = .240$; $\omega_0: \beta_{extra} = -0.0018, p = .453$).

We believe that these results are consistent with the interpretation that the amount of damage in amygdala-lesioned monkeys is overestimated. More importantly, they suggest that the targeted lesion of the amygdala is mainly responsible for the behavioral effects observed in these monkeys and thus, our findings are not affected by extra-amygdala damage.

Model evaluation

The authors used k-fold cross-validation to compare the models. It would be more informative to use AIC since the models have different numbers of parameters. Additionally, since cross-validation is used, why not provide readers with a performance metric such as F1 score or accuracy? Based on the manuscript's graphs, if one converts the $-\log$ likelihood into accuracy, it becomes evident that the differences between the various models are very small. For instance, in Fig. 2b, the maximum difference in accuracy between the Two-system model with static omega and the one with dynamic omega is only 1.3%, a negligible value.

[R1.4] We thank the reviewer for raising this concern. Regarding the use of cross-validation compared to metrics like AIC or BIC, we note that cross-validation is a more powerful and stringent technique for addressing overfitting. Although AIC and BIC attempt to account for overfitting by adding a penalty term to the negative log-likelihood based on the number of free parameters, cross-validation directly tests for overfitting by evaluating the model's performance on data that was not used in the fitting process.

Nonetheless, to demonstrate our results do not depend on the specific goodness-of-fit metric, we also computed the average AIC in each block for different models (**Fig. R1.4** below). Importantly, the goodness-of-fit based on AIC yields consistent results to those using cross-validated negative log-likelihood. The only minor difference is that in the VS-lesioned group of the What-only task, the AIC for the winning model is not statistically different from the second-best model (asterisks in **Fig. R1.4** show significant difference between the best and second-best model). Therefore, goodness-of-fit based AIC and goodness-of-fit based on cross-validated negative log-likelihood yield similar results.

Figure R1.4. Average AIC (normalized by block counts within sessions) based on six models used to fit data in the three groups of monkeys. In all cases, the Dynamic ω - ρ model provides the best fit, consistent with the fitting based on cross-validated negative log-likelihood. Asterisks indicate significant difference between the best and second-best model based on rank sum test.

With regard to the concern about small differences in negative log-likelihoods, it is important to note that these values are average across many instances of the held-out data in the cross-validation. Therefore, while the improvement in negative log-likelihood seems minimal on average, even a small increment in the negative log-likelihood results in a significant improvement in the predictability of the test data as the size of the tested dataset gets large. To explain this point in more detail, we have added a **Supplementary Note 3** (attached below). In brief, even a small difference in the negative log-likelihood, such as 0.05, implies that the test data is approximately $\exp(N \cdot 0.05)$ times more likely to have come from the superior model with the lower negative log-likelihood, where N is the number of blocks. As N gets larger, this quantity also exponentially increases. Considering that the median number of tested blocks across subjects for a given cross-validation instance was 92 in our case ($M \pm SD = 97.11 \pm 30.6$), the negative log-likelihood difference of 0.05 implies that the data is $\exp(N \cdot 0.05) = 99.48$ times more likely to have

originated from the model with lower likelihood. Please, also refer to Mohammadi, ..., Pillows, 2024, *bioRxiv* for a similar argument.

Also to provide an absolute measure for goodness-of-fit similar to the accuracy, we now report *McFadden R-squared*, which is an established measure for assessing how well a model fits the data compared to a null model (McFadden, 1973). Below we have provided revised **Fig. 2a** (control monkeys) and **Fig. 3a** (lesioned monkeys) with additional McFadden R-squared reported in parentheses next to -LL. Finally, we have now included the goodness-of-fit for each model and for individual monkeys in each task in **Supplementary Table 2** (attached below), demonstrating that the Dynamic arbitration model consistently provides a better fit for the choice behavior of all monkeys.

Figure 2. (b) Comparison of goodness-of-fit across models. Plotted is the mean negative log-likelihood over all cross-validation instances for each task: What-only (black), What/Where (gray). Numbers in parenthesis indicate McFadden R² (Eq. 20).

Figure 3. (a) Comparison of the models' goodness-of-fit for choice behavior of amygdala- (left) and VS- (right) lesioned monkeys. Plotted is the mean negative log likelihood over all cross-validation instances for each task: What-only (black), What/Where (gray). Numbers in parenthesis indicate McFadden R².

What-only task (11 monkeys)											
Groups	Controls				Amygdala-lesioned				VS-lesioned		
Subjects	E (1)	G (2)	M (3)	N (4)	C (5)	CB (6)	P (7)	Q (8)	F (9)	A (10)	G (11)
RL _{Stim-only}	25.77	16.34	24.92	25.86	49.24	39.31	43.41	42.85	43.04	42.41	44.53
RL _{Action-only}	54.46	55.18	54.67	54.96	50.14	45.76	50.36	54.51	48.94	48.95	48.95
Static ω (RL _{Stim+Act})	25.65	16.05	24.81	25.77	47.86	37.79	42.44	42.42	37.62	38.63	43.8
Dynamic ω : RPE	25.63	16.06	24.81	25.77	47.73	37.67	42.36	42.40	37.35	38.24	43.8
Dynamic ω : V_{cho}	25.39	14.97	24.66	25.69	47.63	37.46	42.15	42.13	36.24	37.69	43.74
Dynamic ω - ρ (V_{cho})	25.38*	14.96*	24.55*	25.65*	47.49*	37.39*	42.15*	42.13*	36.24*	37.53*	43.73*

What/Where task (13 monkeys)													
Groups	Controls						Amygdala-lesioned				VS-lesioned		
Subjects	G (2)	R (12)	U (13)	V (13)	W (14)	B (16)	Be (17)	En (18)	MJ (19)	Z (20)	F (9)	A (10)	G (11)
RL _{Stim-only}	38.27	41.21	49.24	42.2	44.92	42.48	53.16	47.25	45.47	51.27	54.27	53.48	52.67
RL _{Action-only}	47.76	47.3	43.89	43.42	41.36	40.33	36.83	45.32	47.27	47.73	34.92	33.69	36.83
Static ω (RL _{Stim+Act})	26.07	30.16	35.88	25.65	26.26	22.59	33.37	37.08	35.51	43.67	33.6	32.9	35.36
Dynamic ω : RPE	25.92	30.09	35.76	24.62	25.25	21.68	32.38	36.73	33.32	43.47	33.44	32.67	34.93
Dynamic ω : V_{cho}	24.92	29.32	34.72	22.88	23.31	18.53	31.88	35.86	33.44	42.57	32.68	32.23	34.43
Dynamic ω - ρ (V_{cho})	24.84*	29.19*	34.56*	22.88*	23.29*	18.51*	31.86*	35.66*	33.33*	42.41*	32.62*	32.18*	34.43*

Supplementary Table 2. Cross-validated negative log-likelihoods of RL models, separately for each monkey in a given task. Reported are the mean negative log-likelihoods for each monkey across all tested block instances (five-fold cross-validation), rounded to the nearest hundredth. Asterisks indicate the model with the lowest value of negative log-likelihood for each task. The initial and the number in parenthesis indicate a unique identifier for each monkey, as shown in **Supplementary Fig. S1**.

Supplementary Note 3

Our model fitting relies on maximum likelihood estimation (MLE), a method that maximizes the likelihood function ($\mathcal{L}(\theta_i; \mathbf{y}) = P(\mathbf{y}|\theta_i)$), ensuring that the observed data (\mathbf{y}) is the most likely outcome under the model defined by the parameters (θ_i). In our case, the data consist of the animals' choices on each trial of the experiment ($C(t)$, where $C(t) = 1$ and 0 for left and right choices), leading to the optimization of the following expression for each block of 80 trials:

$$\ell = \ln \mathcal{L}(\theta_i; \mathbf{y}) = \ln [P(\mathbf{y}|\theta_i)] = \ln \left[\prod_{t=1}^{80} \{P_{Left}(t) C(t) + (1 - P_{Left}(t))(1 - C(t))\} \right] \quad (Eq. S1)$$

where $P_{Left}(t)$ is the probability of choosing left as predicted by the model (see Eq. 5 in the Methods), and the logarithm of the likelihood function ($\ell = \ln \mathcal{L}$, referred to as the log-likelihood) is used to perform the optimization more conveniently.

We then used cross-validated negative log-likelihoods ($-LL$) of the alternative models to identify the best-fitting model, where a smaller negative LL indicates a better fit. Notably, even a small improvement in negative LL reflects a significant increase in the likelihood that the superior model better explains the observed choice behavior. To illustrate this, consider an example comparing the negative log-likelihoods of two models, i and j ($-LL_i$ and $-LL_j$):

$$\Delta LL = -LL_i - (-LL_j) \quad (\text{Eq. S2})$$

where LL_i is the mean log-likelihood of model i across the N test blocks:

$$LL_i = \frac{1}{N} \sum_{b=1}^N \ell(y_b|\theta_i) \quad (\text{Eq. S3})$$

where $\ell(y_b|\theta_i)$ represents the optimized log-likelihood of the model i for the data in block b (y_b), as described in Eq. S1 above. Noting that all blocks consist of 80 trials, it then follows:

$$\Delta LL = \frac{1}{N} \sum_{b=1}^N (\ell(y_b|\theta_j) - \ell(y_b|\theta_i)) = \frac{1}{N} \sum_{b=1}^N \left\{ \ln \left[\frac{P(y_b|\theta_j)}{P(y_b|\theta_i)} \right] \right\}, \quad (\text{Eq. S4})$$

and therefore,

$$N \times \Delta LL = \sum_{b=1}^N \left\{ \ln \left[\frac{P(y_b|\theta_j)}{P(y_b|\theta_i)} \right] \right\} = \ln \left[\prod_{b=1}^N \frac{P(y_b|\theta_j)}{P(y_b|\theta_i)} \right]. \quad (\text{Eq. S5})$$

By taking exponential on both sides, we obtain

$$e^{N \times \Delta LL} = \frac{\prod_b P(y_b|\theta_j)}{\prod_b P(y_b|\theta_i)}, \quad (\text{Eq. S6})$$

which leads to the relationship between the likelihoods of the two models:

$$\prod_b P(y_b|\theta_j) = e^{N \times \Delta LL} \times \prod_b P(y_b|\theta_i). \quad (\text{Eq. S7})$$

Considering that the median number of tested blocks across subjects for a given cross-validation instance was 92 ($M \pm SD = 97.11 \pm 30.6$, with a minimum of 48, corresponding to 20% of the blocks), even a slight difference in the log-likelihood, such as $\Delta LL = 0.05$ (indicating the superiority of Model j), implies that the test data is approximately $e^{92 \times 0.05} = 99.48$ times more likely to have originated from Model j compared to Model i on average.

Reviewer #2 (Remarks to the Author):

The presented study aimed to elucidate the distinct contributions of the ventral striatum and the amygdala to learning from various sources of reward uncertainty, a crucial aspect of understanding how neural circuits coordinate to enable adaptive behavior.

The authors designed a novel task that manipulated reward probabilities and the stimulus feature associated with these probabilities. By examining the learning patterns of subjects with lesions to either the ventral striatum (VS) or the amygdala, they inferred the role of these circuits in learning from different sources of uncertainty. Computational models fitted to the animals' behavior revealed dissociable impairments following VS and amygdala lesions. Based on their results, the authors propose that the amygdala is causally involved in the process of arbitrating between internal models to guide animals' behavior by inferring appropriate reward contingencies.

In their task, where reward probability can be specified as a function of either stimulus identity or location, VS lesions result in a systematic bias toward attributing reward probability to stimulus location, regardless of the true task contingency. In contrast, amygdala lesions lead to an impairment in appropriately attributing reward probabilities to either task contingency. The dynamic arbitration model employed to analyze the data was well-suited to the research question and allowed the authors to reconcile previous conflicting findings regarding the amygdala's role in reward-based learning.

Overall, the results support the authors' conclusion that the amygdala plays a causal role in arbitrating which internal model reward fluctuations are assigned to, and using that to appropriately select the internal model to guide behavior. There is also a very clear bias toward action-based learning in VS lesion animal. In the case of amygdala lesions it appears that neither stimulus identity or location are correctly inferred as determining reward probabilities, whereas in VS lesion case reward probabilities are largely attributed to stimulus location. Could both these deficits can be interpreted as different types of arbitration errors as opposed to the interpretation that the arbitration process is intact after VS lesions?

[R2.1] We thank the reviewer for the overall positive evaluation of our work and for their valuable feedback. With regard to the alternative interpretation that VS-lesioned animals exhibit a different type of deficit in arbitration, there are multiple reasons why our results do not support this interpretation, as explained below.

It is true that, for both amygdala- and VS-lesioned monkeys, the effective arbitration weights (Ω) are biased toward action-based strategy (smaller values correspond to more action-based) compared to the control monkeys. Nevertheless, to determine the nature of this behavioral tendency, we examined parameters ρ and ω (specially, initial ω , ω_0) in our full model, as these key parameters together determine the total “effective” arbitration

weights (Ω ; **Eq. 12**). In other words, the lower values of Ω could be primarily driven either by lower ω or by lower ρ , corresponding to different mechanisms. Compared to ω , ρ is less directly linked to the arbitration process itself. Instead, it serves as a constant that measures the baseline (time-independent) ratio between the two types of value signals, assumed to be fixed for each subject, as it reflects the permanent effect of the lesion on the strength of the signals from the two learning systems.

Examining these two parameters reveals the difference between the two lesioned groups). Amygdala-lesioned monkeys have ρ comparable to those of controls (**Fig.4a–c**) but have greatly reduced initial ω (**Fig. 4e inset**). In contrast, VS-lesioned monkeys have greatly reduced ρ compared to controls and Amyg group (**Fig.4a–c**) but less significant effect on initial ω . In the simulations results with $\rho = 0.4$ (approximating VS-lesioned group; **Fig. 4f**), it is shown that lower ρ value alone is sufficient to induce the biased update toward action-based system largely regardless of the initial ω values ($\psi_+ - \psi_-$ in blue colors, for $\omega_0 < \sim 0.55$; **Fig. 4f**, top panel). Another way to interpret this is that, even if we use the arbitration weight parameter ω_0 of the control monkeys ($\omega_0 \approx 0.4$), a system with $\rho = 0.4$ will still show a general bias toward the action-based system as arbitration process cannot reverse the impact of a smaller stimulus-based signal. Therefore, these results led us to conclude that the observed “deficits” for VS-lesioned monkeys can be mainly explained by the reduction in the baseline stimulus-value signals (lower ρ), rather than arbitration itself.

Nonetheless, to incorporate the reviewer’s feedback and avoid future confusion, we now have streamlined the above explanation by adding the following sentence in the **Results** (Line #516):

“It is worth noting that the simulations using $\rho = 0.4$ (**Fig. 4f, i**), which mimics the reduction in the relative baseline strength of stimulus-value signals due to VS lesions, result in the biased update rates toward action-based system for many of the ω_0 values (blue lines in **Fig. 4f**), including $\omega_0 \sim 0.37$, which matches the initial values for control monkeys. Therefore, the consistent adoption of action-based strategy in the What/Where task can be sufficiently accounted by reduced ρ value, without the need for additional constraints on ω_0 . These results support the notion that the impairments observed in VS-lesioned monkeys during this task can be fully explained by a reduction in stimulus-value signals, with minimal involvement of deficits in arbitration processes.”

Figure 4. Comparison of relative weighting and sensitivity of choice to stimulus- and action-based signals across control and lesion groups, and their simulations. (a) Plots show the mean and individual values (each point represents a monkey) of the relative strength of two systems on choice (ρ), separately for each group and task. **(b, c)** Comparison of sensitivity of choice to signals from the two systems, separately for each group and task. β_1 is the common sensitivity of choice (inverse temperature) estimated for each session and ρ is the relative strength of the two systems as in panel a. Insets show violin plots of the paired differences ($\Delta\beta = \beta_{\text{stim}} - \beta_{\text{action}}$), and asterisks indicate significant group effects as determined by mixed-effects analysis ($p < .05$). Only VS-lesioned monkeys showed larger sensitivity to action- than stimulus-based systems during both tasks, consistent with the role of the VS in stimulus-based learning. **(d–f)** Plots show simulated trajectory of ω and the difference in the effective arbitration rates ($\psi_+ - \psi_-$). Each line represents averaged trajectories (10,000 simulated blocks) of ω with different initial values (ω_0) and specified values of ρ and β_1 during the What task. All other parameters are fixed ($\alpha_+ = \alpha_- = 0.5$, $\beta_0 = 0$, $\zeta = 0.3$, $\alpha_\omega = 0.2$, $\zeta_\omega = 0.05$). Black, red, and gray arrows show the trajectory simulated with ω_0 equal to the mean of control, amygdala-lesioned, and VS-lesioned monkeys, showing $\psi_+ > \psi_-$, $\psi_+ \approx \psi_-$, and $\psi_+ < \psi_-$, respectively. Horizontal line (trial 40) indicates reversal. **Inset in panel e** shows distributions of ω_0 across the three groups during the What/Where task and asterisks indicate significant group difference (mixed-effects analysis). **(g–i)** Plots show simulated trajectory of ω and performance, $P(\text{Better})$. Conventions are the same as in d–f.

Minor comments:

- It would be helpful to refer to specific sections and equations in the methods when providing additional details about the analyses.

[R2.2] Thank you for this helpful suggestion. In the main text, we now have added relevant equation numbers and subsection titles whenever we refer to the specific section of Methods. In addition, throughout the Methods section we have added appropriate equation numbers for clarity, for example whenever we refer to ERDS (Eq. 1), the effective arbitration rates (Eqs. 17–18), or any of the RL models described.

- Near equations 3 and 4, where the learning rate (α) and decay parameter are first introduced, it would be useful to specify how these parameter values were chosen. Presumably, these are distinct from the learning rate and decay parameters for the dynamic weighting model presented in equation 9.

[R2.3] Yes, the reviewer is correct. We tried to clarify this by adding a note after Eq. 9 that those refer to different free parameters. Also in relation to the next comment (R2.4), we have added a **Supplementary Table 1** (provided below) which specifies all model parameters and their fitting ranges.

Model	Model description (# of parameters)	Description of parameters and their fitting ranges
RL _{Stim-only}	Single-system RL model that learns stimulus values only. (5)	(1) α_+ : learning rate for rewarded trials, [0 1] (2) α_- : learning rate for unrewarded trials, [0 1] (3) β_0 : side bias, positive if preferring right option, [-1 1] (4) β_T : inverse temperature, [1 100] (5) ζ : decay or forgetting rate for the unchosen option, [0 1]
RL _{Action-only}	Single-system RL model that learns action values only. (5)	Parameters (1)–(5): same as the first model.
Static ω (RL _{Stim+Act})	Two-system model with a fixed weighting between stimulus and action values. (6)	Parameters (1)–(5): same as the first model. (6) ω_{static} : fixed arbitration weight
Dynamic ω : RPE	Two-system model with dynamic weighting, with reliability based on RPE . (8)	Parameters (1)–(5): same as the first model. (6) ω_0 : initial arbitration weight, [0 1] (7) α_ω : update rate for arbitration weight, [0 1] (8) ζ_ω : passive decay rate for arbitration weight, [0 1]
Dynamic ω : V_{cho}	Two-system model with dynamic weighting, with reliability based on V_{Chosen} . (8)	Parameters (1)–(5): same as the first model. (6) ω_0 : initial arbitration weight, [0 1] (7) α_ω : update rate for arbitration weight, [0 1] (8) ζ_ω : passive decay rate for arbitration weight, [0 1]
Dynamic ω - ρ (V_{cho})	Two-system models with dynamic weighting (with reliability based on V_{Chosen}) and separate baseline signals for stimulus- and action-based learning. (9)	Parameters (1)–(5): same as the first model. (6) ω_0 : initial arbitration weight, [0 1] (7) α_ω : update rate for arbitration weight, [0 1] (8) ζ_ω : passive decay rate for arbitration weight, [0 1] (9) ρ : a hyperparameter, estimated for each subject, measuring the baseline ratio of signals from the stimulus-based to that from the action-based system, [0 1]

Supplementary Table 1. Reinforcement learning models used to fit choice data, description of parameters, and their fitting ranges. Each row provides a short description of a given RL model, its parameters, and the range of values used for fitting to each session. Note that the first five parameters are common across all models.

- In the methods section describing each model, it would be helpful to explicitly specify all the free parameters, which parameters were fit, and which were hyperparameters.

[R2.4] We appreciate the reviewer for this valuable suggestion. We have now added **Supplementary Table 1** (provided above) which offers a brief description of the RL models, their parameters, and their range of values used for fitting.

- I found the analyses presented in Fig 3e-g somewhat challenging to follow, particularly in terms of understanding the key insights they were intended to convey beyond the results already presented in Fig 3b-c. Additionally, I had difficulty locating the explanation for how the baseline for these analyses was determined within the methods section.

[R2.5] We thank the reviewer for pointing out the lack of clarity for the analyses on the effective arbitration rates. As hinted in the main text, this quantity is analogous to the learning rate for updating the value estimates of the choice options based on each learning strategy (Eq. 3). The key difference is that while the learning rate determines the rate of update in the value of choice options in response to reward feedback (within each valuation system), the arbitration rate determines the rate of update for arbitration weight (Ω) in response to the reliability difference (ΔReI) (within the arbitration module). The motivation behind this analysis was that, analogous to many studies that analyze learning rates to examine the dynamics of value adjustments in response to rewarded and unrewarded trials, the effective arbitration rate could similarly offer insight into the dynamics of arbitration under different conditions. For example, many studies have reported higher learning rates for rewarded trials than for unrewarded trials (i.e., $\alpha_+ > \alpha_-$), a phenomenon referred to as “positivity bias” (Palminteri & Lebreton, 2022 *Trends*). Furthermore, theoretical studies have suggested that this bias can help maximize overall reward (Cazé & van der Meer, 2013 *Biol. Cybern.*; Lefebvre et al., 2022 *Neural Comput.*). In our case, we sought to examine if the effective arbitration rate for the correct learning strategy in a given block environment (i.e., ψ_+ in What blocks and ψ_- in Where blocks) is larger than the rate for incorrect learning strategy. This bias would reduce noise in incorporating feedback, enabling more efficient arbitration and ultimately leading to improved performance.

With regard to the baseline of this analysis, the control monkeys serve as the reference point, which as predicted, exhibit larger ψ_+ than ψ_- during What blocks and larger ψ_- than ψ_+ during Where blocks. The comparison of these two quantities also serve as the basis for the simulations results in **Fig. 4**, where we aim to capture the lesioned animal’s pattern with two key parameters, ρ and ω_0 . (Please see our response in **R2.1** above.)

To clarify this point in more detail, we have updated the **Methods** subsection on *Effective arbitration rates*. In particular, we have added the following equation and additional explanations to the **Methods** (Line #994) to highlight the parallel between the learning rates (α_+ and α_- ; Eq. 3) and the effective arbitration rate (ψ_+ and ψ_-):

“To capture the overall update rates in arbitration weight Ω , we defined the “effective” arbitration rates that quantify the overall rate of change toward either the stimulus- or action-based system (Fig. 3d). Analogous to the update rule in the valuation system (Eq. 3), an update rule for the effective arbitration weight can be written as follows:

$$\Omega(t + 1) = \begin{cases} \Omega(t) + \psi_+(t)(1 - \Omega(t)) & \text{if } \Delta\Omega(t) > 0, \\ \Omega(t) + \psi_-(t)(0 - \Omega(t)) & \text{if } \Delta\Omega(t) < 0. \end{cases} \quad (\text{Eq. 16})$$

where ψ_+ and ψ_- represent the effective arbitration rate toward the stimulus-based and action-based systems, respectively ($\Delta\Omega = 0$ corresponds to no change in arbitration). Unlike the learning rates (α_+ and α_-) that are fitted to each session, the effective arbitration rates (ψ) are estimated for each trial (**Fig. 3e–g**). More specifically, the effective arbitration rate on trial t when Ω increases, shifting choice toward the stimulus-based system, is defined as follows:

$$\psi_+(t) = \frac{[\Delta\Omega(t)]^+}{1 - \Omega(t)}, \quad (\text{Eq. 17})$$

whereas the effective arbitration rate when Ω decreases, biasing choice toward the action-based system, is defined as follows:

$$\psi_-(t) = \frac{[\Delta\Omega(t)]^-}{0 - \Omega(t)}, \quad (\text{Eq. 18})$$

Similar to the learning rates for rewarded and unrewarded trials (α_+ and α_-), which capture differential updates based on different reward outcomes, the effective arbitration rates capture differential update rates toward the correct and incorrect learning systems based on the difference in reliability. Specifically, we tested whether the effective arbitration rate of the correct system in a given block type (i.e., ψ_+ in What blocks and ψ_- in Where blocks) is distinct from that of the incorrect system, as larger effective rates for the correct system would reduce noise in incorporating feedback, enabling more efficient arbitration and, ultimately, leading to improved performance.”

Reviewer #3 (Remarks to the Author):

This study examines the effect of lesions of ventral striatum (VS) and amygdala in monkeys performing a decision task in which there are blocks in which the reward probability is determined by stimulus identity ‘what blocks’ and blocks where the reward probability is determined by stimulus location on the screen ‘where blocks’. The task is interesting because it requires the subjects to dynamically arbitrate between two different possible strategies (stimulus-based vs action based) when assigning credit to guide subsequent choices. On the basis of the pattern of deficits observed in the lesion groups, combined with computational modelling, the authors argue that VS lesions impair stimulus-based learning hence biasing the animals towards an action-based strategy, whereas amygdala lesions impair both stimulus and action-based learning, but critically also affect the arbitration processes that determine which strategy is used.

The experiment and computational modelling are well motivated and well designed, and the study has potential to be an interesting and valuable contribution to the field. However, there appears to be a major issue with how the statistics have been done which will cause a high risk of false positive findings.

We thank the reviewer for their positive evaluation of our study, for highlighting important issues related to the use of statistics, and for their valuable suggestions for mitigating those issues. We hope that our response below, along with the inclusion of additional statistical tests, addresses their concerns.

The experiment design is inherently cross-subject as the authors are comparing the effect of lesions between different groups of animals. However, the authors appear not to have used subject as the experimental unit for statistical analyses, but rather have used within-subject observations as the statistical unit. I am not 100% sure whether the experimental unit was blocks or trials, but the number of experimental units used for statistical comparisons between groups was much larger than the number of subjects, as seen by the degrees of freedom reported, e.g. line 156: “amygdala-lesioned monkeys performed significantly better than VS-lesioned monkeys (two-sample t-test: $t(1832) = 17.72$, $p = 6.05 \times 10^{-65}$)”.

Using repeated within-subject observations as the experimental unit for statistical comparison in a between-subject experimental design massively inflates the risk of false positive findings of difference between the groups. This is because the hypothesis being tested is whether the observations from the two groups come from the same distribution, which they necessarily do not because the groups contain different subjects, and there is inevitably cross-subject variability. I have created an IPython notebook which demonstrates the issue using simulated data, which can be viewed here: <https://notebooksharing.space/view/56f8440250013f8bd6f5ef38aba12054e690b92dad5419855c44bbca23966a8>. Another way of stating the issue is that statistical tests typically assume that observations are independent and identically distributed (IID), but this

assumption is not met when working with repeated observations from multiple subjects, due to cross-subject variability.

It is essential to fix these statistical issues before the scientific content of the paper can be evaluated. Please can the authors:

- Use appropriate statistical tests which take cross-subject variability into account when testing for differences between the lesion groups. The simplest approach would be to average observations within subject to obtain a single number per subject, then test for significance between groups using standard statistical tests with subject as the experimental unit. More power might be obtained by using mixed-effects regression analyses, to enable observation-level data to be compared between groups while appropriately taking between-subject variance into account. Permutation tests where subjects are permuted between groups can also be useful in situations such as testing for a difference in parameters of RL model fits, where it may be necessary to combine data from multiple subjects to accurately fit model parameters.

[R3.1] We thank the reviewer for pointing out these important issues regarding the use of statistics and for their valuable suggestions. We agree that, in the original manuscript, we did not use the most appropriate statistical tests to account for subject variability, which may have led to inflated false positive results.

In terms of statistical units, the degrees of freedom reported previously referred to the block data, which was the main unit of analysis. Following the reviewer's suggestions, we have now primarily employed mixed-effects analysis with subjects (monkeys) as random effects to account for cross-subject variability, as each animal performed a large number of blocks across multiple session days. As a result, we have revised the entire manuscript based on the new statistical approach. Below, we outline the changes resulting from the use of these new statistical tests. Overall, while some previously significant results are no longer statistically significant with the revised method, the results supporting the main claims of the study remain qualitatively the same.

Changes in the **Methods** section (*Data analysis and statistical tests*; Line #1080):

"All analyses were carried out using MATLAB (MathWorks). All comparisons were performed using appropriate statistical tests reported throughout the text. For each test, we report the exact p-values and effect sizes when appropriate. All statistical tests used in this study were two-sided unless otherwise noted explicitly. In cases where highly significant results caused the software to return p-values of zero due to numerical precision limits, we report the smallest representable floating-point number at machine precision (i.e., 4.94×10^{-324}).

Because the data were collected across different animals, we primarily utilized linear mixed-effects regression analyses for between-group comparisons, with group assignment as a fixed effect and subjects as a random effect, to appropriately account for between-subject variance. When performing a within-group significance test to compare a paired set of samples (e.g., $\Delta\beta = \beta_{\text{stim}} - \beta_{\text{action}}$, $\Delta\psi = \psi_+$

– ψ), we fitted mixed-effects models with an intercept along and random effects of subjects. We then tested whether this intercept significantly differed from zero, as indicated by the coefficient β_0 . For comparing block-wise performance within each group, we included random effects of subjects and fixed effects of block types (*What-only*, *What*, *Where*) and reward uncertainty (measured as the variance of the variance outcome¹³), and their interaction. Specifically, the variance was calculated as $p_B^*(1-p_B)$, where p_B represents the reward probability of the better option (i.e., $p_B = \{0.8, 0.7, 0.6\}$).

In the **Results** section, we have updated all previous statistical tests accordingly. We have confirmed that while some of the previously significant results are no longer significant with the revised statistical methods, the results supporting the main claims of the study remain qualitatively unchanged. For more details, please see the below revisions to the following subsections under the **Results**:

Evidence for multiple learning systems and their interaction: This subsection mainly compares the block-wise performance across different block types in control monkeys. While the p-values have changed, the significance remains the same. Please note the changes in blue below (Line #117):

“Using mixed-effects analysis which accounts for subject variability, we observed that the probability of choosing the more rewarding option, $P(\text{Better})$, during the *What-only* task was significantly higher than in either the *What* blocks (main effects of block type; $\beta_{\text{What}} = -0.232$, $p = 1.14 \times 10^{-24}$) or the *Where* blocks ($\beta_{\text{Where}} = -0.212$, $p = 1.26 \times 10^{-20}$) of the *What/Where* task, which involved additional uncertainty about the correct model of the environment. We also found that expected uncertainty affects performance, measured by $P(\text{Better})$: across all block types and tasks, performance improved as it became easier to discriminate between the reward probabilities of the two options (main effects of reward variance; $\beta_{\text{var}(P)} = -2.43$, $p = 1.10 \times 10^{-169}$).

Influences of amygdala and ventral striatum lesions on learning and choice behavior: this subsection mainly compares the block-wise performance across block types and between groups of monkeys. Here, the significance also remains qualitatively the same. Please note the changes in blue below (Line #160):

“During the *What-only* task, amygdala-lesioned monkeys exhibited the largest impairment in performance ($P(\text{Better})$) across all three reward schedules: $M \pm SD = 0.581 \pm 0.11$; main effect of group in mixed-effects analysis; $\beta_{\text{VS}} = 0.054$, $p = .0061$; $\beta_{\text{control}} = 0.173$, $p = 3.25 \times 10^{-21}$; **Fig. 1e**). VS-lesioned monkeys also showed impairment compared to the control monkeys (main effect of group in mixed-effects analysis; $\beta_{\text{VS}} = -0.120$, $p = 1.97 \times 10^{-12}$; **Fig. 1e**). Although previous studies¹⁷ have reported additional differences in the behavior of amygdala- and VS-lesioned monkeys, the better performance of VS- compared to amygdala-lesioned monkeys (**Fig. 1e** inset) is surprising, given the established role of VS for stimulus-based learning^{26,27}, which is required for the *What-only* task.

During the *What* blocks of the *What/Where* task, however, amygdala-lesioned monkeys performed significantly better than VS-lesioned monkeys (mixed-effects analysis with group \times block type; $\beta_{\text{amyg:What}} = 0.0855$, $p = 1.67 \times 10^{-12}$; **Fig. 1f** inset), while both groups were impaired relative to control monkeys (amygdala: $\beta_{\text{amyg:What}} = -0.0679$, $p = 3.48 \times 10^{-52}$; VS: $\beta_{\text{VS:What}} = -0.180$, $p = 7.94 \times 10^{-7}$). In the *Where* blocks that did not require stimulus-based learning, only

amygdala-lesioned monkeys showed impairments in performance relative to controls ($\beta_{\text{amyg:Where}} = -0.0212$, $p = 4.48 \times 10^{-6}$; **Fig. 1g** inset). VS-lesioned performance was comparable to that of controls ($\beta_{\text{VS}} = -0.00871$, $p = .811$) and better than that of amygdala-lesioned monkeys ($\beta_{\text{amyg:Where}} = -0.0361$, $p = .00298$), suggesting no deficits in action-based learning in VS-lesioned monkeys.”

Deficits in arbitration due to amygdala but not VS lesions: this subsection mainly compares the difference in between the two effective arbitration rates ($\psi_+ - \psi_-$). Here, we note that some of the significance changed by using the mixed-effect analysis. Most notably, the difference in two arbitration rates in the amygdala group is now no longer significant for both block types of the What/Where task, which more strongly supports our hypothesis about the amygdala’s role in identifying and/or retaining the correct model of the environment, or biasing arbitration toward it. Please note the changes in blue below (Line #411):

“When the stimulus-based system was more reliable, the effective arbitration rates toward the stimulus-based system were larger in control monkeys (mixed-effects model with a single intercept; What-only task: $\beta_0 = 0.0940$, $p = 1.55 \times 10^{-269}$; What block of What/Where task: $\beta_0 = 0.0395$, $p = 1.47 \times 10^{-9}$; **Fig. 3e, f**). Similarly, in the Where blocks, where the action-based system was more reliable, the effective arbitration rate toward the action-based system was significantly larger than that toward the stimulus-based system ($\beta_0 = -0.0641$, $p = 4.94 \times 10^{-324}$; **Fig. 3g**).

In contrast, amygdala-lesioned monkeys showed the minimum differentiation between adjustments toward the more and less reliable (correct and incorrect) systems. Notably, during the What/Where task, amygdala-lesioned monkeys exhibited no significant difference between two arbitration rates during either block type (mixed-effects analysis on $\Delta\psi$ with a single intercept; What: $\beta_0 = 0.0149$, $p = .140$; Where: $\beta_0 = 0.00225$, $p = .823$; **Fig. 3f, g**). In contrast, VS-lesioned monkeys exhibited an overall large bias in arbitration rates toward the action-based system (i.e., higher ψ_-) during both block types (mixed-effects analysis on $\Delta\psi$ with a single intercept; What: $\beta_0 = -0.060$, $p = 2.33 \times 10^{-20}$; Where: $\beta_0 = -0.0684$, $p = 8.41 \times 10^{-26}$; **Fig. 3f, g**).

In the What-only task, with reduced uncertainty about the model of the environment, the difference between the two arbitration rates in amygdala-lesioned monkeys was positive ($\beta_0 = 0.0146$, $p = .00257$; **Fig. 3e**) but much smaller than that of control monkeys ($\beta_{\text{Control}} = 0.0796$, $p = 8.95 \times 10^{-32}$). VS-lesioned group also exhibited higher arbitration rates toward the stimulus-based system ($\beta_0 = 0.0186$, $p = .0143$), which aligns with the recovered performance observed in these monkeys.

Together, these results suggest that amygdala lesions impair arbitration between the two learning systems by eliminating differential updates for the correct and incorrect (more and less reliable) systems. This indicates that amygdala is critical for identifying and/or retaining the correct model of the environment, or biasing arbitration toward it. In contrast, VS lesions mainly impair stimulus-based learning and increase the overall arbitration bias toward action-based learning.”

Dynamic interaction between learning and arbitration processes and the impact of initial state: Here, we have also replaced most of the previous statistical tests (t-test and

rank sum test) with mixed-effects analysis with subjects as random effects, accounting for variability between subjects. In particular, the inset in **Fig. 4e** (attached below), showing the distributions of initial arbitration weight (ω_0), now reflects non-significant difference between control and VS-lesioned monkeys using the mixed-effects model (fixed effect of group and random effect of subjects). We have also added a permutation test comparing the model-estimated ρ values between controls and lesioned groups, which was previously missing. Please note the changes in blue below (Line #440):

“Considering the observed effects of amygdala and VS lesions on arbitration dynamics, we next examined the estimated parameters from the best-fit model (Dynamic ω - ρ). In this model, ρ captures whether there is an overall reduction in baseline value signals from the stimulus-based system relative to the action-based system. Consistent with the hypothesized role of VS in stimulus learning, we found that the estimated values of ρ were on average smaller in VS-lesioned monkeys compared to controls (permutation test for difference in group mean; $p = .0044$), indicating a larger baseline reduction in stimulus-value signals relative to action-value signals in VS-lesioned monkeys (**Fig. 4a**). In contrast, we found no such evidence for reduction in ρ in amygdala-lesioned monkeys relative to controls (permutation test for difference in group mean; $p = .869$). As a result, VS-lesioned monkeys exhibited smaller difference in choice sensitivity to stimulus- and action-value signals ($\Delta\beta = \beta_{\text{stim}} - \beta_{\text{action}}$), biased toward action-value signals, compared to controls in both the What-only task (mixed-effects analysis on $\Delta\beta$, main effect of group; $\beta_{\text{VS}} = -7.07$, $p = .0114$; **Fig. 4b** inset) and the What/Where task ($\beta_{\text{VS}} = -3.85$, $p = .0115$; **Fig. 4c** inset). Interestingly, this was not the case for amygdala-lesioned monkeys, which exhibited no significant difference compared to controls in either task (What-only: $\beta_{\text{amyg}} = -2.84$, $p = .224$; **Fig. 4b** inset; $\beta_{\text{amyg}} = 0.804$, $p = .643$; **Fig. 4c** inset). Within-group comparison of β_{stim} and β_{action} also reflected this pattern, as VS-lesioned monkeys tended to exhibit larger sensitivity of choice to action-based signals (β_{action}) than to stimulus-based signals (β_{stim}) during both tasks (mixed-effects analysis on $\Delta\beta$ with a single intercept; What-only: $\beta_0 = -2.80$, $p = .0540$; What/Where: $\beta_0 = -1.470$, $p = .000502$; **Fig. 4b, c**). In comparison, amygdala-lesioned monkeys tended to show larger β_{stim} than β_{action} in both tasks (mixed-effects analysis on $\Delta\beta$ with a single intercept; What-only: $\beta_0 = 2.796$, $p = .116$; What/Where: $\beta_0 = 2.804$, $p = .0480$; **Fig. 4b, c**). This was also the case for control monkeys (mixed-effects analysis; What-only: $\beta_0 = 3.532$, $p = 4.87 \times 10^{-5}$; What/Where: $\beta_0 = 2.80$, $p = .000809$; **Fig. 4b, c**). These results suggest that unlike VS lesions, amygdala lesions did not affect the relative strength of stimulus-value vs. action-value signals. Therefore, consistent with previous observations, the deficit observed in amygdala-lesioned monkeys cannot be solely attributed to impairments in the stimulus-based learning. Instead, they suggest deficits in arbitration processes that subsequently affect learning and decision-making processes.

To confirm this point, we examined the initial arbitration weights that determine the weights of the two systems on choice at the beginning of each block, when the monkeys are oblivious to the correct model of the environment during the What/Where task. We note that amygdala- and VS-lesioned monkeys did not significantly differ in the initial Ω (effective ω) values (mixed-effects analysis on Ω_0 , main effect of group; $\beta_{\text{amyg}} = 0.021$, $p = .399$; compare Ω of the first trial in **Fig. 3b** and **Fig. 3c**). However, by examining the initial arbitration weights (ω_0) before scaling by ρ , we found that amygdala-lesioned monkeys had significantly smaller ω_0 values compared to both VS-lesioned monkeys (mixed-effects analysis; $\beta_{\text{VS}} = 0.142$, $p = .0332$; **Fig. 4e** inset) and control monkeys ($\beta_{\text{control}} = 0.191$, $p = .0126$). This means that larger values of ρ in amygdala-lesioned monkeys were offset by lower ω_0 values to yield Ω_0 comparable to VS-

lesioned monkeys. Therefore, deficits due to amygdala lesions can be **mainly attributed to the reduction in the initial weight (ω_0)** and subsequent interaction between arbitration and learning processes, whereas deficits due to VS lesions are largely caused by reduction in the relative baseline strength of stimulus-value to action-value signals, measured by ρ .

To further **validate this idea through model simulations**, we **generated** the choice behavior of the Dynamic ω - ρ model by adjusting two key parameters **to mimic the effects of brain lesions**: baseline ratio of **the weights of the stimulus- to action-value signals (ρ)** and the initial arbitration weight (ω_0). These two parameters reflected the most consistent effect of lesions across the two tasks, with reduced ρ in VS-lesioned monkeys (**Fig.4a–c**; $\beta_{VS} = -195$, $\rho = .00279$; mixed-effects analysis on compiled ρ across all groups/tasks) and reduced ω_0 in amygdala-lesioned monkeys ($\beta_{amyg} = -0.191$, $\rho = .0171$; mixed-effects analysis on compiled ω_0 across groups/tasks)”

Figure 4. Comparison of relative weighting and sensitivity of choice to stimulus- and action-based signals across control and lesion groups, and their simulations. (a) Plots show the mean and individual values (each point represents a monkey) of the relative strength of two systems on choice (ρ), separately for each group and task. **(b–c)** Comparison of sensitivity of choice to signals in the two systems, separately for each group and task. β_1 is the common sensitivity of choice (inverse temperature) estimated for each session and ρ is the relative strength of the two systems as in panel a. Insets show violin plots of the paired differences ($\Delta\beta = \beta_{stim} - \beta_{action}$), and **asterisks indicate significant group effects as determined by mixed-effects analysis ($p < .05$)**. Only VS-lesioned monkeys showed larger sensitivity to action- than stimulus-based systems during both tasks, consistent with the role of the VS in stimulus-based learning. **(d–f)** Plots show simulated trajectory of ω and the difference in the effective arbitration rates ($\psi_+ - \psi_-$). Each line represents averaged trajectories (10,000 simulated blocks) of ω with different initial values (ω_0) and specified values of ρ and β_1 during the What task. All other parameters are fixed ($\alpha_+ = \alpha_- = 0.5$, $\beta_0 = 0$, $\zeta = 0.3$, $\alpha_\omega = 0.2$, $\zeta_\omega = 0.05$). Black, red, and gray arrows show the trajectory

simulated with ω_0 equal to the mean of control, amygdala-lesioned, and VS-lesioned monkeys, showing $\psi_+ > \psi_-$, $\psi_+ \approx \psi_-$, and $\psi_+ < \psi_-$, respectively. Horizontal line (trial 40) indicates reversal. Inset in panel e shows distributions of ω_0 across the three groups during the What/Where task and asterisks indicate significant group difference (mixed-effects analysis). (g–i) Plots show simulated trajectory of ω and performance, $P(\text{Better})$. Conventions are the same as in d–f.

Contribution of amygdala to long-term adjustments of behavior. The significance tests for these analyses examining long-term adjustment in behavior (**Fig. S8**, attached below) has been updated with mixed-effects models which include random effects of the subject. Instead of simply fitting a line across subjects, we have additionally considered random slopes of subjects for the main effect of the fraction of sessions completed, to further account for the variability in the long-term adjustment at the subject level. The previous significant effect on the controls for initial effective arbitration weight (Ω_0 ; **Fig. S8d** below) is now found to be insignificant with this updated regression model. The main text for this figure has been updated as follows (Line #573):

“For consistent use of stimulus-based strategy, we observed a significant decrease in $ERDS_{\text{Stim}}$ in VS-lesioned monkeys (VS: $\beta_{\text{session}(\%)} = -0.207$, $p = 1.03 \times 10^{-5}$), whereas there was no evidence for changes in control or amygdala-lesioned monkeys across time (Control: $\beta_{\text{session}(\%)} = -0.0592$, $p = .285$; amygdala: $\beta_{\text{session}(\%)} = -0.0108$, $p = .802$; **Fig. S8a**). Specifically, despite their impaired stimulus-based learning, monkeys with VS lesions were able to increase their adoption of stimulus-based strategy over time. Consistently, VS-lesioned monkeys also decreased their adoption of action-based strategy as reflected in the positive slope of $ERDS_{\text{Action}}$ over time ($\beta_{\text{session}(\%)} = 0.0953$, $p = .0249$; **Fig. S8b**). There was no evidence of such an effect in control monkeys ($\beta_{\text{session}(\%)} = -0.0155$, $p = .428$) or in monkeys with amygdala lesions ($\beta_{\text{session}(\%)} = -0.0147$, $p = .414$). Interestingly, consistent with previous results, the complementary changes in model adoption in VS-lesioned monkeys were also reflected in increased median RT over time in these monkeys ($\beta_{\text{session}(\%)} = 12.523$, $p = 3.90 \times 10^{-4}$; **Fig. S8c**) but not in control monkeys ($\beta_{\text{session}(\%)} = 5.25$, $p = .276$) or amygdala-lesioned monkeys ($\beta_{\text{session}(\%)} = -7.64$, $p = .168$). This was accompanied by a long-term increase in the initial effective arbitration weights Ω_0 (toward stimulus-based system) **only for the VS-lesioned monkeys** ($\beta_{\text{session}(\%)} = 0.398$, $p = 1.43 \times 10^{-6}$; **Fig. S8d**).”

The **Methods** section explaining this analysis has been also adjusted as follows (Line #1110):

“To study the long-term adjustment in behavior across the time course of the experiment (Supplementary **Fig. S8**), we calculated $ERDS_{\text{Stim}}$, $ERDS_{\text{Action}}$ for each block (Eq. 1) and regressed them on the proportion of the sessions completed (%) as the predictor variable within each subject using mixed-effects models. To further account for the variability in the long-term adjustment at the subject level, we considered random slopes and intercepts for the effect of sessions within each subject. We report the regression coefficient and p-value for the main effect of the fraction of sessions completed (β_{sess} or $\beta_{\text{session}(\%)}$). For visualization purposes only, we plot the simple least-squares lines in **Fig. S8**.”

Figure S8. Contribution of the amygdala to behavioral adjustments over long timescales. (a–b) Time course of entropy of reward-dependent strategy on stimulus identity ($ERDS_{Stim}$, a) and performed action ($ERDS_{Action}$, b) for controls (black), amygdala- (red), and VS-lesioned (blue) monkeys during the What-only task. Number of blocks completed was normalized by each monkey into percentages (error bars indicate s.e.m. across subjects). Straight lines indicate least-squares lines regressing ERDS on the fraction of sessions completed. Asterisks indicate significance of regression from mixed-effects analyses ($p < .05$). Sub-panels to the right show regression coefficients for the proportion of sessions completed (%) for each group (corresponding to the slope of the fitted lines but accounting for subject variability). (c–d) Time course of median RT (c) and initial effective arbitration weight Ω_0 (d) within each block for the entirety of the What-only task. Conventions are the same as in panels a–b.

In addition, we have significantly adjusted the statistical tests used for the analysis in the **Supplementary Note 1** (updated figure attached below). Please note that we are also reporting percentages of trials using mean and SEM across subjects in the plots. Because our previous two-sample tests (rank sum) did not address cross-subject variability and thus inflate the risk of getting false positive results, we instead adopted a mixed-effects analysis and tried to identify the unique effect of the dominant strategy from other potential confounds. Compared with the previous results from Wilcoxon rank sum test, we note that the significance effect has changed only for controls and amygdala groups in What-only task (**Fig. SN1a**: previously significant; **Fig. SN1d**: previously insignificant). Furthermore, the previous two-sample z-test for proportions (for comparing performance) have been replaced with a generalized mixed-effects model (random effects of subjects) with binomial distribution. Please see the updated **Supplementary Note 1** below (Line #1326):

“Supplementary Note 1

Considering previous findings on faster decision making during action-based compared with stimulus-based tasks²², we hypothesized that reaction time (RT) on a given trial depends on the learning system that controls the behavior more strongly on that trial. To test this hypothesis, we categorized trials as either stimulus- or action-dominant by directly comparing $ERDS_{Stim}$ and $ERDS_{Action}$ (computed

from a moving window of ten trials). Trials that had $ERDS_{Stim} < ERDS_{Action}$ were categorized as stimulus-dominant, and action-dominant if $ERDS_{Action} < ERDS_{Stim}$. To assess whether the dominant strategy had a significant effect on RT, we performed mixed-effects analysis and identified the unique effect of the dominant strategy from other potential confounds. Specifically, the mixed-effects model included random effect of subjects and the following predictors as fixed effects: dominant strategy, coded as stim-dominant (0) or action-dominant (1), whether the monkey has chosen the better option (1) or not (0), reward schedule or uncertainty, trial number within a block, block number within a session, and session number within the subject. We also included interaction between dominant strategy and choice of better option, as the latter could depend on the adopted strategy. Variables were normalized for each monkey to yield comparable standardized regression coefficients.

For the What-only task in control monkeys, the vast majority of the analyzed trials (79.4%) were classified as stimulus-dominant (**Fig. SN1a**). Interestingly, there was still a proportion of trials with $ERDS_{Action} < ERDS_{Stim}$ ($8.66\% \pm 1.25\%$) even though action-based value/strategy was irrelevant for performing the task. Therefore, although the What-only task lacks the task-imposed, objective uncertainty about the correct model of the environment, monkeys still considered this uncertainty and the alternative model of the environment. The mixed-effects analysis indicated that the dominant strategy was not a significant predictor of RT ($\beta_{Act-Dominant} = -0.0181$, $p = .279$; $M \pm SEM$ across subjects for stimulus-dominant RT: 223.8 ± 29.5 ; action-dominant RT: 225.2 ± 34.9). To compare the performance for these two types of trials, we ran a similar generalized mixed-effects model predicting animals' choice of better option (binomial distribution with logit link function), with the same random and fixed effects. Using this analysis, we found that action-dominant trials predicted significantly lower performance ($\beta_{Act-Dominant} = -0.708$, $p = 3.54 \times 10^{-238}$), as reflected in the P(Better) of action-dominant trials (0.621 ± 0.010) and stimulus-dominant (0.790 ± 0.016) trials. These results suggest that most action-dominant trials happened when action values were used instead of stimulus values to make decisions, resulting in more erroneous responses.

For the What/Where task, the proportions of stimulus-dominant and action-dominant trials were reflected in the respective block types: the What blocks were marked by a higher proportion of stimulus-dominant trials ($59.5\% \pm 5.99\%$; **Fig. SN1b**), whereas the majority of trials within the Where blocks were categorized as action-dominant ($63.6\% \pm 6.41\%$; **Fig. SN1c**). Critically, performance was higher for the correct strategy: in the What blocks, performance was significantly higher for stimulus-dominant (0.820 ± 0.035) than action-dominant (0.607 ± 0.019) trials ($\beta_{Act-Dominant} = -0.959$, $p = 4.94 \times 10^{-324}$), whereas in the Where blocks, P(Better) was higher for action-dominant (0.791 ± 0.025) than for stimulus-dominant (0.597 ± 0.018) trials ($\beta_{Act-Dominant} = 0.846$, $p = 4.94 \times 10^{-324}$). In terms of RT, we found that for both block types, action-dominant strategy predicted significantly shorter RT ($\beta_{Act-Dominant} = -0.0926$, $p = 1.38 \times 10^{-27}$). Consistently, categorization of trials based on comparison of ERDS (stimulus-dominant vs. action-dominant) yielded a larger distinction in RTs than categorization simply based on block type (What vs. Where), as reflected by the smaller proportion of variance explained by block type predictor (R^2 equal to 0.005 and 0.0081 for block type and dominant strategy; comparison of partial R^2 values). These results show that entropy-based metrics could be used to identify the adopted model on a given trial and that RT reflected the adopted strategy, with the stimulus-based strategy yielding consistently longer RT than the action-based strategy. This could be explained by the fact that the monkeys can prepare the left or right movement early on to make decisions based on actions, whereas a stimulus-based decision requires the animals to identify the target and then plan the movement only after the stimuli appear. We note that the results remain qualitatively the same when the analyses are restricted to the last 20 trials of each block, where performance has plateaued.

In brain-lesioned monkeys, the effect of the dominant strategy on RT was more consistent across two tasks, with the action-dominant strategy significantly predicting shorter RTs (**Fig. SNd-i**). Furthermore, the correct strategy for a given block type predicted significantly higher performance. That is, action-dominant strategy predicted lower performance during the What-only task (amygdala: $\beta_{\text{Act-dominant}} = -0.230$, $p = 1.35 \times 10^{-63}$; VS: $\beta_{\text{Act-dominant}} = -0.459$, $p = 1.65 \times 10^{-143}$) and the What blocks of What/Where task (amygdala: $\beta_{\text{Act-dominant}} = -0.665$, $p = 8.26 \times 10^{-308}$; VS: $\beta_{\text{Act-dominant}} = -0.320$, $p = 701 \times 10^{-52}$), whereas it predicted higher performance in the Where blocks of What/Where task (amygdala: $\beta_{\text{Act-dominant}} = 0.546$, $p = 1.01 \times 10^{-152}$; VS: $\beta_{\text{Act-dominant}} = 0.550$, $p = 1.93 \times 10^{-101}$). ”

Figure SN1. Distributions of reaction time (RT) for stimulus-dominant and action-dominant trials, separately for different tasks and block types. Each trial was categorized as either stimulus- or action-dominant by comparing $ERDS_{\text{Stim}}$ and $ERDS_{\text{Action}}$. Percentages indicate proportions of trials under each category, reported as mean \pm SEM across subjects (remaining percentages correspond to trials where both strategies were equally dominant). (a–c) RT data from control monkeys during the What-only (a) and the What/Where tasks (b,c). (d–f) RT data from amygdala-lesioned monkeys during the What-only (d) and the What/Where tasks (e,f). (g–i) RT data from VS-lesioned monkeys during the What-only (g) and What/Where tasks (h,i). Circles in the violin plots represent means and black horizontal lines represent medians of the distributions. Reported are regression coefficient and corresponding p-values for the main effect of action-dominant strategy, with negative (positive) values indicating that action-dominant trials predict shorter (longer) RT. In the What/Where task (b–c, e–f, h–i), RT was consistently shorter for action-dominant trials across all groups.

Lastly, we have updated the statistics in **Fig. S6** (attached below) using the updated mixed-effects model that includes random effects of subjects. Here, the results have not changed but only the beta coefficients and p-values have been adjusted. The main text referring to these results has also been updated as follows (Line #320):

“We found that in the single-system model that learned the stimulus-outcome contingencies only (RL_{Stim-Only}), ERDS_{Stim} was only weakly predictive of ERDS_{Action} (**Supplementary Fig. S6b**; $\beta = -0.006$, $p = 7.27 \times 10^{-45}$). In contrast, in both the static and dynamic two-system models, ERDS_{Stim} was negatively predictive of ERDS_{Action}, thus reproducing the competitive interaction between the two systems (**Supplementary Fig. S6c**; $\beta = -0.064$, $p = 2.7 \times 10^{-56}$; **Supplementary Fig. S6d**; $\beta = -0.088$, $p = 1.68 \times 10^{-56}$). These simulation results further support the presence of multiple learning systems and their dynamic interaction, even in an environment where one of the systems was not beneficial for performing the task.”

Figure S6. Comparison of empirical and simulated ERDS in control monkeys during the What-only task provides evidence for presence of multiple learning systems. (a) Scatter plot of entropy of reward-dependent strategy based on stimulus identity (ERDS_{Stim}, X-axis) vs. action (ERDS_{Action}, Y-axis) in control monkeys during the What-only task. Reported are regression coefficient (β) and its p-value for the regression of ERDS_{Action} on ERDS_{Stim}, using mixed-effects analysis with subjects as a random effect. Histograms in the X- and Y-axis show distributions of ERDS_{Stim} and ERDS_{Action}, respectively. (b) Simulated metrics using estimated parameters of the model with stimulus-learning system only (RL_{Stim-only}). (c) Simulated metrics using estimated parameters of the static two-system model (RL_{Stim+Action+Static} ω). (d) Simulated metrics using estimated parameters of the two-system model with dynamic adjustment using V_{chosen} as the reliability measure (Dynamic $\omega : V_{cho}$). The Dynamic ω model better captures the variability in ERDS_{Action} compared to the Static ω model shown in panel (c).

- When comparing data from different lesion groups in plots, ensure that the error bars or confidence regions use cross-subject variance, and report the experimental unit used in the figure legend (e.g. Error bars = SEM across subjects).

[R3.2] We appreciate this feedback and agree that plotting SEM across subjects provides a better visualization of cross-subject variability in the data. We have adjusted the figures that include any error bars (which were previously SEM across blocks) to reflect SEM across subjects:

1. **Fig. 1e-g:** performance is now visualised with mean and SEM across the subjects, as below:

Figure 1. (e–g) Time course of performance of the monkeys in each group, measured as the probability of choosing the better option, $P(\text{Better})$, separately for different tasks and block types. Insets show averaged performance by reward schedules. **Error bars = SEM across subjects.**

2. **Fig. 3e-g:** the mean and errorbar for effective arbitration rates are now computed across the subjects:

Figure 3. (e–g) Plotted is the time course of effective arbitration rates toward the stimulus-based or action-based system separately for each task (indicated on the top) and for each group of monkeys (controls: black; VS-lesioned: blue; amygdala-lesioned: red). Solid and dotted lines indicate effective arbitration rates toward stimulus-based (ψ_+) and action-based systems (ψ_-), respectively. Insets show the mean paired differences between the two arbitration rates within each block after reversal ($\Delta\psi = \psi_+ - \psi_-$). **Error bars = SEM across subjects.**

3. **Fig. S2:** this is a newly added supplementary figure for the distribution of ERDS by reward schedule, containing insets with error bars = SEM across subjects.

Figure S2. Distributions of conditional entropy of reward-dependent strategy (ERDS) by block type and reward schedule. Distributions of ERDS based on stimulus identity ($ERDS_{Stim}$, in solid lines) or action ($ERDS_{Action}$, in solid lines) computed from each block of trials during the What-only (first column) and the What/Where task (second and third columns). Insets represent mean values (Error bars = SEM across subjects). Asterisks indicate significant effect of reward uncertainty on ERDS, using mixed-effects analysis with subjects as a random effect (*: $p < .05$, **: $p < .01$, ***: $p < .001$). Reward uncertainty of each block was measured as the variance of outcome equal to $p_{Better} * (1 - p_{Better})$. (a–c) $ERDS_{Stim}$ (in solid lines) and $ERDS_{Action}$ (in dotted lines) in control monkeys during the What-only (a) and the What/Where tasks (b,c). Colors indicate reward schedules (black: 80/20; dark gray: 70/30, light gray: 60/40). (d–f) ERDS in amygdala-lesioned during the What-only (d) and the What/Where tasks (e,f). Colors indicate reward schedules (brown: 80/20; red: 70/30, orange: 60/40). (g–i) ERDS in VS-lesioned during the What-only (g) and the What/Where tasks (h,i). Colors indicate reward schedules (navy: 80/20; blue: 70/30, cyan: 60/40).

4. **Fig. S3:** this was previously Fig. S2, which is adjusted with means & SEM across subjects as below:

Figure S3. (d–f) Summary results for the panels in a–c. Plotted are mean values of $\Delta ERDS = ERDS_{Stim} - ERDS_{Action}$, by each group and reward schedule during What-only (d) and What/Where

(e,f) tasks. With larger reward uncertainty, animals' strategies became relatively more biased toward the incorrect strategy (increasing $\Delta ERDS$ in the What blocks and decreasing $\Delta ERDS$ in the Where block). Asterisks next to the plots indicate significant effect of reward uncertainty (variance) for the respective group indicated by colors, as determined by mixed-effects analysis (including reward variance as fixed effect and subjects as a random effect). (*: $p < .05$, **: $p < .01$, ***: $p < .001$). It is worth noting that, while $ERDS_{Action}$ was not explicitly modulated by reward uncertainty during the What-only task for any of the groups (**Supplementary Fig. S2a,d,g**), competitive interactions between $ERDS_{Stim}$ and $ERDS_{Action}$ still exist for the task. This suggests that $ERDS_{Stim}$ cannot be solely responsible for the observed effect in $\Delta ERDS$.

5. **Fig. S9, S10:** the significance tests on the distributions of the fitted parameters between groups, which previously used Wilcoxon rank sum test, have been replaced with mixed-effects analysis with subjects as the random effect. We now report the main effects of the group to indicate significance. The mean values on X-axis have been also updated to reflect mean and SEM across subjects. We note that the overall significance of the difference has been reduced compared to the previous results from rank sum tests.

6.

Figure S9. Distributions of estimated model parameters in What/Where task across three animal groups. Plotted are the distributions of estimated parameters of the best model (Dynamic ω - ρ with V_{cho}) fitted to the choice behaviors of controls (black), amygdala- (red), and VS-lesioned (blue) monkeys during the What/Where task. α_{+} : learning rate on rewarded trials (a). β_1 : common inverse temperature for stimulus- and action-based systems (b). β_0 : side bias term, positive if preferring right option (c). α_{-} : learning rate on unrewarded trials (d). ζ : decay or forgetting rate for the unchosen option (e). α_{ω} : arbitration transition rate (f). ω_0 : initial arbitration weight on the first trial of each block (g). ζ_{ω} : decay rate for arbitration weight ω toward initial value (h). Asterisks indicate significant effects of group difference (mixed-effects analysis with random effects of subjects; *: $p < .05$, **: $p < .01$, ***: $p < .001$). Circles in the violin plots indicate medians, and the numbers on the X-axis indicate mean \pm SEM across subjects. ρ parameter is assumed to be fixed for each subject and is shown in Fig. 4a.

Figure S10. Distributions of estimated model parameters in What-only task across three animal groups. Plotted are the distributions of estimated parameters of the best model (Dynamic ω - ρ with V_{cho}) fitted to the choice behaviors of controls (black), amygdala- (red), and VS-lesioned (blue) monkeys during the What-only task. α_+ : learning rate on rewarded trials (a). β_1 : common inverse temperature for stimulus- and action-based systems (b). β_0 : side bias term, positive if preferring right option (c). α_- : learning rate on unrewarded trials (d). ζ : decay or forgetting rate for the unchosen option (e). α_ω : arbitration transition rate (f). ω_0 : initial arbitration weight on the first trial of each block (g). ζ_ω : decay rate for arbitration weight ω toward initial value (h). Asterisks indicate significant effects of group difference (mixed-effects analysis with random effects of subjects; *, $p < .05$, **, $p < .01$, ***, $p < .001$). Circles in the violin plots indicate medians, and the numbers on the X-axis indicate mean \pm SEM, across subjects. ρ parameter is assumed to be fixed for each subject and is shown in Fig. 4a.

7. **Fig. S11:** Finally, the significant tests for comparing fitted parameters, which previously used Wilcoxon rank sum test, have been also replaced with mixed-effects analysis with subjects as the random effect.

Figure S11. Comparison between dynamic models with or without passive decay in arbitration weight, and estimated parameters in the better-fitting model. (a) Goodness of fit using five-fold cross-validation of dynamic models with or without passive decay for ω . Numbers in parenthesis indicate McFadden R² (Eq. 20) [R1.4]. Model with the additional passive decay mechanism better accounts for the choice behavior in all groups, especially during the What/Where task. (b) Distribution of estimated parameters for sessions in controls (black), amygdala (red), and VS (blue) groups during the What-only task, showing parameters for α_ω in the above model without passive decay in ω . Asterisks indicate significant effects of group difference (mixed-effects analysis with random effects of subjects; *, $p < .05$, **, $p < .01$, ***, $p < .001$). Circles in the violin plots indicate medians, and the numbers on the X-axis indicate mean \pm SEM, across subjects. (c) Same plots as in (b) but for the above model with passive decay in ω , showing parameters for α_ω (model arbitration/update rate, left) and ζ_ω (decay rate for ω , right). Arbitration rates α_ω tended to be larger than the passive decay rate ζ_ω in control group (mixed-effects analysis on $\alpha_\omega - \zeta_\omega$; controls: $\beta_0 = 0.140$, $p = .00312$; amygdala: $\beta_0 = 0.015$, $p = .824$; VS: $\beta_0 = -0.129$, $p = 2.76 \times 10^{-4}$). (d) Same plot as in (b) but for the What/Where task, in the model without passive decay in ω . (e) Same plots as in (c) but for the What/Where task, in the model with passive decay in ω . Arbitration rates α_ω were significantly larger than the passive decay rate ζ_ω across all groups, suggesting that α_ω is the primary source of the transitions in ω (mixed-effects analysis on $\alpha_\omega - \zeta_\omega$ with a single intercept; controls: $\beta_0 = 0.140$, $p = 1.04 \times 10^{-6}$; amygdala: $\beta_0 = 0.164$, $p = 7.89 \times 10^{-10}$; VS: $\beta_0 = 0.203$, $p = 1.09 \times 10^{-21}$).

- Also on the subject of stats, if the authors want to argue that there are interactions between lesion group and task condition (i.e. what-only, what, where) then they should directly test for significant interactions using e.g. ANOVA or regression analyses.

[R3.3] We thank the reviewer for this additional comment. Please note that we have included the interaction between group and block type (i.e. *What*, *Where*) when making between-group comparison of block-wise performance during What/Where tasks (Line #170; reproduced below from **[R3.1]**):

“During the What blocks of What/Where task, however, amygdala-lesioned monkeys performed significantly better than VS-lesioned monkeys (mixed-effects analysis with group \times block type; $\beta_{\text{amyg:What}} = 0.0855$, $p = 1.67 \times 10^{-12}$; **Fig. 1f** inset), while both groups were impaired relative to control monkeys (amygdala: $\beta_{\text{amyg:What}} = -0.0679$, $p = 3.48 \times 10^{-52}$; VS: $\beta_{\text{VS:What}} = -0.180$, $p = 7.94 \times 10^{-7}$). In the Where blocks that did not require stimulus-based learning, only amygdala-lesioned monkeys showed impairments in performance relative to controls ($\beta_{\text{amyg:Where}} = -0.0212$, $p = 4.48 \times 10^{-6}$; **Fig. 1g** inset). VS-lesioned performance was comparable to that of controls ($\beta_{\text{VS}} = -0.00871$, $p = .811$) and better than that of amygdala-lesioned monkeys ($\beta_{\text{amyg:Where}} = -0.0361$, $p = .00298$), suggesting no deficits in action-based learning in VS-lesioned monkeys.”

Moreover, please note that for determining the cross-task effect of brain lesions on the estimated model parameters, we included random effects of task condition instead of explicitly specifying its interaction with the group. More specifically, the model equation was written as follows: $parameter \sim group + (group|task) + (1|subject:task)$. Namely, the model included a fixed effect of group (*control*, *amygdala*, *VS*) and random effects of subjects and tasks (*What-only* or *What/Where*), along with a random slope of group for each task. This was done because specifying the interaction term between group and task condition would also pick up on significant group differences with the opposite direction of effect. For example, compared to controls, the learning rate on unrewarded trials was smaller for VS-lesioned monkeys during the What-only task but larger during the What/Where task. As our goal here was to identify model parameters that are significantly modulated in the lesioned group with the consistent direction of effects across both task conditions, we enforced the same sign for the main effects of the group by adopting the above model. This point has been explained in the **Methods** as follows (Line #1130):

“To identify the model parameters that are significantly modulated in the lesioned group with the consistent direction of effects across task conditions (*What-only* and *What/Where*), we adopted the following mixed-effects model: $parameter \sim group + (group|task) + (1|subject:task)$. Namely, the model included a fixed effect of group (*control*, *amygdala*, *VS*) and random effects of subjects and tasks with a random slope of each group in each task. This formulation assumes a single, uniform effect of the group regardless of task condition, thereby identifying parameters that are consistently modulated by brain lesions. Note that subjects were nested within tasks to allow for a separate baseline for each task, as some of the monkeys participated in both the What-only and What/Where tasks. We tested this mixed-effects model on each of the parameters compiled across groups and tasks, with control monkeys as the reference group.”

Other points:

- Please provide a diagram summarising the flow of subjects through the different stages of the experiment, as some but not all subjects did both ‘what-only’ and ‘what-where’

versions of the task and it would be useful to visually summarise the structure of the experiment to help the reader understand what was done in what order.

[R3.4] We thank the reviewer for this great suggestion. We have now added a summary diagram of all subjects in a new **Supplementary Fig. 1a** and have adjusted all the figure numbers accordingly. In this process, we also fixed an error in the Methods (Line #786) where we had meant to say that only *one* (not four) of the six controls in 2021 study was the same monkey used for the two previous experiments (Monkey #2 in the diagram). Accordingly, the total unique number of monkeys has been now adjusted to twenty (not seventeen).

Figure S1. Summary of monkeys used in the current study, with references to the original datasets, and the lesion extents mapped for animals with bilateral excitotoxic lesions to amygdala (amyg) or ventral striatum (VS). (a) A diagram summarizing all twenty monkeys used for the experiments. Colors indicate lesion groups (black: control; red: amygdala-lesioned; blue: VS-lesioned). The Initial and number on each monkey indicate a unique identifier for each subject. The plus sign (+) indicates new monkeys that were newly trained for the experiment. Monkeys in faded colors (next to the arrows) indicate those used in the previous experiment.

- Please include a supplementary figure summarising the extent of the lesions. I know this has been included in previous publications but it would be useful for the reader to have it in this manuscript.

[R3.5] We appreciate this suggestion. We have now added the summary figure for the amygdala and ventral striatum lesions in **Supplementary Fig. 1b–d**, with specification of the individual monkey numbers from the previous diagram (**Supplementary Fig. 1a**).

Figure S1. (b) Extent of lesions in the three monkeys with bilateral lesions to ventral striatum (VS) who performed the What-only and What/Where task. Adapted from Rothenhoefer, K. M. et al. Effects of Ventral Striatum Lesions on Stimulus-Based versus Action-Based Reinforcement Learning. *Journal of Neuroscience* 19 July 2017, 37 (29) 6902-6914; DOI: 10.1523/JNEUROSCI.0631-17.2017. (c) Extent of lesions in the four monkeys with bilateral lesions to amygdala who performed the What-only task. This article was published in *Neuron* Volume 92, Costa VD, Dal Monte O, Lucas DR, Murray EA, Averbeck BB, "Amygdala and Ventral Striatum Make Distinct Contributions to Reinforcement Learning," 505-517, Copyright Elsevier (2016). (d) Extent of lesions in the four monkeys with bilateral lesions to amygdala who performed the What/Where task. Used with permission of Oxford University Press, from "Effects of Amygdala Lesions on Object-Based Versus Action-Based Learning in Macaques," Taswell CA, Costa VD, Basile BM, Pujara MS, Jones B, Manem N, Murray EA, Averbeck BB, 31(1), 2021; permission conveyed through Copyright Clearance Center, Inc.

Reviewer #4 (Remarks to the Author):

In the manuscript entitled 'Contribution of amygdala to dynamic model arbitration under uncertainty', Jae Hyung Woo and colleagues investigate the role of the amygdala in choosing which among two models of the environment (stimulus-based vs. action-based) best predict reward outcomes. They combined analyses of previously published experimental data and computational modeling to quantify concurrent learning in monkeys performing a probabilistic learning task with three forms of uncertainty: uncertainty about the better option, uncertainty about the correct model of the environment, and uncertainty about when reward associations change. The data from different groups was analyzed: bilateral lesions to the amygdala; bilateral lesions of the ventral striatum; control. They used entropy-based metrics to quantify consistency in choice and learning based on stimulus- and action-based learning over time. They found elements suggestive of a dynamic, competitive interaction between stimulus-based and action-based learning, and interpret their results as a distinct role of the amygdala. Specifically, they propose that the amygdala adjusts the initial balance between the two learning systems, thereby altering the interaction between arbitration and learning that shapes the time course of both learning and choice behaviors.

Overall, I have a mixed feeling after reading this manuscript. Since the experimental data is mostly (if not all, see question below) taken from previously published studies, only the computational analyses are new. Some of these analyses are nice and insightful. But overall there is no strong computational result that comes out of this work, since the modeling work is incomplete (see detailed comments), since many model parameters change at the same time, and since alternative interpretations are possible at multiple stages of the work.

We sincerely appreciate the effort the reviewer dedicated to reviewing our manuscript and for their detailed comments. Although the data used in our study have been previously published, the main contribution of our work is to integrate all the previous data into a common conceptual framework. More specifically, the main contribution of our study is to investigate the interaction between stimulus- and action-based learning systems, instead of examining them in isolation as in the original three studies. To that end, we used a new conceptual framework based on model arbitration to re-analyze the data while also considering the models used in the previous studies (single system, stimulus-only and action-only models). This allowed us to better account for and understand the behavior of both control and lesioned monkeys. It also helped connect findings from the three original studies and provided explanations for previously reported contradictory effects of amygdala lesions (e.g., the puzzling effect of amygdala lesions on facilitating reversals during stimulus-based task). The goal of this study was thus to provide a common conceptual framework across multiple datasets, which have not been done previously for monkeys or other animals. Critically, our new computational models consistently capture the choice behavior of monkeys across all groups and all experiments better than the models used in the original three studies, while revealing a novel role of amygdala in reward learning. Please see below for our point-by-point responses, which better clarify

the contributions of our computational framework and what the computational models reveal.

LIMITED ORIGINALITY

First, it is important to mention that this paper presents a novel series of analyses of data previously published in:

- Costa, V. D., Dal Monte, O., Lucas, D. R., Murray, E. A. & Averbeck, B. B. Amygdala and Ventral Striatum Make Distinct Contributions to Reinforcement Learning. *Neuron* 92, 505– 517 (2016).
- Rothenhoefer, K. M. et al. Effects of Ventral Striatum Lesions on Stimulus-Based versus Action-Based Reinforcement Learning. *J. Neurosci.* 37, 6902–6914 (2017).
- Taswell, C. A. et al. Effects of Amygdala Lesions on Object-Based Versus Action-Based Learning in Macaques. *Cereb. Cortex* 31, 529–546 (2021).

About the data from the What/Where task, the authors wrote (lines 704-705) that 'The remaining control and any of the amygdala-lesioned monkeys were not used in any previous studies.' However, they have in this paragraph only mentioned studies 14 and 22. Could the authors please double confirm that the control and amygdala-lesioned monkeys were not used in study 15 (Taswell et al. 2021) either? Study 15 indeed states that 'The subjects included 10 male rhesus macaques with weights ranging from 6 to 11 kg [... with] four of the male monkeys received bilateral excitotoxic lesions of the amygdala [...] Four out of the six unoperated control monkeys were the same monkeys used in a previous study (Costa et al. 2016). Five out of the six unoperated control monkeys were the same monkeys from an additional study (Rothenhoefer et al. 2017). All remaining monkeys were not previously used in the studies mentioned above.'

[R4.1] We thank the reviewer for asking these details. While we mentioned in the original manuscript that the data used in this study are from previous studies, we have clarified this point further in the revised version. To address the reviewer's questions, we have now added a diagram summarizing all monkeys that performed the experiments in a chronological order (**Supplementary Fig. 1a** copied below). In the process, we also identified an error regarding the monkeys used across experiments: only *one* of the control monkeys (monkey #2), and not four as previously stated, and all three VS-lesioned monkeys participated in both the What-only and What/Where tasks. We were previously referring to the monkeys #16 (control) and #17-20 (amygdala) as the monkeys that “were not used in any previous studies.” We tried to clarify this information in the **Methods** section as follows (Line #785):

“*One* of the five control monkeys and all of the three VS-lesioned monkeys were the same monkeys used in the What-only task¹⁷. The second study investigating the effect of amygdala lesions¹⁸ had a total of ten subjects weighing 6–11 kg (controls: $n = 6$; amygdala-lesioned: $n = 4$). *One* of the six unoperated controls *was* the same monkey used in the What-only task¹⁷ and the What/Where task involving VS lesion²⁴. One additional control monkey was used as an unoperated control for the earlier What/Where task only²⁴. *The remaining control and the four*

amygdala-lesioned monkeys were newly trained for this study. See **Supplementary Fig. S1a** for a summary diagram.”

Figure S1. Summary of monkeys used in the current study, with references to the original datasets, and the lesion extents mapped for animals with bilateral excitotoxic lesions to amygdala (amyg) or ventral striatum (VS). (a) A diagram summarizing all twenty monkeys used for the experiments. Colors indicate lesion groups (black: control; red: amygdala-lesioned; blue: VS-lesioned). The initial and number on each monkey indicate a unique identifier for each subject. The plus sign (+) indicates new monkeys that were newly trained for the experiment. Monkeys in faded colors (next to the arrows) indicate those used in the previous experiment.

In the discussion, the authors cannot write 'In addition to replicating these results' when referring to studies 14 and 15 which are the previous studies from the same authors from which the same data has been re-analyzed here. Replicating the results per se would imply that new data is acquired in new animals.

[R4.2] We thank the reviewer for pointing out this confusing point. We meant to say that our results using the new, dynamic two-system models are consistent with the previous results (which were obtained from the single-system models). We agree with the reviewer that this is not a replication per se, and we have changed the wording to clarify this point in the **Discussion**, as follows (second paragraph, Line #628):

“Previous studies using the same dataset have identified deficits in both stimulus-based and action-based learning due to amygdala lesions^{17,18}, but they considered these deficits independently, as they did not examine the interaction between stimulus- and action-based learning. Using a single-system RL model, they found that amygdala lesions reduce choice consistency (sensitivity to value signals) for stimulus-based learning¹⁷ and increase sensitivity to negative feedback (α) for action-based learning¹⁸. We found consistent results using our new two-system model with dynamic arbitration (**Supplementary Fig. S9b, d**), and moreover, provide a unified account based on the dynamic interaction between learning and arbitration under uncertainty.”

One of the conclusions ('The similar impairments observed in amygdala-lesioned monkeys during What and Where blocks cannot be explained by the amygdala's currently assumed role in stimulus-based learning.') has already been made and further analyzed by some of the same authors in [15] (Taswell et al. 2021)

[R4.3] We thank the reviewer for pointing this out. Our intention was not to highlight this finding as a new conclusion of this study but to set the tone for the new analyses in the upcoming section with respect to our new computational model. To clarify this, we have now made it clear that this observation has already been drawn in the previous study, as follows (Line #208):

“As noted in the previous work¹⁸, the similar impairments during the What and Where blocks observed in amygdala-lesioned monkeys cannot be explained by the amygdala’s currently assumed role in stimulus-based learning.”

COMPUTATIONAL MODELING LIMITATIONS

Parts of the computational analyses are nice, but incomplete and not always compelling.

While the authors show good model fitting, with k-fold cross-validation, they then draw conclusions from fixed effects over all monkeys and from tiny likelihood differences. It is important to also perform random effect analyses and show the fit distribution over subjects.

[R4.4] We appreciate the reviewer’s feedback on our computational analyses and suggestions for new analysis. Incorporating the reviewer’s feedback, we have adjusted our cross-validation procedure for the RL models as follows. Previously, as explained in the **Methods**, the blocks data were validated separately by their respective block type (for the What/Where task) and the reward schedule. We note that this categorization may have introduced an artificial division because, in reality, monkeys were oblivious to the true reward schedule or the block type. Especially for the What/Where task, separate validation of What and Where blocks and averaging the results likely leads to less distinction between the Static and Dynamic ω models. For example, Static ω could estimate high fixed ω values to the What blocks and low ω values to the Where blocks, which overestimates the performance when compared to validation scheme that mixes the two block types. To resolve this possible confound, we have simplified the cross-validation procedure by training/testing across all blocks experienced by each subject, without distinguishing the reward schedule or the block type. Also to provide an absolute measure for goodness-of-fit similar to the accuracy, we now report *McFadden R-squared* (in parentheses next to -LL). Please see the below for the revised results. The following results have been reflected accordingly on **Fig. 2a** (controls) and **Fig. 3a** (lesions).

Figure 2. (b) Comparison of goodness-of-fit across models. Plotted is the mean negative log-likelihood over all cross-validation instances for each task: What-only (black), What/Where (gray). Numbers in parenthesis indicate McFadden R² (Eq. 20).

Figure 3. (a) Comparison of the models' goodness-of-fit for choice behavior of amygdala- (left) and VS-lesioned (right) monkeys. Plotted is the mean negative log likelihood over all cross-validation instances for each task: What-only (black), What/Where (gray). Numbers in parenthesis indicate McFadden R².

With regard to the concern about small differences in negative log-likelihoods, it is important to note that these values are average across many instances of the held-out data in the cross-validation. Therefore, while the improvement in negative log-likelihood seems minimal on average, even a small increment in the likelihood results in a significant improvement in the predictability of the test data as the size of the tested dataset gets large. To explain this point in more detail, we have added a **Supplementary Note 3** (attached below). In brief, even a small difference in the negative log-likelihood, such as 0.05, implies that the test data is approximately $\exp(N \cdot 0.05)$ times more likely to have come from the superior model with the lower negative log-likelihood, where N is the number of blocks. As N gets larger, this quantity also exponentially increases. Considering that the median number of tested blocks across subjects for a given cross-validation instance was 92 in our case ($M \pm SD = 97.11 \pm 30.6$), the negative log-likelihood difference of 0.05 implies that the data is $\exp(N \cdot 0.05) = 99.48$ times more likely to have originated from the model with lower likelihood. Please, also refer to Mohammadi, ..., Pillows, 2024, *bioRxiv* for a similar argument.

Regarding the random effects analysis, we agree that this should be performed on different metrics used to evaluate behavior. Therefore, as suggested by the reviewers, we have now primarily employed mixed-effects analysis with subjects (monkeys) as random effects to account for cross-subject variability, given that each animal performed a large number of blocks across multiple session days. As a result, we have revised the statistical analyses throughout the manuscript based on the updated statistical approach. Below, we outline the changes resulting from the use of these new statistical tests. Overall, while some previously significant results are no longer statistically significant with the revised method, the results supporting the main claims of the study remain qualitatively the same.

In contrast to standard metrics, there are important caveats to performing direct statistical tests on the cross-validated measures (e.g., Nadeau & Bengio, 1999 *NeurIPS*), as the repeated use of the same data points across different instances introduces dependence among the log-likelihood estimates. Further, the division of data into training and test samples during cross-validation suggests that the two sets are not independent from each other (Kohavi 1995, *IJCAI*). These factors often lead to violation of the independence assumption required by many of the statistical tests. Therefore, the approach we took here, which is standard in the field, is to estimate the total mean of the negative log-likelihoods, with the assumption that the cumulative mean over a sufficient number of validation instances will asymptotically approach the “true” performance of the model (Hastie, Tibshirani, & Friedman, 2009).

Finally, to address the reviewer’s concern regarding subject variability, we have now added **Supplementary Table 2** below, which includes the goodness-of-fit for each model and for each individual subject. The results show that the full model consistently shows the lowest negative log-likelihoods for all subjects, demonstrating that the Dynamic arbitration model consistently provides a better fit for the choice behavior of all monkeys.

What-only task (11 monkeys)											
Groups	Controls				Amygdala-lesioned				VS-lesioned		
Subjects	E (1)	G (2)	M (3)	N (4)	C (5)	CB (6)	P (7)	Q (8)	F (9)	A (10)	G (11)
RL _{Stim-only}	25.77	16.34	24.92	25.86	49.24	39.31	43.41	42.85	43.04	42.41	44.53
RL _{Action-only}	54.46	55.18	54.67	54.96	50.14	45.76	50.36	54.51	48.94	48.95	48.95
Static ω (RL _{Stim+Act})	25.65	16.05	24.81	25.77	47.86	37.79	42.44	42.42	37.62	38.63	43.8
Dynamic ω : RPE	25.63	16.06	24.81	25.77	47.73	37.67	42.36	42.40	37.35	38.24	43.8
Dynamic ω : V _{cho}	25.39	14.97	24.66	25.69	47.63	37.46	42.15	42.13	36.24	37.69	43.74
Dynamic ω - ρ (V _{cho})	25.38 *	14.96 *	24.55 *	25.65 *	47.49 *	37.39 *	42.15 *	42.13 *	36.24 *	37.53 *	43.73 *

What/Where task (13 monkeys)													
Groups	Controls						Amygdala-lesioned				VS-lesioned		
Subjects	G (2)	R (12)	U (13)	V (13)	W (14)	B (16)	Be (17)	En (18)	MJ (19)	Z (20)	F (9)	A (10)	G (11)
RL _{Stim-only}	38.27	41.21	49.24	42.2	44.92	42.48	53.16	47.25	45.47	51.27	54.27	53.48	52.67
RL _{Action-only}	47.76	47.3	43.89	43.42	41.36	40.33	36.83	45.32	47.27	47.73	34.92	33.69	36.83
Static ω (RL _{Stim+Act})	26.07	30.16	35.88	25.65	26.26	22.59	33.37	37.08	35.51	43.67	33.6	32.9	35.36
Dynamic ω : RPE	25.92	30.09	35.76	24.62	25.25	21.68	32.38	36.73	33.32	43.47	33.44	32.67	34.93
Dynamic ω : V _{cho}	24.92	29.32	34.72	22.88	23.31	18.53	31.88	35.86	33.44	42.57	32.68	32.23	34.43
Dynamic ω - ρ (V _{cho})	24.84 *	29.19 *	34.56 *	22.88 *	23.29 *	18.51 *	31.86 *	35.66 *	33.33 *	42.41 *	32.62 *	32.18 *	34.43 *

Supplementary Table 2. Cross-validated negative log-likelihoods of RL models, separately for each monkey in a given task. Reported are the mean negative log-likelihoods for each monkey across all tested block instances (five-fold cross-validation), rounded to the nearest hundredth. Asterisks indicate the model with the lowest value of negative log-likelihood for each task. The initial and the number in parenthesis indicate a unique identifier for each monkey, as shown in **Supplementary Fig. S1**.

Supplementary Note 3

Our model fitting relies on maximum likelihood estimation (MLE), a method that maximizes the likelihood function ($\mathcal{L}(\theta_i; \mathbf{y}) = P(\mathbf{y}|\theta_i)$), ensuring that the observed data (\mathbf{y}) is the most likely outcome under the model defined by the parameters (θ_i). In our case, the data consist of the animals' choices on each trial of the experiment ($C(t)$, where $C(t) = 1$ and 0 for left and right choices), leading to the optimization of the following expression for each block of 80 trials:

$$\ell = \ln \mathcal{L}(\theta_i; \mathbf{y}) = \ln [P(\mathbf{y}|\theta_i)] = \ln \left[\prod_{t=1}^{80} \{P_{Left}(t) C(t) + (1 - P_{Left}(t))(1 - C(t))\} \right] \quad (Eq. S1)$$

where $P_{Left}(t)$ is the probability of choosing left as predicted by the model (see Eq. 5 in the Methods), and the logarithm of the likelihood function ($\ell = \ln \mathcal{L}$, referred to as the log-likelihood) is used to perform the optimization more conveniently.

We then used cross-validated negative log-likelihoods ($-LL$) of the alternative models to identify the best-fitting model, where a smaller negative LL indicates a better fit. Notably, even a small improvement in negative LL reflects a significant increase in the likelihood that the superior model better explains the observed choice behavior. To illustrate this, consider an example comparing the negative log-likelihoods of two models, i and j ($-LL_i$ and $-LL_j$):

$$\Delta LL = -LL_i - (-LL_j) \quad (\text{Eq. S2})$$

where LL_i is the mean log-likelihood of model i across the N test blocks:

$$LL_i = \frac{1}{N} \sum_{b=1}^N \ell(y_b|\theta_i) \quad (\text{Eq. S3})$$

where $\ell(y_b|\theta_i)$ represents the optimized log-likelihood of the model i for the data in block b (y_b), as described in Eq. S1 above. Noting that all blocks consist of 80 trials, it then follows:

$$\Delta LL = \frac{1}{N} \sum_{b=1}^N (\ell(y_b|\theta_j) - \ell(y_b|\theta_i)) = \frac{1}{N} \sum_{b=1}^N \left\{ \ln \left[\frac{P(y_b|\theta_j)}{P(y_b|\theta_i)} \right] \right\}, \quad (\text{Eq. S4})$$

and therefore,

$$N \times \Delta LL = \sum_{b=1}^N \left\{ \ln \left[\frac{P(y_b|\theta_j)}{P(y_b|\theta_i)} \right] \right\} = \ln \left[\prod_{b=1}^N \frac{P(y_b|\theta_j)}{P(y_b|\theta_i)} \right]. \quad (\text{Eq. S5})$$

By taking exponential on both sides, we obtain

$$e^{N \times \Delta LL} = \frac{\prod_b P(y_b|\theta_j)}{\prod_b P(y_b|\theta_i)}, \quad (\text{Eq. S6})$$

which leads to the relationship between the likelihoods of the two models:

$$\prod_b P(y_b|\theta_j) = e^{N \times \Delta LL} \times \prod_b P(y_b|\theta_i). \quad (\text{Eq. S7})$$

Considering that the median number of tested blocks across subjects for a given cross-validation instance was 92 ($M \pm SD = 97.11 \pm 30.6$, with a minimum of 48, corresponding to 20% of the blocks), even a slight difference in the log-likelihood, such as $\Delta LL = 0.05$ (indicating the superiority of Model j), implies that the test data is approximately $e^{92 \times 0.05} = 99.48$ times more likely to have originated from Model j compared to Model i on average.

Moreover, I don't find the interpretations about the way the model captures monkey behavior compelling, since all model parameters change. There is no clear result, where a single parameter significantly varies, so as to interpret behavioral properties and deficits as being due to a specific mechanism.

Rather, the computational analyses remain at a descriptive level: posterior predictive analyses are lacking; the study also lacks simulations of ablated versions of the model to show that specific artificial lesions produce the same behavior as the animals.

[R4.5] We thank the reviewer for raising these concerns, some of which are related to a lack of clarity in our presentation and others to misunderstandings of our results. We first note that, even for the control monkeys, there is a high variability in the fitted parameters across sessions. This is expected when fitting RL models to large datasets of binary choices. Therefore, while it is true that many of the parameters vary between different groups of monkeys and tasks, our model-based analyses are primarily focused on the parameters related to the arbitration process, as our study centers on the interaction between two learning strategies and the effects of brain lesions on this process. Specifically, **Fig. 4** in the manuscript has centered on the two key parameters in the arbitration model: relative baseline strength of stimulus-value to action-value signals (ρ), and the initial arbitration weight (ω_0).

Indeed, these are the two main parameters of the model that consistently distinguish each of the lesioned groups from controls across tasks. More specifically, by employing mixed-effects analysis on fitted parameters compiled across all groups and tasks, we found significant main effect of amygdala group ($\beta_{amyg} = -0.191$, $p = .0171$) on ω_0 values, suggesting a consistent reduction in ω_0 compared to controls across both tasks. In contrast, VS-lesioned monkeys do not show such tendency ($\beta_{VS} = -0.071$, $p = .420$). Instead, VS-lesioned monkeys were most consistently distinguished from controls in terms of relative sensitivity of choice (**Fig. 4b–c** insets) arising from the ρ parameter (**Fig. 4a**). This shows a significant negative effect of the VS group across tasks ($\beta_{VS} = -5.27$, $p = .0014$), whereas amygdala group does not show this tendency ($\beta_{amyg} = -0.44$, $p = .771$).

Thus in our framework, the “lesions” correspond to the alteration of some of the model parameters to mimic the behavior of the lesioned animals: (1) reduction in ω_0 for amygdala-lesions, capturing a bias in initial arbitration weight toward action-based system; and (2) reduction in ρ for VS-lesions, capturing reduced sensitivity of choice to stimulus-value signals. Accordingly, to examine the behavioral impact of the mechanisms specific to ρ and ω_0 , in our analyses presented in **Fig. 4d–f**, we have tried to keep as many model parameters constant as possible across simulations.

Importantly, by modulating the ρ and ω_0 parameters to simulate the artificial “lesions,” we have shown that the behavioral tendency observed in monkey data (**Fig. 3**), including the difference between the effective arbitration rates ($\psi_+ - \psi_-$; **Fig. 4d–f**) and the resulting performance (**Fig. 4g–i**), can be qualitatively captured by our model. These results indeed

demonstrate that our model can reproduce behavioral tendencies of both control and lesioned monkeys.

Regarding the reviewer's concern about the analyses remaining at the descriptive level, we fully acknowledge the importance of performing posterior predictive analyses to confirm that the best RL model captures the relevant aspects of the data. For this very reason, many of our analyses were aimed in this direction, including the simulations of effective arbitration rates and performance in **Fig. 4** (as explained above) and the simulation of competitive interaction during the What-only task (**Fig. S6**; for controls). In particular, the model validation analysis in **Fig. S7** (for brain-lesioned groups) were intended to show how the dynamic two-system model better captures the behavioral patterns in the data than the single-system models, thus providing further rationale for the use of the two-system model in analyzing the effects of lesions from the perspective of arbitration processes. Lastly, although the simulation results in **Fig. 5** were mainly for illustrative purposes to highlight the interplay between learning and arbitration, they also make qualitative predictions about performance after the reversals based on the combination of adopted parameters. This provides a valuable framework for future studies with a priori predictions about the behavior.

Therefore, we believe that our study includes a significant number of posterior analyses, comparable to that of descriptive analyses, and that have tested whether the dynamic, reliability-based arbitration between two learning systems is a valid modeling framework for the examined datasets.

To summarize, the overall flow of our analyses using multiple computational approaches involved the following steps: (i) providing evidence for the presence of multiple learning systems; (ii) fitting various RL models (including those in the original studies) to animals' choice data and demonstrating the superiority of the dynamic two-system models in accounting for data across all conditions (**Fig. 2b, 3a**); (iii) model and parameter recovery to ensure that the parameters of the winning model are well recovered and performing descriptive analysis of behavior based on the estimated parameters (**Fig. 2c–e, 3b–g, 4a–c**); (iv) validating that the model also captures the key aspects of the data through simulations (e.g., **Fig. S6, S7**); (v) analyzing key parameters related to arbitration (ω , ρ) to identify the specific effects of lesions on the arbitration process (**Fig. 4d–f**); (vi) illustrating how dynamic interactions between learning and arbitration can create diverse behavior, with implications for understanding results of other amygdala-lesion studies (**Fig. 5**).

Nevertheless, to provide readers with more quantitative comparisons for the brain-lesioned group, we have included additional plots in **Fig. S7** below (new panels **a–f**), which explicitly compare the distance between the empirical and simulated distributions using the Kolmogorov-Smirnov test. These provide additional evidence that the parameters estimated from the arbitration models capture the behavioral effects of the lesions. Moreover, to clarify that we are simulating the effects of lesions on the arbitration process (**Fig. 4**), we have provided additional statements in **Introduction** and **Results** as following:

(Line #83; in the last paragraph of the **Introduction**)

“Additionally, we developed several new reinforcement learning (RL) models that, along with previous models, were used to fit choice behavior on a trial-by-trial basis. These new models extended the previous ones by incorporating static or dynamic arbitration among alternative learning systems, based on different signals. We then examined the best models and their estimated parameters, particularly those related to the arbitration process, to pinpoint the roles of the amygdala and ventral striatum in reward learning and arbitration. Moreover, by modulating the key parameters of the model, we simulated and replicated the distinct behavioral signatures of amygdala or VS lesions. Together, by utilizing the above methods, we provide evidence for interactions between stimulus-based and action-based learning under uncertainty, uncover mechanisms underlying arbitration between the two systems, explore how arbitration and learning processes interact, and determine the amygdala’s contributions to arbitration and overall behavior.”

(Line #483; in the **Results** explaining the analysis presented in **Fig. 4d–f**.)

“To further validate this idea through model simulations, we generated the choice behavior of the Dynamic ω - ρ model by adjusting two key parameters to mimic the effects of brain lesions: baseline ratio of the weights of the stimulus- to action-value signals (ρ) and the initial arbitration weight (ω_0). These two parameters reflected the most consistent effect of lesions across the two tasks, with reduced ρ in VS-lesioned monkeys (**Fig. 4a–c**; $\beta_{VS} = -195$, $p = .00279$; mixed-effects analysis on compiled ρ across all groups/tasks) and reduced ω_0 in amygdala-lesioned monkeys ($\beta_{amyg} = -0.191$, $p = .0171$; mixed-effects analysis on compiled ω_0 across groups/tasks).”

(Line #516; toward the very end of the section titled *Dynamic interaction between learning and arbitration processes and the impact of initial state* in **Results**):

“It is worth noting that the simulations using $\rho = 0.4$ (**Fig. 4f, i**), which mimics the reduction in the relative baseline strength of stimulus-value signals due to VS lesions, result in the biased update rates toward action-based system for many of the ω_0 values (blue lines in **Fig. 4f**), including $\omega_0 \sim 0.37$, which matches the initial values for control monkeys. Therefore, the consistent adoption of action-based strategy in the What/Where task can be sufficiently accounted by reduced ρ value, without the need for additional constraints on ω_0 . These results support the notion that the impairments observed in VS-lesioned monkeys during this task can be fully explained by a reduction in stimulus-value signals, with minimal involvement of deficits in arbitration processes.”

Figure S7. Model validation results demonstrate that the model with dynamic arbitration better captures the relative strength of the two strategies compared to the model with no arbitration. (a) Plotted are cumulative distribution functions (CDF) of empirical (magenta) and simulated (red) values of the relative strength of two strategies, $ERDS_{Stim} - ERDS_{Action}$ in amygdala-lesioned monkeys. Shaded bars indicate 95% confidence interval. Reported values are the test statistics (D -values) and its p -value from the Kolmogorov-Smirnov test, comparing the distance between the distributions of data and the indicated model (D_{data}) or between the distributions of two models (D_{models} , colored font). (b) CDF of empirical and simulated values of $ERDS_{Stim} - ERDS_{Action}$ for the What blocks during the What/Where task. Conventions are the same as in (a). (c) Same plot as in (b) but for the Where blocks during the What/Where task. (d-f) Same plots as in (a-c) but for VS-lesioned monkeys. (g-i) Mean values of $ERDS_{Stim} - ERDS_{Action}$ shown in (a-f), broken down by reward schedules (magenta/cyan lines for empirical data, same as those shown in Fig. S3d-f, and red/blue lines for model simulations), during the What-only (g) and the What/Where task (h,i). Error bars = SEM. across subjects. The two-system model with dynamic arbitration, the Dynamic $\omega-\rho$ model, more effectively captures the data compared to the single-system models without arbitration.

Moreover, to make a substantial contribution that justifies publishing yet another paper with the same data, the authors should show that the specific behavioral deficits produced by lesioning a specific part of the arbitration model enables to explain why the same authors with the same data have previously interpreted amygdala lesions as 'reduc[ing]

choice consistency (sensitivity to value signals) for stimulus-based learning and increas[ing] sensitivity to negative feedback (α^-) for action-based learning'. Similarly, they should produce VS-equivalent lesions on the arbitration model and help explain why they previously interpreted VS-lesions as producing 'deficits in learning to select rewarding images but not rewarding actions', as displaying 'more influence by negative feedback and [...] lower choice consistency than controls', and as 'only affect[ing] learning in the stochastic task' of Costa et al 2016.

[R4.6] Regarding the lesioning of the specific part of the arbitration model, please see our previous response (**[R4.5]**). To better clarify the relationships between our findings and previous findings using the same datasets, we have revised the quoted passage as follows (also see **[R4.2]**) (Line #631):

“Using a single-system RL model, they found that amygdala lesions reduce choice consistency (sensitivity to value signals) for stimulus-based learning¹⁷ and increase sensitivity to negative feedback (α^-) for action-based learning¹⁸. We found consistent results using our new two-system model with dynamic arbitration (**Supplementary Fig. S9b, d**), and moreover, provide a unified account based on the dynamic interaction between learning and arbitration under uncertainty. Specifically, our simulation results mimicking amygdala lesions (**Fig. 4**) suggest that a biased initial state strongly favoring action-based learning is the key feature of the deficits observed in amygdala-lesioned monkeys. This strong initial bias altered the interaction between decision-making, learning, and arbitration processes, making the arbitration update rates between two systems to be less distinguishable from each other compared to the controls. Importantly, VS-lesioned monkeys exhibit a smaller sensitivity to the stimulus-based compared to the action-based signals. As a result, VS lesions led to an overall bias in arbitration update rates toward action-based learning in both What and Where blocks.”

To clarify, the results mentioned in this quoted passage were referring to the previous observations from the single-system models. As the reviewer has noticed, those studies have mainly focused on the learning rates and the choice consistency which specify the choice mechanisms of the single-system model, which assumes that learning happens about just a single attribute of the environment. In our current study, our aim was to provide additional insights and nuances to what has been not attributed before, by utilizing the two-system model that specifies the (previously unaddressed) mechanism of arbitration, which accordingly has been found to account for the choice data significantly better than the single-system models. After finding the evidence that two learning systems provide much better fit to choice data, we have dove into the mechanism of their interactions and arbitration based on their reliability. As noted in our above response in **[R4.5]**, the single-system models are insufficient to address this aspect of the behavior (i.e., arbitration between two distinct learning strategies; **Fig. S7**), and the effects of brain lesions are simulated by altering the key parameters of the model (ρ and ω_0) to mimic the effect of lesions on monkeys' choice behavior (**Fig. 4**). Since these simulations kept other parameters fixed, they suggest that other factors such as the learning rates are not the key contributors to the observed deficits specific to the arbitration process. Therefore, while the dynamic arbitration model reflects some of the previous results obtained using the single-system models (e.g., increased sensitivity to negative feedback in lesioned

animals), its novelty lies in its ability to capture the arbitration process and how it depends on the specific lesion.

Nonetheless, we have conducted additional analyses to directly address the reviewer's question regarding how our current findings using the dynamic arbitration (two-system) model relate to, or can account for, the previous results from one-system models. First, with respect to the reduced choice consistency (inverse temperature) in lesioned animals, we note that the evidence for the reduced choice consistency is much weaker for the two-system model compared to one-system model. For example, according to the dynamic two-system model examined in our study, both lesioned groups were associated with overall smaller yet non-significant reduction in inverse temperature compared to controls in the What-only task (mixed-effects analysis; $\beta_{\text{amyg}} = -13.57$, $p = .0581$; $\beta_{\text{VS}} = -11.56$, $p = .169$; see updated **Supplementary Fig. S10b** reproduced below as **Fig. R4.6.1**, left panel). This contrasts with the single-system (Stim-only) model which exhibits more pronounced reduction in the inverse temperature ($\beta_{\text{amyg}} = -13.9$, $p = .00546$; $\beta_{\text{VS}} = -14.68$, $p = .0103$; panel on the right in **Fig. R4.6.1**). This discrepancy is due to the overall smaller inverse temperature of the one-system models compared to the two-system model across all groups. This is an indication that one-system models make less precise predictions on the value difference, and consequently results in the reduced choice consistency to compensate for the lack of precision. Another way of stating this is that since the one-system models lack the mechanisms to account for choice processes generated by arbitration, this unaccounted portion of choice was instead estimated as being "inconsistent." (note the worse quality of fit in the one-system models also supports this notion) Therefore, we believe that the limited predictive power of one-system models in explaining lesioned monkeys' behavior was previously masked under the effects of low choice consistency compared to controls in the original study (Costa et al., 2016).

In comparison, with regards to the learning rates, we find that the increased sensitivity to negative feedback (α -) in amygdala-lesioned monkeys is relatively well preserved across both models. That is, larger α - in amygdala group compared to controls during the What/Where task is reflected in both the dynamic arbitration model ($\beta_{\text{amyg}} = 0.169$, $p = .00895$; see updated **Supplementary Fig. S9d** reproduced below as **Fig. R4.6.2**) and the single-system (Act-only) model ($\beta_{\text{amyg}} = 0.367$, $p = 5.91 \times 10^{-10}$; corresponding to the previous study's finding). These results reflect the fact that the sensitivity to reward feedback is a more local aspect of behavior which can be sufficiently explained by single-system models without invoking more complex arbitration components. As these results have been discussed in detail in the previous study (Taswell et al., 2021), we have simply noted this point as our study was more focused on the arbitration.

The reviewer's other quoted passage ('deficits in learning to select rewarding images but not rewarding actions'), which we believe is taken from Rothenhoefer et al. (2017), is consistent with the VS group's overall relative arbitration weight (ρ) being biased toward action-based learning in the What/Where task. Through simulations in **Fig. 4f**, we have shown that assuming a reduction in ρ can reproduce the behavioral patterns in VS-

lesioned monkeys. As previously mentioned in our response [R4.5], we have explained this point in more detail as follows (Line #516):

“It is worth noting that the simulations using $\rho = 0.4$ (Fig. 4f, i), which mimics the reduction in the relative baseline strength of stimulus-value signals due to VS lesions, result in the biased update rates toward action-based system for many of the ω_0 values (blue lines in Fig. 4f), including $\omega_0 \sim 0.37$, which matches the initial values for control monkeys. Therefore, the consistent adoption of action-based strategy in the What/Where task can be sufficiently accounted by reduced ρ value, without the need for additional constraints on ω_0 . These results support the notion that the impairments observed in VS-lesioned monkeys during this task can be fully explained by a reduction in stimulus-value signals, with minimal involvement of deficits in arbitration processes.”

Lastly, regarding the distinction between the stochastic and deterministic tasks of Costa et al. (2016), we would like to mention that the deterministic portion of this data had been excluded from our analysis, because the What/Where task did not have the corresponding deterministic schedule. Therefore, we believe that addressing the distinction between deterministic and stochastic schedule is out of scope for our current study (Also see [R4.14] for a related response, delineating the current challenges and motivation of the study). To make it more explicit that we have excluded the deterministic schedule, we have updated the **Methods** section describing the What-only task as follows (Line #771):

Monkeys performed the deterministic task (100/0 reward schedule) after the data collection for the stochastic task had been completed. Here, we focus on our analyses of the task’s stochastic variant to match the reward schedules used in the What/Where task (which only contained stochastic schedules, as described below), and therefore, we have excluded the deterministic portion of the data from our analyses.

Figure R4.6.2. Comparison of negative learning rate by two-system model (left) and one-system, action-only model (right; similar to Taswell et al., 2021) during What/Where task. Asterisks indicate significant effects of group difference (mixed-effects analysis with random effects of subjects).

The correlations with reaction times is not satisfying either, because error rates constitute a confounding factor.

The arbitration model's ability to account for 'VS lesions hasten[ing] the monkeys' choice reaction times' should also be tested.

Overall, additional analyses are required to verify that the authors are not just interpreting small differences in the identified patterns which could simply be due to noise.

[R4.7] We were not sure which part of our manuscript the reviewer is referring to in their comment about the “VS hasten[ing] the monkeys choice response times.” Our RT analysis here was mainly focused on the effect of adopted strategy (stimulus vs. action), rather than the effect of specific lesions on the RT. In the **Supplementary Note 1**, however, we linked the “faster decision making during action-based compared with stimulus-based tasks” to the previous finding that VS-lesioned monkeys exhibited overall faster RT compared to the other groups (Rothenhoefer et al., 2017; Costa et al., 2016).

First, we note that RL models are not generally designed to capture the temporal dynamics of the reaction time within a trial, because they do not include a mechanism for simulating the decision-making process in real time (e.g., as in drift-diffusion or attractor network models). Nonetheless, to provide a more comprehensive analysis of RT's link to arbitration, we performed additional regression analyses to predict the trial-by-trial RT from various model-derived behavioral variables, including dynamic arbitration weight and value difference. More specifically, we used mixed-effects model with a random effect of subjects and main effects of the following predictors: arbitration weight (Ω), absolute overall value difference ($|\Delta OV|$), whether the animal has chosen the correct option (1) or not (0), trial number within a block, block number within a session, and session number within the subject. Variables were normalized (within each subject) to yield comparable standardized regression coefficients.

From this analysis, we found that the arbitration weight significantly predicted RT, demonstrating the predictive power of the estimated parameters of the model. Results from this analysis are provided as a new **Supplementary Note 2** (included below). Briefly, we found that VS-lesioned monkeys in the What-only task exhibited a large,

significant effect of Ω on RT ($\beta_{\Omega} = 0.194$, $p = 4.94 \times 10^{-324}$), indicating that more action-based strategy led to faster decisions (**Fig. SN2** below). The effect was smaller but significant for the What/Where task, where VS-lesioned monkeys showed significant effects of Ω on RT ($\beta_{\Omega} = 0.0562$, $p = 1.60 \times 10^{-91}$). Combining these results, the dynamic arbitration model could potentially account for the hastened RT as follows: in the What-only task, VS group seems to show a large modulatory effect of Ω on RT, such that a mixture of action-based strategy results in the significantly reduced RT compared to the other groups. In the What/Where task, the effect of Ω on RT is relatively moderate, but as VS group heavily relies on action-based strategy in this task (i.e., small Ω), the resulting RT is also predicted to be smaller. Overall, the Ω regressor is a significant predictor of RT across conditions and could therefore point to the potential role of arbitration in the hastened RT due to VS lesions. Please see the next page for the added results in detail, under **Supplementary Note 2**.

Based on these results, we have updated the main text as follows (Line #276):

“For the What-only task, we found a small yet significant correlation between the effective arbitration weight and RT (Spearman’s $r = .094$, $p = 1.24 \times 10^{-4}$). In comparison, Ω and RT were highly correlated in the What-Where task ($r = .414$, $p = 3.78 \times 10^{-245}$; **Fig. 2e**). However, because these simple correlations do not control for other confounding factors such as choice confidence (measured by value difference), choice accuracy (choosing the better or worse option), and long-term drift in RT, we conducted further regression analyses. These analyses included these factors along with other model-derived measures to predict trial-by-trial RT (**Supplementary Note 2**). Using these analyses, we found that across all groups and conditions, higher Ω predicted longer RT (**Fig. SN2**). This means that slower (respectively, faster) RT occurred when larger weights were assigned to the stimulus-based (respectively, action-based) system, consistent with the previous analysis in which action-dominant trials (determined using ERDS) accompanied by faster RT.”

We believe that these additional analyses should address the reviewer’s concern about confounding factors related to RT. While the simple correlation between Ω and RT (e.g., **Fig. 2e**) suggests an overall association between the two variables, these regression-based analyses also control for the other confounding factors such as the subjective estimate of animal’s confidence (based on value difference), choice of the “correct” options, and long-term drift in RT (which can be considered as “noise” irrespective of the adopted strategy).

Supplementary Note 2

In **Supplementary Note 1**, we showed that the more dominantly adopted strategy has a significant effect on RT, with a stimulus-dominant strategy requiring longer RT regardless of the given block type. Based on these findings, we also tested whether the predictors derived from the RL model (Dynamic ω - ρ) yield consistent results. That is, we hypothesized that the RL model's arbitration weight Ω , quantifying the relative weight of stimulus- and action-based system in driving choice, is a significant modulator of RT. This quantity provides a more fine-grained measure of relative strategy compared to the ERDS-based approach used in **Supplementary Note 1**, which binarizes the otherwise continuous degree of relative strategy into action- or stimulus-dominant. To this end, we used a mixed-effects analysis with a random effect of subjects and main effects of the following predictors: arbitration weight (Ω), absolute overall value difference ($|\Delta OV| = |OV_{\text{Left}} - OV_{\text{Right}}|$), whether the animal's choice was the "correct" option (in terms of reward probability), trial number within a block, block number within a session, and session number within the subject. Note that in this analysis, the block type (for the What/Where task) and reward uncertainty were not included as predictors, because these were objective variables unknown to the monkeys and were instead captured by arbitration weight and the value difference (reflecting monkey's subjective estimates).

Our analysis based on model-derived estimates (**Fig. SN2**) showed significant positive effects of arbitration weight on RT across all tasks and groups, supporting the view that action-based decision is faster than stimulus-based decision. More specifically, the regression weight of arbitration weight for control monkeys during the What-only task was significant ($\beta_{\Omega} = 0.0131$, $p = 1.10 \times 10^{-5}$; **Fig. SN2a**), but to a lesser degree compared to the brain-lesioned monkeys (amygdala: $\beta_{\Omega} = 0.0489$, $p = 4.90 \times 10^{-60}$; **Fig. SN2c**; VS: $\beta_{\Omega} = 0.194$, $p = 4.94 \times 10^{-324}$; **Fig. SN2e**). This could be related to the fact that arbitration weights in control monkeys were saturated toward the stimulus-based system (see **Fig. 2d**), in contrast to the lesioned groups that exhibited more mixture of two strategies (e.g., **Fig. 3b,c**). In particular, the largest proportion of variance in VS group's RT was accounted by the arbitration weight ($R^2_{\Omega} = 0.0339$, followed by $R^2_{\text{Block\#}} = 0.0218$ and $R^2_{|\Delta OV|} = 0.0141$; partial R^2 values), suggesting its significant role in reducing RT of VS-lesioned monkeys. Furthermore, across all groups, we found significant effects of $|\Delta OV|$, such that the larger distinction between two choice options led to significantly shorter RT (controls: $\beta_{|\Delta OV|} = -0.0763$, $p = 2.58 \times 10^{-132}$; amygdala: $\beta_{|\Delta OV|} = -0.0595$, $p = 7.13 \times 10^{-85}$; VS: $\beta_{|\Delta OV|} = -0.123$, $p = 2.63 \times 10^{-207}$). These results illustrate the RL model's ability to capture the animals' subjective value estimates and demonstrate that the observed effects of the arbitration weight on RT are unique from other included effects.

During the What/Where task, which required more explicit arbitration between two strategies, all groups showed highly significant effects of Ω on RT, with higher Ω predicting slower RT (controls: $\beta_{\Omega} = 0.102$, $p = 4.94 \times 10^{-324}$; amygdala: $\beta_{\Omega} = 0.174$, $p = 4.94 \times 10^{-324}$; VS: $\beta_{\Omega} = 0.0562$, $p = 1.60 \times 10^{-91}$). These results further support the claim that the relative dominance of the strategy, even when measured in a continuous scale, can significantly predict the animals' RT. Of particular relevance to the VS lesion is the significant effect of arbitration weight on RT: in the What-only task, VS group seems to show a large modulatory effect of Ω on RT, such that a mixture of action-based strategy results in the significantly reduced RT compared to the other groups. In the What/Where task, the effect of Ω on RT is relatively moderate compared to the first task, yet since VS group heavily relies on action-based strategy for this task (small Ω), the resulting RT is also predicted to be smaller. Overall, these results point to the potential role of arbitration in the hastening RT due to VS lesions.

Figure SN2. Regression of reaction time (RT) on arbitration weight and other task-related variables.

Plotted are standardized regression coefficients from a mixed-effects model with subjects as a random effect, for given lesioned groups and tasks. Asterisk indicates significant effect of a given predictor (*: $p < .05$, **: $p < .01$, ***: $p < .001$). *corr. vs. incorr.*: the main effect of choosing the option with higher reward probability (correct option). (a,c,e) Regression coefficients for predicting RT data during What-only task, in controls (a), amygdala- (c), and VS-lesioned (e) monkeys. (b,d,f) Regression coefficients for predicting RT data during What/Where task, in controls (b), amygdala- (d), and VS-lesioned (f) monkeys. Overall, task-related variables have similar impacts on RT in all groups.

ALTERNATIVE BEHAVIORAL INTERPRETATIONS

A neglected aspect which may have played an important role here is the sequentiality between the tasks, and whether having first performed the What-only task sets priors in favor of the stimulus-based strategy for the following What/Where task. In the Methods, the authors state that 'Four of the five controls and all of the three VS-lesioned monkeys were the same monkeys used in the What-only task'. Could the authors please confirm that those monkeys first performed the What-only task and then the What/Where task? What happens when animals start with the latter? What happens when animals start with a Where-only task and then perform a What/Where task? Could the authors disentangle the role of such priors during the What/Where task?

[R4.8] We thank the reviewer for asking these questions. As mentioned in our previous response (in **[R.4.1]**), we have double-checked all the names of the individual monkeys, along with their prior experience and training history. We have summarized this information in the new **Supplementary Fig. 1a** (see above). In terms of the task order, we have confirmed that any monkeys that have participated in both tasks first completed the What-only task, which is also reflected in the year of the original studies (2016 for What-only, and 2017/2021 for What/Where). In other words, there were no monkeys that first performed the What/Where task and then moved to the What-only task. In addition, there were no previous task designs that involved the 'Where-only' task. Importantly, all newly trained monkeys for the What/Where task (i.e., monkey #12-15 and #16-20) were initially trained to learn about stimulus-based reward associations (i.e., corresponding to the What-only task), meaning that all monkeys analyzed for our current study share similar priors (i.e., learning about "What" first and then being introduced to the "Where" condition). To clarify this point, we have added the following information about prior experience and the task training in the **Methods** (*Experimental paradigm, What/Where task*; Line #793):

"Any monkeys that have participated in both the What-only and What/Where tasks (i.e., one control and three VS-lesioned monkeys) first completed the What-only task and then later completed the What/Where task. Notably, all newly trained monkeys that performed the What/Where task were first trained on a simple two-armed bandit with stimulus-based reward associations ("What" condition). After learning about this task, they were then trained with a deterministic version (100/0) of the What/Where task and gradually transitioned into the probabilistic outcomes used for this experiment (80/20, 70/30, 60/40). As such, all monkeys in the study shared the same prior training experience, specifically learning the "What" task first."

There is also a problem with the association the authors make between the What-only task and 'the absence of uncertainty about the correct model of the environment'. This is true only from an ideal observer point of view. This was not true for the monkeys, especially during initial trials of each block, and trials following reversals. This view is supported by the authors' observation that the animal's strategy was less than 80% of the

trials classified as stimulus-dominant in the What-only task (Supplementary Note 1, Line 1155). Further evidence comes from the finding that arbitration models get a lower log likelihood than the stim-only model in the What-only task (Figure 2B), arguing for the animals' uncertainty about whether the action-only strategy was relevant or not in this task. A different protocol would have been required to investigate the impact of VS- and amygdala-lesions in a case of complete absence of uncertainty about the correct strategy.

[R4.9] We appreciate this observation made by the reviewer. We agree with the reviewer that this distinction is only from the ideal observer and the monkeys are indeed oblivious to the ground truths about the experimental design. We have now clarified our statement to indicate that we are not referring to the monkeys' subjective perception but only to the structure of the experiment. Please note the following changes in the text marked in blue:

(Line #115)

“Overall performance was best during the What-only task, in which only the stimulus identity was predictive of reward, and there was no **inherent** uncertainty about the model of the environment **in terms of task structure**.”

(Line #180)

“Overall, these results demonstrate that in the absence of **intrinsic** uncertainty about the correct model of the environment and when this model was stimulus based, VS-lesioned monkeys (with intact amygdala) were able to partially overcome the deficit in stimulus-based learning to a significantly larger degree than amygdala-lesioned monkeys. Under **additional** uncertainty about the correct model of the environment (the What/Where task), however, VS lesions caused significant impairment in the What blocks only, consistent with the role of VS in stimulus-based learning. ”

(Line #567)

“Considering the observed effects of amygdala and VS lesions on learning and decision-making behavior, we investigated long-term adjustments in these behaviors in the absence of **task-imposed, objective** uncertainty about the correct model of environment.”

(Line #603)

“They suggest that in the absence of **additional** uncertainty about the model of the environment, intact amygdala in VS-lesioned monkeys (and not intact VS in amygdala-lesioned monkeys) enabled these animals to slowly improve their stimulus-based learning over time.”

Further, we have added a following note in **Supplementary Note 1** to clarify that the uncertainty is referring only to the objective point of view (Line #1343):

“Interestingly, there was still some proportion of trials with $ERDS_{Action} < ERDS_{Stim}$ ($8.66\% \pm 1.25\%$) even though action-based learning was irrelevant for performing the task. **Therefore, although the What-only task lacks the task-imposed, objective uncertainty about the correct model of the environment, monkeys still considered this uncertainty and the alternative model of the environment.**”

In the same lines (167-170), it is not true that 'amygdala-lesioned monkeys' were 'not [...] able to partially overcome the deficit in stimulus-based learning.' They did show a mild

increase of performance, both before and after the reversal. It is just that VS-lesioned animals better overcame this deficit. In general, the authors should be careful with their too frequent over-statements about results and interpretations.

[R4.10] We thank the reviewer for pointing out this overstatement in the text. We have now rephrased this statement as follows (Line #182):

“... VS-lesioned monkeys (with intact amygdala) were able to partially overcome the deficit in stimulus-based learning to a significantly larger degree than amygdala-lesioned monkeys.”

Lines 202-213 and in Supplementary Figure S1, the difference between ERDS_Stim and EDS_Action is not sufficient by itself to investigate increases in the adoption of 'competing (incorrect) strategy', (1) because there can be other strategies than these two, e.g. alternation, random, etc.; (2) because plotted variations of the difference could be due either to an increase/decrease in ERDS_Stim only, an increase/decrease of ERDS_Action only, or both. Could the authors also plot ERDS_Stim as a function of uncertainty for What-only and What blocks, as well as ERDS_Action as a function of uncertainty for Where blocks, please?

[R4.11] We thank the reviewer for asking these questions. The other strategies mentioned by the reviewer would equally affect the two types of ERDS and moreover, are not directly relevant to the arbitration between alternative models, which is the focus of our study. The reviewer, however, is correct that the effect based on ERDS difference does not necessarily measure the adoption of “incorrect” strategy per se. To address this issue, we have run additional mixed-effects analysis (including subjects as a random effect and reward uncertainty as a fixed effect) to directly test for the adoption of “incorrect” strategy as a function of reward uncertainty (e.g., ERDS_action in the What blocks). We also created separate plots for ERDS_stim and ERDS_action (see **Supplementary Fig. S2** below).

The results of this analysis indicate that the adoption of incorrect strategy (lower ERDS) is mostly reflected in the What/Where task, while the evidence for the What-only task is weaker. Specifically, in the What/Where task, the reward uncertainty had significant effects on “incorrect” strategy across groups in both block types (second & third columns of **Fig. S2**). In contrast, in the What-only task (first column), the uncertainty (variance) of the block schedule was not predictive of ERDS_action in any group (controls: $\beta_{rew} = -0.052$, $p = .294$; amygdala: $\beta_{rew} = 0.129$, $p = .055$; VS: $\beta_{rew} = -0.098$, $p = .570$). However, it is difficult to definitely conclude here that ERDS_stim is solely responsible for the observed effect in $\Delta ERDS$ ($ERDS_{stim} - ERDS_{act}$), as controls and VS group exhibit significant modulation of ERDS_stim by ERDS_loc in this task (controls: $\beta_{ERDS(Act)} = -0.676$, $p = 2.17e-26$; VS: $\beta_{ERDS(Act)} = -0.380$, $p = 2.25e-32$; mixed-effects analysis, predicting $ERDS_{stim}$ with $ERDS_{act}$ and *reward uncertainty*). That is to say, the decrease of stimulus-based strategy could be also be partially driven by increase in the action-based strategy ($ERDS_{action}$), and vice versa, as reflected in the overall

correlation between two ERDS (control: $r = -0.123$, $p = 4.29e-7$; amyg: $r = -0.180$, $p = 1.09e-12$; VS: $r = -0.455$, $p = 4.71e-47$).

Therefore, based on these results, we think that the difference in two ERDS ($\Delta ERDS$) is the more appropriate quantity for examining the adoption of two strategies on a relative scale, considering the competitive interaction between the two learning systems. Nonetheless, we have revised our interpretation of the results to focus solely the effect of reward uncertainty on the relative adoption of two strategies, rather than of making claims about the adoption of “incorrect” strategy (Line #219):

“We found that the reward uncertainty (measured as variance¹³) is predictive of the relative degree of adoption between stimulus- and action-based strategies. More specifically, in the What-only and What blocks of the What/Where task, animals’ strategies became relatively more biased toward action-based (increasing $\Delta ERDS$) as the uncertainty of the reward schedule increased (Supplementary Fig. S3d, e). Consistently, in the Where blocks, they tended to become relatively more stimulus-based under more uncertainty (decreasing $\Delta ERDS$; Supplementary Fig. S3f).”

Figure S2. Distributions of conditional entropy of reward-dependent strategy (ERDS) by block type and reward schedule. Distributions of ERDS based on stimulus identity ($ERDS_{stim}$, in solid lines) or action ($ERDS_{action}$, in solid lines) computed from each block of trials during the What-only (first column) and the What/Where task (second and third columns). Insets represent mean values (Error bars = SEM across subjects). Asterisks indicate significant effect of reward uncertainty on ERDS, using mixed-effects analysis with subjects as a random effect (*: $p < .05$, **: $p < .01$, ***: $p < .001$). Reward uncertainty of each block was measured as the variance of outcome equal to $p_{Better}^*(1 - p_{Better})$. (a–c) $ERDS_{stim}$ (in solid

lines) and $ERDS_{Action}$ (in dotted lines) in control monkeys during the What-only (a) and the What/Where tasks (b,c). Colors indicate reward schedules (black: 80/20; dark gray: 70/30, light gray: 60/40). (d–f) $ERDS$ in amygdala-lesioned during the What-only (d) and the What/Where tasks (e,f). Colors indicate reward schedules (brown: 80/20; red: 70/30, orange: 60/40). (g–i) $ERDS$ in VS-lesioned during the What-only (g) and the What/Where tasks (h,i). Colors indicate reward schedules (navy: 80/20; blue: 70/30, cyan: 60/40).

Lastly, please see below for the revised figure captions for **Fig. S3**. Here we note that increase in one $ERDS$ could be related to the decrease in the other $ERDS$ as suggested by the negative interaction between the two:

Figure S3. (d–f) Summary results for the panels in a–c. Plotted are mean values of $\Delta ERDS = ERDS_{Stim} - ERDS_{Action}$, by each group and reward schedule during What-only (d) and What/Where (e,f) tasks. With larger reward uncertainty, animals' strategies became relatively more biased toward the incorrect strategy (increasing $\Delta ERDS$ in the What blocks and decreasing $\Delta ERDS$ in the Where block). Asterisks next to the plots indicate significant effect of reward uncertainty (variance) for the respective group indicated by colors, as determined by mixed-effects analysis (including reward variance as fixed effect and subjects as a random effect). (*: $p < .05$, **: $p < .01$, ***: $p < .001$). It is worth noting that, while $ERDS_{Action}$ was not explicitly modulated by reward uncertainty during the What-only task for any of the groups (**Supplementary Fig. S2a, d, g**), competitive interactions between $ERDS_{Stim}$ and $ERDS_{Action}$ still exist for the task. This suggests that $ERDS_{Stim}$ cannot be solely responsible for the observed effect in $\Delta ERDS$.

Lines 627-640 of the Discussion, the argument according to which an 'arbitration weight that biases behavior toward the action-based system' could be an explanation for increased flexibility to reversal in stimulus-based tasks does not stand: The reason is that such a bias predicts a lower performance than controls before the reversal, which is not the case in the cited studies (ex : Izquierdo et al. 2013).

[R4.12] We thank the reviewer for raising this concern. We would like to clarify that our model does not make a general prediction about the task performance, particularly regarding the extent to which the initially action-biased arbitration weight reduces performance. As shown in the simulation results (**Fig. 5a-d** reproduced below), the specific dynamics of the task performance depends on the particular set of parameters (including task parameters such as the reward schedule, length of reversals, etc.). In particular, for **Fig. 5b**, which illustrates the case of how the reversal could be facilitated, it can be seen that for many of the initial arbitration weights, the performance saturates quickly. Once the performance saturates, which was the case for the Izquierdo et al.

(2013), it could be difficult to detect significant differences in the performance. Moreover, there are many traces in **Fig. 5a**, which could correspond to the control group in Izquierdo et al.'s study, showing the same level of performance as in **Fig. 5b**, before the reversals. Thus, rather than replicating the observed effects with the exact setup of previous experiments which is beyond the scope of this study, our goal was to provide a theoretical framework that could address how the conflicting findings could have arisen in the literature.

Therefore, we think that our simulation results are consistent with the observations reported in the cited study, although the exact details of the experimental setup differ from the cited experiments. Nonetheless, to better clarify the connection to the simulation results in **Fig. 5b**, we have now explicitly referred to this panel in the **Discussion**, in addition to a statement about the saturation effects before reversal (Line #711):

“Through simulations of our best-fitting model with dynamic interaction between two systems, we found that in certain situations, the lower initial arbitration weight that biases behavior toward the action-based system can actually facilitate adjustments to reversals during stimulus-based learning, especially if the performance has saturated before reversal (**Fig. 5b**).”

Figure 5. Complex interaction between arbitration and learning gives rise to diverse behavioral patterns.

ABUSE OF LANGUAGE

The employed terminology sounds often like an overkill. For instance, why talking about a 'model of the environment', while it is just about the action reference frame (stimulus- vs. space-based)? The authors themselves nuance their terminology in the discussion, stating that a variety of arbitration models exist, and that their model includes a 'more

basic arbitration'. I think they should simplify the terminology from the very beginning of the paper, including title, abstract and introduction, so as not to disappoint readers expecting to find an arbitration mechanisms between different internal world models.

[R4.13] We thank the reviewer for this comment. The “model of the environment” is a commonly used phrase in the reinforcement learning literature to refer to anything that an ‘agent’ can use to predict how the ‘environment’ will respond to performed actions (Sutton & Barto, 2018, Chapter 8). Hence the phrase denotes something more general and abstract than the “internal world models,” and we believe that many readers who are familiar with the reinforcement learning framework would not interpret the term in the latter direction. Nevertheless, the stimulus-based and action-based models can be viewed as internal models of the world, as they “predict” reward outcomes based on stimulus identity or actions.

The only place in the manuscript where we had referred to “internal model” is the first paragraph of the **Introduction**, where we made a generic statement about the existing hypothesis in the literature: *“It has been suggested that the brain tackles such uncertainty by running multiple internal models of the environment, each predicting outcomes based on different attributes of choice options, and using the reliability of these predictions to select the appropriate model to inform choice behavior¹⁻³.”* Here, our intention was not to imply that the stimulus- and action-based learning systems in our RL models correspond to the “internal world models.” Nonetheless, to avoid unnecessary confusion, we have updated the abstract and replaced the term “internal” in the quoted sentence as follows (Line #49), where we had previously referred to “multiple internal models of the environment.”

“It has been suggested that the brain tackles such uncertainty by running multiple predictive models of the environment, each predicting outcomes based on different attributes of choice options, and using the reliability of these predictions to select the appropriate model to inform choice behavior¹⁻³.”

Furthermore, while we acknowledge that the distinction between stimulus- and action-based strategy certainly includes the distinction based on the reference frame of decision , there is an additional, more important learning component in the utilized task. If these tasks were simple decision-making tasks based on different streams of information, then the distinction would focus entirely on the action reference frame. However, performing the What/where tasks require learning to predict reward based on stimulus or action values. As these two distinct “learning systems” track different attributes of the environment and therefore make distinct predictions about reward outcomes for a given choice option, it is natural to assume that those predictions would be weighted based on the reliability of those predictions (corresponding to the arbitration weight in our RL model). This approach is commonly referred to as a “mixture of experts” (as noted in ref #2, O’Doherty et al., 2021), and the empirical evidence for this framework is well established in the literature (O’Doherty et al., 2021; Soltani & Koehlin, 2022). Some of its core assumptions include the idea that separate value predictions exist for distinct systems (e.g., Lee et al., 2014),

that the reliabilities of distinct strategies are represented in the brain (e.g., Charpentier et al., 2020), and that a separate “manager” or arbitrator over the experts modulates the weights of each expert (e.g., Weissengruber et al., 2019).

We believe that our framework is fully in accordance with terminology in the field, as our RL model includes separate stimulus- and action-based systems which make different value predictions (V_{Stim} , V_{Act}), and the reliability of these two systems are compared to assign bigger weight to the more reliable system (arbitration weight). Indeed, many of the “arbitration” models described in the cited studies have utilized a similar convention of assuming two distinct value systems, weighted linearly by a complementary weight parameter like ω (e.g., Lee et al., 2014; Dorfman et al., 2019; Miller et al., 2019; Wurm et al., 2024; Charpentier et al., 2020; Philippe et al., 2024), and each expert is commonly referred to as a “predictive model” or “mixture of models.” As such, our intention was to connect our work to these existing studies that employ a similar framework and terminology.

The authors are also abusing language when citing the studies of Birrell & Brown, Ragozzino et al., Floresco et al., as if their task made it easy to find the more rewarding option: 'In other paradigms, the more rewarding option is easily ascertained, but there is uncertainty about the correct model of the environment and when to choose that option'⁸⁻¹⁰.

[R4.14] We thank the reviewer for this comment and apologize for the misrepresentation of the cited works. By no means we intended to oversimplify their experimental paradigms as being easy or simple to run. Instead, our intention was simply to highlight the deterministic nature of the reward schedule in those studies, where the “correct” option, once chosen, consistently led to a rewarding outcome without any uncertainty. This was not the case in our experimental paradigm, as the probabilistic reward schedule (80/20, 70/30, or 60/40) adds expected uncertainty due to the probabilistic nature of reward outcome. In contrast, in the work by Birrell & Brown (2000), rats were trained to retrieve food from the digging bowls, one of which had the food reward physically (i.e., deterministically) hidden based on the specified dimension (odor, texture, and digging medium). Similarly, the studies by Ragozzino et al. (1999) and Floresco et al. (2006), which utilized cross-maze apparatus to induce set-shifting between response vs. stimulus-based strategies, did not involve probabilistic reward outcome as rats could reliably obtain the food rewards, given that they approach the correct place specified by the current rule. The tasks used in our study also involve unexpected uncertainty due to un-signaled reversals occurring between trials 30 and 50.

To clarify these points, we have revised the quoted passage and other two sentences in the **Introduction** as follows (Line #58, #77). Moreover, to provide additional examples of the studies employing a similar combination of uncertainty as the cited works (i.e., shifting rules of the environment and deterministic reward schedule), we have added two more references below (ref #11-12; analog of Wisconsin Card Sorting Task in monkeys).

“In other paradigms, the correct option is deterministically linked to reward outcomes, but there is uncertainty about the correct model of the environment and when to choose that option⁸⁻¹². Few if any studies have manipulated expected and unexpected uncertainty¹³ about stimulus-action-outcome relationships in conjunction with uncertainty about which model of the reward environment is currently relevant.”

“These included uncertainty about the better option on a given trial (expected uncertainty), uncertainty about the correct model of the environment, and uncertainty about when reward associations change (unexpected uncertainty), thus creating a challenging task that could reveal the role of the amygdala and ventral striatum in all three of these processes.”

11. Mansouri, F. A., Matsumoto, K. & Tanaka, K. Prefrontal Cell Activities Related to Monkeys' Success and Failure in Adapting to Rule Changes in a Wisconsin Card Sorting Test Analog. *J. Neurosci.* **26**, 2745–2756 (2006).
12. Buckley, M. J. *et al.* Dissociable Components of Rule-Guided Behavior Depend on Distinct Medial and Prefrontal Regions. *Science* **325**, 52–58 (2009).

Overall, the way to present the main results constitutes itself overselling, due to the multiple limitations mentioned above about the computational analyses. I think the authors should either modify their claim that they have an implementation model of amygdal function, and rather refer to it as a descriptive model which can be useful to identify latent variables and describe things that happen in the monkey. Alternatively, if the authors want to capture the function of the amygdala, they should not just compare a set of arbitration models, but rather also test alternative models, including simpler models where a single mechanism produces different adaptations to different environmental changes (e.g., metaplasticity, or Wang, Kurth-Nelson and Botvinick's meta reinforcement learning model), instead of models requiring two mechanisms in parallel.

[R4.15] As explained above in response to each of the reviewer's criticisms, we believe our work provides significant results and insights into computational and neural substrates of arbitration between stimulus- and action-based learning, and that we do not exaggerate or oversell our findings.

We were unsure which part of the manuscript the reviewer is referring to in the comment about “[our] claim that [we] have an implementation model of amygdala function,” as this is indeed not what we are claiming. The term “implementation” appears only once in the **Introduction**, where we have made a general statement regarding the current challenges without specific implications for our models (Line #53): “*Although there exist many conceptual and algorithmic solutions to model arbitration²⁻⁴, confirming implementation level details in terms of the operation of neural circuits has remained a challenge due to several factors.*”

Moreover, RL models can serve as both “descriptive” and “mechanistic” models of the behavior, in line with the typical use of these terms in the literature (e.g., Levenstein, ..., Redish, 2023 *JNeurosci*). Specifically in the context of our study, our RL models specify

the underlying mechanism of how distinct learning systems interact with each other and are arbitrated based on their reliability. Using this model, we have aimed to reveal the unexplored role of amygdala by pinpointing the mechanism of the deficits due to lesions, as the result of the interplay between learning and arbitration.

As addressed in more detail in our response above ([R4.5]), in our framework the “artificial lesions” have been captured through modulation of the key parameters related to arbitration. Therefore, the main claim of our study is that the specification of the interaction between two distinct systems is needed to account for the observed behavioral patterns with respect to the adoption of two competing strategies.

Finally, frameworks such as meta-RL (e.g., Wang, ..., Botvinick 2018 *Nat. Neuro*; Grossman, Bari, & Cohen 2020, *Current Biology*) or metaplasticity (e.g., Farashahi, ..., Soltani 2017, *Neuron*) are not intended to address interaction between distinct learning systems that learn about distinct attributes of the environment; rather, they are intended to address how the learning signal within a system (primarily focused on a single attribute) can be flexibly modulated by additional mechanisms. Therefore, we believe that our research question in the current study is better investigated from the perspective of the “mixture of experts” approach using RL (see [R4.13]), which offers mechanistic insight into the nature of interaction between distinct learning systems.

MINOR POINTS

When looking at reaction times in the What-only task, and observing that 'most action-dominant trials happened when reward value estimates based on the two systems were close to each other, resulting in slower and more erroneous responses', I think it's important to state (both in manuscript and supporting information) that the higher error rate may be a possible confound when interpreting RT difference between stimulus- and action-dominant trials. This is especially true given that the authors themselves wrote in Supplementary Material that 'performance was higher for the correct strategy'.

[R4.16] We thank the reviewer for pointing out this possible confound in our analysis. In relation to the reviewer's another comment below (**[R4.17]**), we have performed additional control analyses in **Supplementary Note 1** by restricting trials to the later portion of each block (last 20 trials) when the performance has reached a plateau. Furthermore, to control for other potential confounds and address subject variability within each group, we adopted mixed-effects regression models to identify the unique effect of the dominant strategy from other factors. Specifically, the mixed-effects models include random effect of subjects and the following predictors as fixed effects: dominant strategy coded as stimulus-dominant (0) or action-dominant (1), whether the monkey has chosen the better option (1) or not (0), reward schedule or uncertainty (coded as variance), trial number within a block, block number within a session, and session number within the subject. We also included interaction between dominant strategy and choice of better option, as the latter could depend on the adopted strategy. Variables were normalized by each subject to yield comparable standardized regression coefficients. Overall, these additional predictors were included to account for the confounding effects on RT.

Using this setup, we inspected the regression coefficients for the main effect of dominant strategy, which is positive if action-dominant trials predict longer RT and negative if predicting shorter RT. Compared with the previous results from the rank sum test, we note that the significance effect has changed only for controls and amygdala groups in the What-only task (**Fig.SN1a**: previously significant; **Fig.SN1d**: previously insignificant; see updated figure attached below). Notably, we found a consistent trend where action-dominant trials were associated with faster RT.

If error trials (in which the monkey adopts the "incorrect" strategy) are a significant factor in shortening the RT, this would predict that What blocks have shorter RT for action-dominant trials, and Where blocks have shorter RT for stimulus-dominant trials, corresponding to the error trials in each case. However, we found that action-dominant trials were consistently faster in Where blocks, where the correct strategy was action-based. These results are intuitive considering that the monkeys can prepare the left/right movement early on to make a decision based on actions, whereas a stimulus-based decision requires the animals to identify the target and then plan the movement. We believe this could be the main reason why action-dominant trials tend to have shorter RTs, largely mitigating the confounds from the error rates.

Based on results, we have revised **Supplementary Note 1** as follows (Line #1326):

"Supplementary Note 1

Considering previous findings on faster decision making during action-based compared with stimulus-based tasks²⁴, we hypothesized that reaction time (RT) on a given trial depends on the learning system that controls the behavior more strongly on that trial. To test this hypothesis, we categorized trials as either stimulus- or action-dominant by directly comparing $ERDS_{Stim}$ and $ERDS_{Action}$ (computed from a moving window of ten trials). Trials that had $ERDS_{Stim} < ERDS_{Action}$ were categorized as stimulus-dominant, and action-dominant if $ERDS_{Action} < ERDS_{Stim}$. To assess whether the dominant strategy had a significant effect on RT, we performed mixed-effects analysis and identified the unique effect of the dominant strategy from other potential confounds. Specifically, the mixed-effects model

included random effect of subjects and the following predictors as fixed effects: dominant strategy, coded as stim-dominant (0) or action-dominant (1), whether the monkey has chosen the better option (1) or not (0), reward schedule or uncertainty, trial number within a block, block number within a session, and session number within the subject. We also included interaction between dominant strategy and choice of better option, as the latter could depend on the adopted strategy. Variables were normalized for each monkey to yield comparable standardized regression coefficients.

For the What-only task in control monkeys, the vast majority of the analyzed trials (79.4%) were classified as stimulus-dominant (**Fig. SN1a**). Interestingly, there was still a proportion of trials with $ERDS_{Action} < ERDS_{Stim}$ ($8.66\% \pm 1.25\%$) even though action-based value/strategy was irrelevant for performing the task. Therefore, although the What-only task lacks the task-imposed, objective uncertainty about the correct model of the environment, monkeys still considered this uncertainty and the alternative model of the environment. **[R4.9]** The mixed-effects analysis indicated that the dominant strategy was not a significant predictor of RT ($\beta_{Act-Dominant} = -0.0181$, $p = .279$; $M \pm SEM$ across subjects for stimulus-dominant RT: 223.8 ± 29.5 ; action-dominant RT: 225.2 ± 34.9). To compare the performance for these two types of trials, we ran a similar generalized mixed-effects model predicting animals' choice of better option (binomial distribution with logit link function), with the same random and fixed effects. Using this analysis, we found that action-dominant trials predicted significantly lower performance ($\beta_{Act-Dominant} = -0.708$, $p = 3.54 \times 10^{-238}$), as reflected in the $P(Better)$ of action-dominant trials (0.621 ± 0.010) and stimulus-dominant (0.790 ± 0.016) trials. These results suggest that most action-dominant trials happened when action values were used instead of stimulus values to make decisions, resulting in more erroneous responses.

For the What/Where task, the proportions of stimulus-dominant and action-dominant trials were reflected in the respective block types: the What blocks were marked by a higher proportion of stimulus-dominant trials ($59.5\% \pm 5.99\%$; **Fig. SN1b**), whereas the majority of trials within the Where blocks were categorized as action-dominant ($63.6\% \pm 6.41\%$; **Fig. SN1c**). Critically, performance was higher for the correct strategy: in the What blocks, performance was significantly higher for stimulus-dominant (0.820 ± 0.035) than action-dominant (0.607 ± 0.019) trials ($\beta_{Act-Dominant} = -0.959$, $p = 4.94 \times 10^{-324}$), whereas in the Where blocks, $P(Better)$ was higher for action-dominant (0.791 ± 0.025) than for stimulus-dominant (0.597 ± 0.018) trials ($\beta_{Act-Dominant} = 0.846$, $p = 4.94 \times 10^{-324}$). In terms of RT, we found that for both block types, action-dominant strategy predicted significantly shorter RT ($\beta_{Act-Dominant} = -0.0926$, $p = 1.38 \times 10^{-27}$). Consistently, categorization of trials based on comparison of ERDS (stimulus-dominant vs. action-dominant) yielded a larger distinction in RTs than categorization simply based on block type (What vs. Where), as reflected by the smaller proportion of variance explained by block type predictor (R^2 equal to 0.005 and 0.0081 for block type and dominant strategy; comparison of partial R^2 values). These results show that entropy-based metrics could be used to identify the adopted model on a given trial and that RT reflected the adopted strategy, with the stimulus-based strategy yielding consistently longer RT than the action-based strategy. This could be explained by the fact that the monkeys can prepare the left or right movement early on to make decisions based on actions, whereas a stimulus-based decision requires the animals to identify the target and then plan the movement only after the stimuli appear. We note that the results remain qualitatively the same when the analyses are restricted to the last 20 trials of each block, where performance has plateaued.

In brain-lesioned monkeys, the effect of the dominant strategy on RT was more consistent across two tasks, with the action-dominant strategy significantly predicting shorter RTs (**Fig. SN1d-i**). Furthermore, the correct strategy for a given block type predicted significantly higher performance. That is, action-dominant strategy predicted lower performance during the What-only task (amygdala:

$\beta_{\text{Act-dominant}} = -0.230$, $p = 1.35 \times 10^{-63}$; VS: $\beta_{\text{Act-dominant}} = -0.459$, $p = 1.65 \times 10^{-143}$) and What blocks of What/Where task (amygdala: $\beta_{\text{Act-dominant}} = -0.665$, $p = 8.26 \times 10^{-308}$; VS: $\beta_{\text{Act-dominant}} = -0.320$, $p = 701 \times 10^{-52}$), whereas it predicted higher performance in the Where blocks of What/Where task (amygdala: $\beta_{\text{Act-dominant}} = 0.546$, $p = 1.01 \times 10^{-152}$; VS: $\beta_{\text{Act-dominant}} = 0.550$, $p = 1.93 \times 10^{-101}$).

Figure SN1. Distributions of reaction time (RT) for stimulus-dominant and action-dominant trials, separately for different tasks and block types. Each trial was categorized as either stimulus- or action-dominant by comparing $ERDS_{\text{Stim}}$ and $ERDS_{\text{Action}}$. Percentages indicate proportions of trials under each category, reported as mean \pm SEM across subjects (remaining percentages correspond to trials where both strategies were equally dominant). (a–c) RT data from control monkeys during the What-only (a) and the What/Where tasks (b,c). (d–f) RT data from amygdala-lesioned monkeys during the What-only (d) and the What/Where tasks (e,f). (g–i) RT data from VS-lesioned monkeys during the What-only (g) and the What/Where tasks (h,i). Circles in the violin plots represent means and black horizontal lines represent medians of the distributions. Reported are standardized regression coefficient and corresponding p-values for the main effect of action-dominant strategy, with negative (positive) values indicating that action-dominant trials predict shorter (longer) RT. In the What/Where task (b–c, e–f, h–i), RT was consistently shorter for action-dominant trials across all groups.

To further investigate the effect of strategy on RT in the What/Where task, the authors should restrict their analysis to the last third of each block, where performance has reached a plateau, and at this stage see whether there is a different in RT between stimulus-dominant and action-dominant strategy.

[R4.17] To address this question, we have performed the same mixed-effects analysis as in the previous comment ([R4.16]), but restricting the trials to the last 20 trials of each block (last 25%) where the performance has reached a plateau. The plots for these results are attached below, which are consistent with the above results that include all trials. Please note this point has been noted in the updated **Supplementary Note 1** above (Line #1377).

“We note that the results are qualitatively the same when the same analyses are restricted to the last 20 trials of each block where the performance has reached a plateau.”

Figure R4.17. Control analysis, restricted to the last 20 trials of each block.

There is already an old literature on spatial and object reversal learning, which is not cited here. See for instance Helen Mahut (1971) Spatial and object reversal learning in monkeys with partial temporal lobe ablations. Neuropsychologia. The paper also investigates the effect of amygdala lesions and should thus be discussed here.

[R4.18] We appreciate the reviewer for bringing this work to our attention. Along with another earlier review article (ref 60-61 below), we have cited these studies as foundational investigations into the role of the amygdala across various behavioral paradigms (Line #708). However, we have refrained from discussing the specific results of the study, as its primary emphasis was on other brain regions, and its use of aspiration lesions predates the precision afforded by MRI-guided surgery. As the reviewer is likely aware, aspiration lesions can damage fibers of passage, whereas the excitotoxins used in the current manuscript to create lesions of the amygdala and ventral striatum are fiber sparing. For the past twenty years, Betsy Murray and colleagues have demonstrated the importance of acknowledging this difference and how it can lead to different conclusions about the necessity of a particular brain region in performing specific computations (e.g., Rudebeck et al., 2013. *Nat Neurosci*; Murray, 2025. *Hippocampus*).

“While earlier lesion studies have attributed varying degrees of behavioral deficits to amygdala^{55,56}, its role in instrumental learning has since been a matter of debate due to mixed evidence both in favor of^{17–19,57,58} and against^{14,15,59–61} its involvement in reward learning.”

55. Mahut, H. Spatial and object reversal learning in monkeys with partial temporal lobe ablations. *Neuropsychologia* **9**, 409–424 (1971).

56. Sarter, M. & Markowitsch, H. J. Involvement of the amygdala in learning and memory: A critical review, with emphasis on anatomical relations. *Behav. Neurosci.* **99**, 342–380 (1985).

RESPONSE LETTER TO THE REVIEWERS' COMMENTS AND SUMMARY OF REVISIONS

Title: Contribution of amygdala to dynamic model arbitration under uncertainty

Authors: Woo, Costa, Taswell, Rothenhoefer, Averbeck, & Soltani

We thank the four reviewers for their feedback on our revised manuscript and trust that our responses below, along with the corresponding revisions, address their remaining concerns.

REVIEWER COMMENTS

Reviewer #1 (Remarks to the Author):

About the manuscript:

All my comments were addressed in the revision. I have no further suggestions.

About the code:

I cloned the repo, but the demo does not work, even after including all the dirs in the matlab path.

Reviewer #1 (Remarks on code availability):

I cloned the repo, but the demo does not work, even after including all the dirs in the matlab path.

error:

Unable to find file or directory
'dataset/preprocessed/all_stats_control.mat'.

We thank the reviewer for the feedback and apologize for the error in the codes. All the necessary files are now placed in the correct folders.

Reviewer #2 (Remarks to the Author):

The authors have made substantial updates in response to the comments from the reviewers and my comments have been sufficiently addressed.

We thank the reviewer for the feedback and happy to hear that all of their comments have been addressed.

Reviewer #3 (Remarks to the Author):

The authors have engaged with the concerns raised in my previous review and appear to have made a good faith effort to correctly take cross-subject variability into account in both their statistical analyses and visual presentation of the data.

The updated versions of the plots now showing cross subject SEM error bars are a great improvement, thanks for doing this. The diagram showing the flow of animals through the experiment is also really useful.

The move to using mixed effects models for most statistical tests is sensible, as it provides a principled way to take cross-subject variability into account while maximising statistical power. However, many aspects of how these analyses were setup are not clear from the text, which makes it hard to interpret the reported stats and to tell whether the analyses have been set up appropriately.

Thank you for the positive evaluation of our efforts to improve the manuscript. We sincerely appreciate the further detailed feedback and insight provided by the reviewer. We have carefully reviewed and revised the regression analyses in response to the reviewer's feedback. To ensure consistency, we now use the same reference categories for all categorical variables across all analyses. Full details are provided in the new **Supplementary Table 3**.

Importantly, in our effort to further improve the accuracy of the mixed-effects models and reduce the risk of false positive findings, we have substantially revised the random effects structure in our regression analyses as follows. While our previous analyses included subject-level random intercepts and random slopes for some variables of interest, we have now added random slopes for the effect of time. More specifically, we consider subject-level random slopes for: (i) the percentage of sessions completed so far, capturing long-term adjustment (Cf. **Fig. S8**); and (ii) the number of blocks within a session, capturing the drift in performance or other measures (e.g., due to fatigue or adaptation). We made this decision based on the existing recommendations in the literature on mixed-effects modeling (e.g., Barr et al., 2013; Matuschek et al., 2017), which suggest that including appropriate random effects can reduce the risk of falsely attributing the unexplained variability to the fixed effects (Type-I error). Indeed, we observe that including these two random slopes significantly improves the fit to the data in terms of AIC or log-likelihood test. We only considered their random slopes and not the fixed effects (except when we explicitly consider their main effect), as these are not the main predictors of our interest.

Furthermore, for better reproducibility, we have uploaded the custom analysis codes and the processed block-wise data necessary to reproduce the regression analyses we report in the manuscript. These can be found in the same repository (script name 'Reproduce_mixed_stats.m') at: https://github.com/DartmouthCCNL/woo_etal_amygdala.

Please see below for our detailed point-by-point responses.

To illustrate this, I will provide some examples of reported results:

Line 161: “amygdala-lesioned monkeys exhibited the largest impairment in performance ($P(\text{Better})$) across all three reward schedules: $M \pm SD = 0.581 \pm 0.11$; main effect of group in mixed-effects analysis; $\beta_{\text{VS}} = 0.054$, $p = .0061$; $\beta_{\text{control}} = 0.173$, $p = 3.25 \times 10^{-21}$; Fig. 1e). VS-lesioned monkeys also showed impairment compared to the control monkeys (main effect of group in mixed-effects analysis; $\beta_{\text{VS}} = -0.120$, $p = 1.97 \times 10^{-12}$; Fig. 1e).”

It is unclear from the above how the reported β_{VS} and β_{control} demonstrate that amygdala lesioned monkeys exhibited the largest impairment in performance. It is also unclear why are two different values reported for β_{VS} (0.054 and -0.120), for what is apparently the same analyses of the same figure’s data (main effect of group in mixed-effects analysis, Fig. 1e). The reporting of beta weights and P values associated with the control group (β_{control}) appears inconsistent with the statement at line 1138 in the methods stating: “We tested this mixed-effects model on each of the parameters compiled across groups and tasks, with control monkeys as the reference group”, because the reference group for a categorical variable in a regression typically does not have a beta associated with it.

My best guess at what the authors have done here is that they have run two different regression analyses of the same data using different groups as the reference category, in order to test for different things. For the first reported stats ($\beta_{\text{VS}} = 0.054$, $\beta_{\text{control}} = 0.173$) I assume the amygdala group has been chosen as the reference category such that P values for β_{VS} and β_{control} test for significant differences between these groups and the amygdala group. For the second reported stats ($\beta_{\text{VS}} = -0.120$) I assume the control group have been chosen as the reference category such that the p value for β_{VS} indicates whether the VS group are significantly different from the controls.

If this is correct, the decision to test for differences between the lesion groups by using the amygdala group as the reference group is not ideal because (a) it is an arbitrary choice – why use the amygdala group rather than the VS group, and (b) it is confusing for the reader to report p values for β_{VS} and β_{control} in order to say something about the amygdala group. To test for differences between the two lesion groups it would be preferable to either i) Run a regression with one predictor coding for the difference between control and both lesion groups and a second predictor coding for the difference between the two lesion groups. Or ii) Run a regression with the control group as the reference group and a predictor for each lesion group, then compute the contrast between the lesion groups from the output of this regression (I think the `emmeans` package in R provides functionality to do this, see e.g. <https://www.r-bloggers.com/2019/03/getting-started-with-emmeans/>).

[R3.2.1] We appreciate the reviewer’s detailed feedback on our use of mixed-effects. Yes, the reviewer is correct regarding the statistics reported in Line #161: amygdala group was selected as the reference group in the first stats, and control group in the second stats. We previously chose this approach because it allowed us to directly present the beta coefficients for each verbal comparison—for example, showing how the amygdala group differs from the control and VS groups in one case, and how the VS group is impaired relative to controls in the other.

However, we agree that this choice can be perceived as rather arbitrary and unclear from the readers’ point of view. Following the reviewer’s suggestion, we have now addressed this by running a single regression model (with control monkeys as the reference group) and computing the contrast between the lesion groups manually afterward. We have adopted this scheme for all other mixed-effects models as well. Please see the table below for a summary of the updated analysis, now included as **Supplementary Table 3.1**. The degrees of freedom (DF) now reflect the block-wise data compiled across all three groups of monkeys.

Supplementary Table 3. Summary of the main mixed-effects regression analyses.

Reported below are the list of regression models, along with their hypotheses and corresponding results. Shaded gray color indicates statistics relevant to each hypothesis reported in the text. Control monkeys were selected as the reference group wherever group comparison has been made. For the *blockType* variable in the What/Where task, ‘What’ block was used as the reference category. For the random effects, we included subject-level intercept and random slopes for the long-term adjustment effect, across the entire experiment and within each session. The variable *sess_perc* represents the proportion of the sessions completed and was mean-centered within each subject. Similarly, *block_in_sess* refers to the block number within a session and was also mean-centered within each subject

Supplementary Table 3.1. Comparison of performance across the three groups during the What-only task.

Model#	Block-wise P(Better) ~ group + (1 + sess_perc + block_in_sess subject) #				
Hypothesis #	Performance is significantly reduced in the brain-lesioned group.				
Coeff. #	Estimate	SE	t-stat	DF	P-value
Intercept#	0.7567	0.012535	60.367	4099	0
group_amygdala#	-0.17156	0.017763	-9.6585	4099	7.7214e-22
group_VS#	-0.095504	0.019631	-4.8651	4099	1.1871e-06
Planned contrasts#	Difference between amygdala- and VS-lesioned groups				
group_amyg – group_VS#	-0.076055	0.019662	-3.868	4099	0.00011142

We have updated these results in the main text accordingly (in Line #161):

During the What-only task, amygdala-lesioned monkeys exhibited the largest impairment in performance (*P(Better)*) across all three reward schedules: $M \pm SD = 0.581 \pm 0.11$; main effect of group in mixed-effects analysis; $\beta_{amyg} = -0.172, p =$

7.72×10^{-22} ; contrast between lesion groups: $\beta_{\text{amyg}} - \beta_{\text{VS}} = -0.076$, $p = 1.11 \times 10^{-4}$; **Fig. 1e; Supplementary Table S3.1**). VS-lesioned monkeys also showed impairment compared to the control monkeys (main effect of group in mixed-effects analysis; $\beta_{\text{VS}} = -0.096$, $p = 1.19 \times 10^{-6}$; **Fig. 1e; Supplementary Table S3.1**).

Another example:

Line 170: “During the What blocks of the What/Where task, however, amygdala-lesioned monkeys performed significantly better than VS-lesioned monkeys (mixed-effects analysis with group \times block type; $\beta_{\text{amyg:What}} = 0.0855$, $p = 1.67 \times 10^{-12}$; Fig. 1f inset), while both lesioned groups were impaired relative to control monkeys (amygdala: $\beta_{\text{amyg:What}} = -0.0679$, $p = 3.48 \times 10^{-52}$; VS: $\beta_{\text{VS:What}} = -0.180$, $p = 7.94 \times 10^{-7}$). In the Where blocks that did not require stimulus-based learning, only amygdala-lesioned monkeys showed impairments in performance relative to controls ($\beta_{\text{amyg:Where}} = -0.0212$, $p = 4.48 \times 10^{-6}$; Fig. 1g inset). VS-lesioned performance was comparable to that of controls ($\beta_{\text{VS}} = -0.00871$, $p = .811$) and better than that of amygdala-lesioned monkeys ($\beta_{\text{amyg:Where}} = -0.0361$, $p = .00298$), suggesting no deficits in action-based learning for the VS lesioned monkeys.”

How do stats for predictor $\beta_{\text{amyg:What}}$ test for differences between amygdala and VS groups if the reference groups is the control group as stated in the methods? Why are two different beta reported for the same predictor ($\beta_{\text{amyg:What}} = 0.0855$, $\beta_{\text{amyg:What}} = -0.0679$)? Were separate regressions run for the different block types or were multiple block types included in the regression with block type as a categorical predictor? If the latter then how was the block type predictor coded? Did the random effects structure account for potential differences between subjects in the effect of block type, or just consider a random intercept?

[R3.2.2] We thank the reviewer for asking these clarifying questions. The unclarity here is largely due to following the same scheme as above; i.e., constructing separate regression models for each comparison and selecting different reference categories based on the comparison being made. We have now revised these statistics as described below.

There were two separate beta coefficients for $\beta_{\text{amyg:What}}$ above, because they came from two different regression models: the first comparison was between VS and amygdala (with VS as the reference group), and the second comparison was with respect to controls (with control as the reference group). As noted in the text (*mixed-effects analysis with group \times block type*), all of these regression models had considered interaction with block type as a categorical variable — this was coded with reference coding, again selecting each block type (“What” or “Where”) as the reference category based on the block type being compared. For example, when comparing between controls and the lesion groups during What blocks, “What” was selected as the reference category, such that baseline comparison is made with respect to the performance of controls during “What” blocks. Then, we simply looked at the main effects of each lesion group as these correspond to

the effect of “What” blocks of each lesion group. We may have introduced additional confusion by labeling the coefficients as $\beta_{\text{amyg:What}}$ or $\beta_{\text{VS:What}}$ rather than simply β_{amyg} or β_{VS} , in an incorrect attempt to emphasize the block type. Moreover, these mixed-effects models had included just a random intercept for the subject, which we have now revised to include random slopes as well.

Based on the reviewer’s suggestion, we have now updated regression analyses as follows, and included as **Supplementary Table 3.2** in the manuscript:

Supplementary Table 3.2. Comparison of performance across the three groups during the What/Where task.

Model#	Block-wise P(Better) ~ group*blockType + (1+ blockType + sess_perc + block_in_sess subject)#				
Hypothesis#	Performance is significantly reduced in the brain-lesioned group.				
Coeff.#	Estimate	SE	t-stat	DF	P-value
Intercept#	0.74626	0.022347	33.394	9498	2.7758e-231
group_amyg#	-0.10516	0.03582	-2.9357	9498	0.0033362
group_VS#	-0.20892	0.039254	-5.3222	9498	1.0484e-07
blockType_Where#	-0.025193	0.011474	-2.1958	9498	0.028134
group_amyg: blockType_Where#	0.034786	0.021032	1.6539	9498	0.098172
group_VS: blockType_Where#	0.1717	0.02165	7.9306	9498	2.4265e-15
Planned contrasts#	Group differences during Where blocks				
group_amyg – group_VS# (amyg – VS)	0.10376	0.042722	2.4288	9498	0.015167
group_amyg + group_amyg: blockType_Where# (amyg - control)	-0.07037	0.029571	-2.3797	9498	0.017346
group_VS + group_VS: blockType_Where# (VS - control)	-0.037222	0.031844	-1.1689	9498	0.24248
(group_amyg + amyg:Where) – (group_VS + group_VS:Where)# (amg - VS)	-0.033148	0.03556	-0.93216	9498	0.35128

To reflect these results, we have now revised the corresponding paragraph in the manuscript as follows (Line #172):

During the What blocks of the What/Where task, however, amygdala-lesioned monkeys performed significantly better than VS-lesioned monkeys (mixed-effects analysis, planned contrast between lesion groups: $\beta_{\text{amyg}} - \beta_{\text{VS}} = 0.104$, $p = .0152$; **Fig. 1f** inset; **Supplementary Table S3.2**), while both lesioned groups were impaired relative to control monkeys (amygdala: $\beta_{\text{amyg}} = -0.105$, $p = .00334$; VS: $\beta_{\text{VS}} = -0.209$, $p = 1.05 \times 10^{-7}$). In the Where blocks that did not require stimulus-based learning, only amygdala-lesioned monkeys showed impairments in performance relative to controls (contrast from controls: $b = -0.070$, $p = .0173$; **Fig. 1g** inset; **Supplementary Table S3.2**). VS-lesioned performance was comparable to that of controls (contrast from controls: $b = -0.037$, $p = .242$) yet not significantly better than that of amygdala-lesioned monkeys (contrast between lesion groups: $b = -0.033$, $p = .351$).

To clarify how the statistics have been run, please provide a summary of the different regression analyses that have been run on the data, specifying for each regression:

1. *What data was included in the regression (which tasks, block types, and lesion groups).*
2. *What the dependent variable was and what each observation corresponded to (e.g. performance in one block).*
3. *What predictors were included and how categorical predictors were coded.*
4. *What the fixed and random effects structure was (provide the formula used to specify it).*
5. *What hypothesis the regression was designed to test (e.g. differences between which groups or block types), and which predictor are relevant for testing these.*

Providing the full results tables for the key regressions along with this information would be really helpful for understanding how the stats reported in the text were run.

[R3.2.3] We thank the reviewer for these helpful suggestions. Below, we provide the full list of summary for other key regression analyses that we have performed and reported in the text. As noted above, we have now revised the regression models to fix the issues related to inconsistent use of the reference category, and improved clarity/accuracy of the results where appropriate. We have also provided the new **Supplementary Table 3**, containing full results of the key analyses.

Regression analysis comparing the performance of controls by block types (Line #115-125)

1. Data included: Control group data across both What-only and What/Where tasks
2. Dependent variable: performance (pBetter) during each block
3. Predictors: block type (*{What-only, What, Where}*, with *What-only* as the reference category) and reward uncertainty (measured as variance of reward schedule; continuous variable). For better interpretability of the regression coefficients, we now mean-center the reward variance within each subject, which was not done previously.
4. Model formula: $pBetter \sim 1 + reward_variance + block_type + (1+reward_variance|subject)$.

Note the random slope for 'block_type' was not included here, since the majority of subjects have exclusively completed either What-only or What/Where task. We had previously considered interaction, but because we don't comment on it and are mostly interested in the main effects, we have removed the interaction term for simplicity (block_type:reward_variance). (Adding the interaction term does not alter the significance of main predictors.)

5. Hypothesis: the given task condition (block type) and reward uncertainty (variance) significantly affect performance

Results:

Comparison of performance in control monkeys across all tasks.

Model#	$Block\text{-}wise\ P(Better) \sim blockType + reward_var + (1 + reward_var + sess_perc + block_in_sess subject)\#$				
Hypothesis#	The given block condition (What-only , What , or Where) and reward uncertainty (variance) significantly affect performance.				
Coeff.#	Estimate	SE	t-stat	DF	P-value
Intercept#	0.78439	0.022694	34.564	7609	2.7187e-243
blockType_what#	-0.07462	0.006738	-11.075	7609	2.7377e-28
blockType_where#	-0.086862	0.006765	-12.84	7609	2.3872e-37
reward_var#	-1.9495	0.16745	-11.642	7609	4.597e-31

Note that the procedure for this model had been described previously, which we further clarify as below (Line #1108):

For comparing block-wise performance within each group, we included random effects of subjects and fixed effects of block types (*What-only*, *What*, *Where*) and reward uncertainty (measured as the variance of the outcome¹³), with the following formula: $P(Better) \sim variance + BlockType + (1 + variance + sess_perc + block_in_sess |subject)$. Specifically, the variance was calculated as $p_B^*(1 - p_B)$, where p_B represents the reward probability of the better option (i.e., $p_B = \{0.8, 0.7, 0.6\}$). Variance was further mean-centered by subject for better interpretability of other coefficients.

Analysis of the mean effective arbitration rates (Line #388-439; **Fig. 3e-g** insets): We have now updated these analyses with the following results below. Previously, for *What/Where* task, we had fitted separate regression models to each block type data and obtained a single intercept term to examine whether the quantity $\Delta\psi$ significantly differs from zero. Noting that this may have been an artificial division of data based on true block types (of which the subjects were unaware), we now compile all data for a given task and instead include block type as a predictor, fitting a single regression model for parsimony. Similar to above, we use planned contrasts to test the relevant hypotheses. Results of this analysis have been added as **Supplementary Table 3.3** (*What-only* task) and **Supplementary Table 3.4** (*What/Where* task), attached below (also see **[R3.2.4]** for more comments on this analysis).

#1. Comparison of effective arbitration rates in controls, *What-only* task

1. Data included: Control group data in *What-only* task
2. Dependent variable: mean paired difference between the two effective arbitration rates ($\Delta\psi = \psi_+ - \psi_-$) in each block
3. Predictors: none (other than fixed intercept, representing the group mean)
4. Model formula: $\Delta\psi \sim 1 + (1 + sess_perc + block_in_sess |subject)$
5. Hypothesis: in intact controls during *What-only* task (where stimulus learning is the “correct” strategy), the effective arbitration rates toward the stimulus-based system (ψ_+) are larger than those toward the action-based system (ψ_-), namely $\Delta\psi > 0$.

Supplementary Table 3.3. Comparison of effective arbitration rates ($\Delta\psi = \psi_+ - \psi_-$) across the three groups during the *What-only* task.

Model#	Block-wise $\Delta\psi \sim 1 + (1 + \text{sess_perc} + \text{block_in_sess} \text{subject}) \#$				
Hypothesis#	Arbitration rates toward stimulus system are significantly larger than those toward action system in controls, and this effect is reduced in amygdala-lesioned monkeys				
Coeff.#	Estimate	SE	t-stat	DF	P-value
Intercept#	0.092809	0.004193	22.134	4099	1.2391e-102
group_amygdala#	-0.078571	0.006064	-12.958	4099	1.1508e-37
group_VS#	-0.070159	0.007225	-9.7103	4099	4.7004e-22
Planned contrasts#	Mean of $\Delta\psi$ in brain lesioned groups				
Intercept + group_amyg#	0.014238 (amyg mean)	0.0043799	3.2507	4099	0.0011606
Intercept + group_VS#	0.022649 (VS mean)	0.005884	3.8493	4099	0.00012028
group_amyg – group_VS#	-0.0084115	0.0073352	-1.1467	4099	0.25156

#2. Comparison of effective arbitration rates in What/Where task:

1. Data included: All three groups data during What/Where task
2. Dependent variable: mean paired difference between the two effective arbitration rates ($\Delta\psi = \psi_+ - \psi_-$) in each block.
3. Predictors: group assignment (control as reference) and block type (“What” or “Where,” with the former as the reference category)
4. Model formula: $\Delta\psi \sim 1 + \text{group} * \text{block_type} + (1 + \text{block_type} + \text{sess_perc} + \text{block_in_sess} | \text{subject})$
5. Hypothesis: in intact controls during What/Where task (where there is uncertainty about the correct model of the environment), the effective arbitration rates toward the correct system (i.e., ψ_+ in What blocks and ψ_- in Where blocks) are larger than those toward the incorrect system (i.e., ψ_- in What blocks and ψ_+ in Where blocks). Namely $\Delta\psi > 0$ in What blocks and $\Delta\psi < 0$ in Where blocks. Amygdala-group is not differentiated in the two rates.

Supplementary Table 3.4. Comparison of effective arbitration rates across the three groups during the What/Where task.

Model#	Block-wise $\Delta\psi \sim \text{group} * \text{blockType} + (1 + \text{blockType} + \text{sess_perc} + \text{block_in_sess} \text{subject}) \#$				
Hypothesis#	Effective arbitration rates toward the correct system (for a given block) are appropriately biased in controls, but amygdala group is not differentiated in the two rates.				
Coeff.#	Estimate	SE	t-stat	DF	P-value
Intercept#	0.044358	0.007343	6.0409	9498	1.5905e-09
group_amyg#	-0.032996	0.011793	-2.798	9498	0.0051522
group_VS#	-0.097925	0.012858	-7.6156	9498	2.8745e-14
blockType_Where#	-0.07444	0.016061	-4.6347	9498	3.6214e-06
group_amyg: blockType_Where#	0.05961	0.026653	2.2365	9498	0.025341
group_VS: blockType_Where#	0.054152	0.0286	1.8934	9498	0.058337
Planned contrasts#	Mean of $\Delta\psi$ during Where blocks in controls, Mean of $\Delta\psi$ during each block type in amygdala group				
Intercept + blockType_Where#	-0.030082 (Control, Where)	0.011626	-2.5874	9498	0.0096838
Intercept + group_amyg#	0.011362 (Amyg, What)	0.0092275	1.2314	9498	0.21822
Intercept + group_amyg + blockType_Where+group_amyg: blockType_Where#	-0.003467 (Amyg, Where)	0.015619	-0.22199	9498	0.82432

Other Post-hoc contrasts#	-0.053566 (VS, What)	0.010555	-5.0748	9498	3.9533e-07
#	-0.073854 (VS, Where)	0.017323	-4.2633	9498	2.034e-05

To reflect these results, we have now revised the relevant parts of the manuscript as follows (Line #393-429):

We found that in control monkeys, the effective arbitration rates toward the stimulus-based (ψ_+) or action-based (ψ_-) system diverged toward the end of a block, reflecting the adoption of the correct model of the environment. That is, when the stimulus-based system was more reliable, the effective arbitration rates toward the stimulus-based system were larger than those toward the action-based system (mixed-effects model with a single fixed intercept, representing the mean $\Delta\psi$; What-only task: $\beta_0 = 0.0930$, $p = 1.24 \times 10^{-102}$; What block of What/Where task: $\beta_0 = 0.0444$, $p = 1.59 \times 10^{-9}$; **Fig. 3e, f; Supplementary Table S3.3, S3.4**). Similarly, in the Where blocks, where the action-based system was more reliable, the effective arbitration rate toward the action-based system was significantly larger than that toward the stimulus-based system ($\beta_0 = -0.0301$, $p = .00968$; **Fig. 3g**).

In contrast, amygdala-lesioned monkeys showed the minimum differentiation between adjustments toward the more and less reliable (correct and incorrect) systems. Notably, during the What/Where task, amygdala-lesioned monkeys exhibited no significant difference between two arbitration rates during either block type (mixed-effects analysis on $\Delta\psi$ with a single intercept; What: $\beta_0 = 0.0114$, $p = .218$; Where: $\beta_0 = -0.00347$, $p = .824$; **Fig. 3f, g; Supplementary Table S3.4**). In contrast, VS-lesioned monkeys exhibited an overall large bias in arbitration rates toward the action-based system (i.e., higher ψ_-) during both block types (mixed-effects analysis on $\Delta\psi$ with a single intercept; What: $\beta_0 = -0.0536$, $p = 3.95 \times 10^{-7}$; Where: $\beta_0 = -0.0739$, $p = 2.03 \times 10^{-5}$; **Fig. 3f, g; Supplementary Table S3.4**).

Analysis of the baseline inverse temperature:

Lastly, we had also examined whether the estimated model parameters ($\Delta\beta = \beta_{\text{stim}} - \beta_{\text{action}}$) significantly differ *between* the groups (Line #456-463; **Fig. 4b-c** insets). One main difference for the analysis of model parameters is that, since the parameters are estimated at the level of session, the unit of analysis is each session (as opposed to blocks) and thus do not differentiate between the two block types of What/Where task (Note that two block types were randomly interleaved within each session, and the dynamic arbitration model estimates a single set of parameters across the given set of blocks).

1. Data included: all three groups data during each task (either What-only or What/Where)
2. Dependent variable: relative sensitivity to stimulus- and action-value signals ($\Delta\beta = \beta_{\text{stim}} - \beta_{\text{action}}$), estimated for each session by the winning RL model

3. Predictors: group assignment, with controls as a reference category
4. Model formula: $\Delta\beta \sim 1 + \text{group} + (1|\text{subject})$
5. Hypothesis: the difference in effective arbitration rates ($\Delta\psi$) is significantly different from zero (i.e., either $\psi_+ > \psi_-$ or $\psi_+ < \psi_-$). More specifically, for intact controls we hypothesized $\psi_+ > \psi_-$ during What-only and What blocks of What/Where task, and $\psi_+ < \psi_-$ during Where blocks of What/Where task, considering the normative advantage for performing each block type. In contrast, we were agnostic about the presence of such effects in lesioned groups).

Results: Updated as Supplementary Table 3.6-7

Supplementary Table 3.6. Comparison of relative sensitivity to stimulus- and action-value signals ($\Delta\beta = \beta_{\text{stim}} - \beta_{\text{action}}$) across the three groups during the What-only task.

Model#	Session-wise $\Delta\beta \sim \text{group} + (1 + \text{sess_perc} \text{subject}) \#$				
Hypothesis#	Relative sensitivity to stimulus is not different between control and amygdala group, but it is reduced in VS group.				
Coeff.#	Estimate	SE	t-stat	DF	P-value
Intercept#	4.9348	1.7631	2.799	240	0.0055428
group_amygdala#	-2.6925	2.4859	-1.0831	240	0.27984
group_VS#	-7.4768	2.7272	-2.7415	240	0.0065759
Planned contrasts#	Difference between amygdala and VS groups				
group_amyg - group_VS#	4.7843	2.7204	1.7587	240	0.079906

Supplementary Table 3.7 Comparison of relative sensitivity to stimulus and action value signals ($\Delta\beta = \beta_{\text{stim}} - \beta_{\text{action}}$) across the three groups during the What/Where task.

Model#	Session-wise $\Delta\beta \sim \text{group} + (1 + \text{sess_perc} \text{subject}) \#$				
Hypothesis#	Relative sensitivity is not different between control and amygdala group, but it is reduced in VS group.				
Coeff.#	Estimate	SE	t-stat	DF	P-value
Intercept#	2.3268	0.95107	2.4465	529	0.014749
group_amygdala#	0.54218	1.5067	0.35984	529	0.71911
group_VS#	-3.53	1.6544	-2.1337	529	0.033323
Planned contrasts#	Difference between amygdala and VS groups				
group_amyg - group_VS#	4.0722	1.7883	2.2771	529	0.02318

The corresponding results have been updated in the text as follows (Line #456):

As a result, VS-lesioned monkeys exhibited a smaller difference in choice sensitivity to stimulus- and action-value signals ($\Delta\beta = \beta_{\text{stim}} - \beta_{\text{action}}$), with a bias toward the action-value signals, compared to controls in both the What-only task (mixed-effects analysis on $\Delta\beta$, main effect of group; $\beta_{\text{VS}} = -7.48$, $p = .00658$; **Fig. 4b** inset; **Supplementary Table S3.6**) and the What/Where tasks ($\beta_{\text{VS}} = -3.53$, $p = .0333$; **Fig. 4c** inset). This was not the case for amygdala-lesioned monkeys, which exhibited no significant difference compared to controls in either task (What-only: $\beta_{\text{amyg}} = -2.69$, $p = .280$; **Fig. 4b** inset; $\beta_{\text{amyg}} = 0.542$, $p = .719$; **Fig. 4c** inset; **Supplementary Table S3.6**).

Analysis on the long-term adjustment: for this set of analyses presented in the final section of the Results, we refer the reviewer to our response in **[R3.2.5]** below.

***Supplementary section**

Below are analyses or statistics reported in the Supplementary information. To maintain brevity and avoid clutter, we chose not to include the full results table for these findings, as they are less central to our main conclusions. Nevertheless, we have now reviewed and revised all mixed-effects models to be consistent with the above approach, fully including random slopes for all predictors where appropriate. We also explicitly specify the model formulas used for each analysis.

Distributions of reaction time for stimulus-dominant and action-dominant trials

(Supplementary Note 1): In this analysis, we had previously considered simple mixed-effects models with the random intercepts for subjects only. For improved accuracy, we have now included random slopes for all the fixed effects. The revised results are as follows. Note that there are only minor changes in the reported statistics, and the conclusion (that action-dominant trials are faster in RT) remain the same.

Line #1353 (in the text for Supplementary Note 1):

Specifically, the mixed-effects model included **subject-level random intercepts and random slopes for the following fixed effects predictors:** dominant strategy, coded as stim-dominant (0) or action-dominant (1), whether the monkey has chosen the better option (1) or not (0), reward schedule or uncertainty, trial number within a block, block number within a session, and session number within the subject.

Figure SN1. Distributions of reaction time (RT) for stimulus-dominant and action-dominant trials, separately for different tasks and block types. Each trial was categorized as either stimulus- or action-dominant by comparing $ERDS_{Stim}$ and $ERDS_{Action}$. Percentages indicate proportions of trials under each category, reported as mean \pm SEM across subjects (remaining percentages correspond to trials where both strategies were equally dominant). (a–c) RT data from control monkeys during the What-only (a) and the What/Where tasks (b, c). (d–f) RT data from amygdala-lesioned monkeys during the What-only (d) and the What/Where tasks (e, f). (g–i) RT data from VS-lesioned monkeys during the What-only (g) and the What/Where tasks (h, i). Circles in the violin plots represent means and black horizontal lines represent medians of the distributions. Reported are standardized regression coefficient and corresponding p-values for the main effect of action-dominant strategy, with negative (positive) values indicating that action-dominant trials predict shorter (longer) RT. In the What/Where task (b, c, e, f, h, i), RT was consistently shorter for action-dominant trials across all groups.

Regression of reaction time (RT) on arbitration weight and other task-related variables (Supplementary Note 2): Similar to above, we have now revised the previous random intercept-only to include random slopes for all of the main predictors, retaining the most conservative setting to guard against the Type I error.

Line #1438 and onward:

To this end, we used a mixed-effects analysis with **subject-level random intercepts and random slopes for the following fixed effects predictors**: arbitration weight (Ω), absolute overall value difference ($|\Delta OV| = |OV_{\text{Left}} - OV_{\text{Right}}|$), whether the animal's choice was the "correct" option (in terms of higher reward probability), trial number within a block, block number within a session, and session number within the subject. Note that in this analysis, the block type (for the What/Where task) and reward uncertainty were not included as predictors, because these were objective variables unknown to the monkeys and were instead captured by arbitration weight and the value difference (reflecting monkey's subjective estimates).

Our analysis based on model-derived estimates (**Supplementary Fig. SN2**) showed **overall positive** effects of arbitration weight on RT across all tasks and groups, supporting the view that action-based decision is faster than stimulus-based decision. **During the What-only task, the regression weights of arbitration weight for control and amygdala-lesioned monkeys were positive although not significant** (controls: $\beta_{\Omega} = 0.0113$, $p = .632$; **Supplementary Fig. SN2a**; amygdala: $\beta_{\Omega} = 0.0536$, $p = .149$; **Supplementary Fig. SN2c**), while VS-lesioned monkeys showed significant effect ($\beta_{\Omega} = 0.162$, $p = 5.27 \times 10^{-3}$; **Supplementary Fig. SN2e**). This could be related to the fact that arbitration weights in control monkeys were saturated toward the stimulus-based system in this task (see **Fig. 2d**), in contrast to the lesioned groups that exhibited more mixture of two strategies (e.g., **Fig. 3b, c**). In particular, the largest proportion of variance in VS group's RT was accounted by the arbitration weight ($R^2_{\Omega} = 3.97 \times 10^{-5}$, followed by $R^2_{\text{Block\#}} = 4.16 \times 10^{-6}$ and $R^2_{|\Delta OV|} = 7.27 \times 10^{-7}$; partial R^2 values), suggesting its significant role in reducing RT of VS-lesioned monkeys. Furthermore, across all groups, we found significant effects of $|\Delta OV|$, such that the larger distinction between two choice options led to significantly shorter RT (controls: $\beta_{|\Delta OV|} = -0.0824$, $p = .00909$; amygdala: $\beta_{|\Delta OV|} = -0.064$, $p = 1.45 \times 10^{-4}$; VS: $\beta_{|\Delta OV|} = -0.118$, $p = 3.88 \times 10^{-11}$). These results illustrate the RL model's ability to capture the animals' subjective value estimates and demonstrate that the observed effects of the arbitration weight on RT are unique from other included effects.

During the What/Where task, which required more explicit arbitration between two strategies, all groups showed significant effects of Ω on RT, with higher Ω predicting slower RT (controls: $\beta_{\Omega} = 0.117$, $p = 4.63 \times 10^{-5}$; amygdala: $\beta_{\Omega} = 0.161$, $p = 4.30 \times 10^{-12}$; VS: $\beta_{\Omega} = 0.0475$, $p = .0273$). These results further support the claim that the relative dominance of the strategy, even when measured in a continuous scale, can significantly predict the animals' RT.

Supplementary Figure SN2. Regression of reaction time (RT) on arbitration weight and other task-related variables. Plotted are standardized regression coefficients from a mixed-effects model with subjects as a random effect, for given lesioned groups and tasks. Asterisk indicates significant effect of a given predictor (*: $p < .05$, **: $p < .01$, ***: $p < .001$). *corr. vs. incorr.*: the main effect of choosing the option with higher reward probability (correct option). (a, c, e) Regression coefficients for predicting RT data during What-only task, in controls (a), amygdala- (c), and VS-lesioned (e) monkeys. (b, d, f) Regression coefficients for predicting RT data during What/Where task, in controls (b), amygdala- (d), and VS-lesioned (f) monkeys. Overall, task-related variables have similar impacts on RT in all groups.

Conditional entropy of reward-dependent strategy (ERDS) for stimulus and action, by block type and reward schedule (Fig. S2 & Fig. S3): We have also revised the mixed-effects model used to determine the significance of reward uncertainty (variance), by including random slopes for this variable as well as two additional slopes controlling for the effect of long-term adjustment. In these plots, since we did not make any between-group comparisons, we simply fitted individual regression models for each group for each task. We now explicitly specify the model formula used as follows:

Figure S2. Distributions of conditional entropy of reward-dependent strategy (ERDS) by block type and reward schedule. Distributions of the ERDS based on stimulus identity ($ERDS_{Stim}$, in solid lines) or action ($ERDS_{Action}$, in solid lines) computed from each of 80-trial blocks during What-only (first column) and What/Where task (second and third columns). Insets represent mean values (Error bars = SEM across subjects). Asterisks indicate significant effect of reward uncertainty on ERDS, using mixed-effects analysis with subjects as a random effect (*: $p < .05$, **: $p < .01$, ***: $p < .001$), with the following model specifications: **What-only:** $ERDS \sim reward_var + (1+reward_var+sess_perc+block_in_sess |subject)$; **What/Where:** $ERDS \sim reward_var*blockType + (1+reward_var*blockType +sess_perc+block_in_sess |subject)$. Reward uncertainty of each block was measured as the variance of outcome¹³, i.e., $p_{Better}*(1-p_{Better})$. (a–c) $ERDS_{Stim}$ (in solid lines) and $ERDS_{Action}$ (in dotted lines) in control monkeys during What-only (a) and What/Where tasks (b, c). Colors indicate reward schedules (black: 80/20; dark gray: 70/30, light gray: 60/40). (d–f) ERDS in amygdala-lesioned during What-only (d) and What/Where tasks (e, f). Colors indicate reward schedules (brown: 80/20; red: 70/30, orange: 60/40). (g–i) ERDS in VS-lesioned during What-only (g) and What/Where tasks (h, i). Colors indicate reward schedules (navy: 80/20; blue: 70/30, cyan: 60/40).

Figure S3. Interactions between stimulus-based and action-based learning depend on the uncertainty of the reward environment. (d–f) Summary results for the panels in a–c. Plotted are mean values of $\Delta ERDS = ERDS_{Stim} - ERDS_{Action}$, by each group and reward schedule during What-only (d) and What/Where (e,f) tasks. With larger reward uncertainty, animals' strategies became relatively more biased toward the incorrect strategy (increasing $\Delta ERDS$ in the What blocks and decreasing $\Delta ERDS$ in the Where block). Asterisks next to the plots indicate significant effect of reward uncertainty (variance) for the respective group indicated by colors, as determined by mixed-effects analysis (same models as in Supplementary Fig. S2; *, $p < .05$, **, $p < .01$, ***, $p < .001$). It is worth noting that, while $ERDS_{Action}$ was not explicitly modulated by reward uncertainty during the What-only task for any of the groups (Supplementary Fig. S2a, d, g), competitive interactions between $ERDS_{Stim}$ and $ERDS_{Action}$ still exist for the task. This suggests that $ERDS_{Stim}$ cannot be solely responsible for the observed effect in $\Delta ERDS$.

Measuring the interaction between stimulus- and action-based strategy in controls during What-only task (Fig. S6):

In this section of the text (#314-331), we had tested how faithfully various RL models can reproduce the negative interaction between the two learning strategies (as reflected by ERDS). Note that, since our intention here was to measure the association between ERDS while accounting for the repeated measurements per subject (rather than the predictive power of one variable against the other), we have simply fitted the model as $ERDS_{Action} \sim ERDS_{Stim} + (1|subject)$ previously. However, to make the model more appropriate and accurate, we have now mean-centered the predictor ($ERDS_{Stim}$) before fitting the same model. This approach ensures that the fixed effects reflect how deviations from an individual's mean $ERDS_{Stim}$ predict changes in the $ERDS_{Action}$, rather than being confounded by overall differences between individuals (e.g., similar to correlation, but preserving the scale). Note that random slopes are not appropriate here because our goal is to estimate a single, population-level within-subject effect of the predictor.

We now briefly include this rationale in the Methods, and have revised Supplementary Fig.S6 below. Overall, the values have only changed slightly and our conclusion has remained the same (i.e., that dynamic arbitration model best captures the competitive interaction between two strategies). To more directly illustrate this point, we have further added Kolmogorov-Smirnov test comparing the distributions of $\Delta ERDS (= ERDS_{Stim} - ERDS_{Action})$ between the data and the simulated model in panels b–d.

Line #1036 in Methods:

To test whether the observed relationship between stimulus- and action-based learning indeed required competition between the two learning systems and was not due to task structure (**Supplementary Fig. S6**), we simulated choice behavior using single-system or two-system models and computed model-simulated $ERDS_{Stim}$ and $ERDS_{Action}$ (Eq. 1). To measure the association between the two measures and isolate the within-subject effect, we mean-centered the predictor ($ERDS_{Stim}$) and fitted the mixed-effects model as $ERDS_{Action} \sim ERDS_{Stim} + (1|subject)$ to account for subject variability.

Figure S6. Comparison of empirical and simulated ERDS in control monkeys during the What-only task provides evidence for presence of multiple learning systems. (a) Scatter plot of entropy of reward-dependent strategy based on stimulus identity ($ERDS_{Stim}$, X-axis) vs. action ($ERDS_{Action}$, Y-axis) in control monkeys during the What-only task. Reported are regression coefficient (β) and its p-value for the regression of $ERDS_{Action}$ on $ERDS_{Stim}$, using mixed-effects analysis with subjects as a random effect ($ERDS_{Action} \sim ERDS_{Stim} + (1|subject)$, mean-centered predictors by subject). Histograms in the X- and Y-axis show distributions of $ERDS_{Stim}$ and $ERDS_{Action}$, respectively. (b) Simulated metrics using estimated parameters of the model with stimulus-learning system only (RL_{Stim-only}). Reported in bottom right is the test statistics (D-values) and its p-value from the two-sample Kolmogorov-Smirnov test, comparing the distance between the distributions of $\Delta ERDS = ERDS_{Stim} - ERDS_{Action}$ from the data (a)

and the simulated model. (c) Simulated metrics using estimated parameters of the static two-system model (RL_{Stim+Action}+Static ω). (d) Simulated metrics using estimated parameters of the two-system model with dynamic adjustment using V_{chosen} as the reliability measure (Dynamic ω : V_{cho}). The Dynamic ω model better captures the variability in ERDS_{Action} compared to the Static ω model shown in panel (c), as reflected in the more similar regression coefficient (β) to the data and the smaller distance to the empirical distribution of ΔERDS (D-values).

Group difference in fitted parameters (Fig.S9-11): Finally, using the same consistent scheme for mixed-effects (controls as reference category and using contrasts analysis), we have updated the figures in **Supplementary Fig. S9, Fig.S10, and Fig.S11** as follows. Please note that we now include subject-level random slopes for the percentage of sessions completed (*sess_perc* variable) as done in other analyses.

Figure S9. Distributions of estimated model parameters in the What/Where task across the three groups. Plotted are the distributions of estimated parameters of the best model (Dynamic ω - ρ with V_{cho}) fitted to the choice behaviors of controls (black), amygdala- (red), and VS-lesioned (blue) monkeys during the What/Where task. α_+ : learning rate on rewarded trials (a). β_1 : common inverse temperature for stimulus- and action-based systems (b). β_0 : side bias term, positive if preferring right option (c). α_- : learning rate on unrewarded trials (d). ζ : decay or forgetting rate for the unchosen option (e). α_ω : arbitration transition rate (f). ω_0 : initial arbitration weight on the first trial of each block (g). ζ_ω : decay rate for arbitration weight ω toward initial value (h). Asterisks indicate significant effects of group difference (mixed-effects analysis with contrasts, *parameter ~ group + (1+sess_perc|subject)*); *: $p < .05$, **: $p < .01$, ***: $p < .001$). Circles in the violin plots indicate medians, and the numbers on

the X-axis indicate mean \pm SEM. across subjects. ρ parameter is assumed to be fixed for each subject and is shown in Fig. 4a.

Figure S10. Distributions of estimated model parameters in the What-only task across the three animal groups. Plotted are the distributions of estimated parameters of the best model (Dynamic ω - ρ with V_{cho}) fitted to the choice behaviors of controls (black), amygdala- (red), and VS-lesioned (blue) monkeys during the What-only task. α_{+} : learning rate on rewarded trials (a). β_1 : common inverse temperature for stimulus- and action-based systems (b). β_0 : side bias term, positive if preferring right option (c). α_{-} : learning rate on unrewarded trials (d). ζ : decay or forgetting rate for the unchosen option (e). α_{ω} : arbitration transition rate (f). ω_0 : initial arbitration weight on the first trial of each block (g). ζ_{ω} : decay rate for arbitration weight ω toward initial value (h). Asterisks indicate significant effects of group difference (mixed-effects analysis with contrasts, $parameter \sim group + (1+sess_perc|subject)$); *: $p < .05$, **: $p < .01$, ***: $p < .001$). Circles in the violin plots indicate medians, and the numbers on the X-axis indicate mean \pm SEM across subjects. ρ parameter is assumed to be fixed for each subject and is shown in Fig. 4a.

Figure S11. Comparison between dynamic models with or without passive decay in arbitration weight, and estimated parameters in the better-fitting model. (a) Goodness of fit using five-fold cross-validation of dynamic models with or without passive decay for ω . Numbers in parenthesis indicate McFadden R^2 (Eq. 20). Model with the additional passive decay mechanism better accounts for the choice behavior in all groups, especially during the What/Where task. **(b)** Distribution of estimated parameters for sessions in controls (black), amygdala (red), and VS (blue) groups during the What-only task, showing parameters for α_ω in the above model without passive decay in ω . Asterisks indicate significant effects of group difference (mixed-effects analysis with random effects of subjects, $parameter \sim group + (1+sess_perc|subject)$); *: $p < .05$, **: $p < .01$, ***: $p < .001$). Circles in the violin plots indicate medians, and the numbers on the X-axis indicate mean \pm SEM across subjects. **(c)** Same plots as in (b) but for the above model with passive decay in ω , showing parameters for α_ω (model arbitration/update rate, left) and ζ_ω (decay rate for ω , right). Arbitration rates α_ω tended to be larger than the passive decay rate ζ_ω in control and amygdala groups (mixed-effects analysis on paired difference, $\alpha_\omega - \zeta_\omega \sim 1 + (1+sess_perc|subject)$); controls: $\beta_0 = 0.0497$, $p = .515$; amygdala: $\beta_0 = 0.067$, $p = .459$; VS: $\beta_0 = -0.0929$, $p = .112$). **(d)** Same plot as in (b) but for the What/Where task, in the model without passive decay in ω . **(e)** Same plots as in (c) but for What/Where task, in the model with passive decay in ω . Arbitration rates α_ω were significantly larger than the passive decay rate ζ_ω across all groups, suggesting that α_ω is the primary source of the transitions in ω (mixed-effects analysis on $\alpha_\omega - \zeta_\omega$; controls: $\beta_0 = 0.269$, $p = 2.18 \times 10^{-25}$; amygdala: $\beta_0 = 0.215$, $p = 3.02 \times 10^{-24}$; VS: $\beta_0 = 0.163$, $p = 1.93 \times 10^{-16}$).

A related statistical issue is that in several places the text imply differences between block types or groups, but the stats reported do not appear to correspond to direct comparisons between the groups or conditions that would be needed to demonstrate a significant difference. E.g.:

Lines 411:426: “When the stimulus-based system was more reliable, the effective arbitration rates toward the stimulus-based system were larger in control monkeys” and “In contrast, amygdala-lesioned monkeys showed the minimum differentiation between adjustments toward the more and less reliable (correct and incorrect) systems”. These statements seem to imply differences between groups and/or conditions, but the only P values reported are for intercept predictors β_0 which do not appear suitable for making statements about such differences.

[R3.2.4] We thank the reviewer for pointing out this concern, as our previous phrasing seems to have caused confusion. Our intention was not to make a group comparison on the effective arbitration rates between control and amygdala group; rather, we simply tested whether the quantity of interest, $\Delta\psi = \psi_+ - \psi_-$, is significantly different from zero within each group. We have now clarified the quoted portion as follows (Line #393):

We found that in control monkeys, the effective arbitration rates toward the stimulus-based (ψ_+) or action-based (ψ_-) system diverged toward the end of a block, reflecting the adoption of the correct model of the environment. That is, when the stimulus-based system was more reliable, the effective arbitration rates toward the stimulus-based system were larger than those toward the action-based system.

Nonetheless, we acknowledge the reviewer’s point that we had not made a direct comparison between control and amygdala. To address this, we have now added the following analysis that directly compares the absolute value of $\Delta\psi$ between control and amygdala groups, to explicitly test whether amygdala-lesioned group exhibits minimum differentiation between two arbitration rates under uncertainty about correct model of the environment (i.e., What/Where task):

1. Data included: All three groups data (control, amygdala, VS) in What/Where task
2. Dependent variable: mean *absolute* paired difference between the two effective arbitration rates ($|\Delta\psi| = |\psi_+ - \psi_-|$) in each block.
3. Predictors: group assignment (control as reference category)
4. Model formula: $\Delta\psi \sim 1 + \text{group} + (1|\text{subject})$
5. Hypothesis: Amygdala-lesioned group exhibits less distinction between two effective arbitration weights compared to intact controls

Results: Updated in the text as **Supplementary Table 3.5** (attached below).

Supplementary Table 3.5. Comparison of differentiation in two arbitration rates between groups during the What/Where task.

Model#	Block-wise $ \Delta\psi \sim group + (1 + sess_perc + block_in_sess subject) \#$				
Hypothesis#	Amygdala group shows minimal differentiation between the two rates.				
Coeff.#	Estimate	SE	t-stat	DF	P-value
Intercept#	0.055852	0.006268	8.9114	9501	5.9665e-19
group_amygdala#	-0.025358	0.010026	-2.5293	9501	0.011445
group_VS#	0.0028819	0.010941	0.26341	9501	0.79224
Planned contrasts#	Difference between amygdala and VS groups				
group_amyg – group_VS#	-0.02824	0.011902	-2.3727	9501	0.017676

We have added these results in Line #429:

Overall, amygdala-lesioned monkeys were characterized by the least amount of differentiation between the two arbitration rates (mixed-effects analysis on $|\Delta\psi|$; contrast from controls: $b = -0.0254$, $p = .0114$; contrast from VS: $b = -0.0282$, $p = .0177$; **Supplementary Table S3.5**).

Additionally, we had previously described the rationale for this analysis in **Methods**, which we have now further clarified as follows (Line #1096):

Because the data were collected across different animals, we primarily utilized linear mixed-effects regression analyses (using MATLAB *fitlme* function) for between-group comparisons, with the group assignment as a fixed effect and subjects as a random effect, to appropriately account for between-subject variance. To maximize the accuracy of models and reduce Type-I error, we considered subject-level intercepts and additional random slopes for long-term adjustment effects, both across the experiment (*proportion of session completed*) and within each session day (*block within a session*). When performing a within-group significance test to compare a paired set of samples (e.g., $\Delta\beta = \beta_{stim} - \beta_{action}$, $\Delta\psi = \psi_+ - \psi_-$), we fitted mixed-effects models with a fixed intercept and subject-level random effects ($data \sim 1 + (1+sess_perc+block_in_sess|subject)$), where the main intercept represents the mean value of the paired difference. We then tested whether this intercept significantly differed from zero, as indicated by the coefficient β_0 .

Line 573: “we observed a significant decrease in ERDS_{stim} in VS-lesioned monkeys (VS: $\beta_{session}(\%) = -0.207$, $p = 1.03 \times 10^{-5}$), whereas there was no evidence for changes in control or amygdala-lesioned monkeys across time (Control: $\beta_{session}(\%) = -0.0592$, $p = .285$; amygdala: $\beta_{session}(\%) = -0.0108$, $p = .802$.” Again, a difference between groups is being suggested in the absence of any statistics associated with a direct comparison between the groups. A difference in significant is not itself evidence of a significant difference (see <https://doi.org/10.1198/000313006X152649>). Please ensure that all statements about differences between groups and/or conditions are supported by appropriate statistical tests.

[R3.2.5] We thank the reviewer for this insightful comment and providing us with a helpful reference. The reviewer's main concern is that we have primarily used statements about significance/non-significance within individual groups to imply significant group differences. Although we think that the differing presence/absence of significance across groups are interesting phenomena worth reporting on their own (and therefore have kept previously reported statistics), we agree with the general idea that such comparison is not direct evidence for significant difference between groups. Therefore, we have included additional analyses that more directly compare whether the effect is significantly different between groups.

For the quoted portion of the text, we have now updated the description of our procedure as follows (Line #1127 in **Methods**):

To study the long-term adjustment in behavior across the time course of the experiment (**Supplementary Fig. S8**), we calculated $ERDS_{Stim}$, $ERDS_{Action}$ (Eq. 1), and median RT for each block and regressed them on the proportion of the sessions (total block number) completed as the predictor variable. This regressor was further mean-centered by subject for better interpretability of other coefficients. Initial effective arbitration weight (Ω_0), which was estimated for each session, was analyzed at the session level. To further account for the variability in adjustment at the subject level, we considered random slopes and intercepts for the effect of sessions within each subject using mixed-effects models. We included all group data from the What-only task and used planned contrasts to infer slopes for each group and the group differences in the slopes. Full results are reported in **Supplementary Table S3.11-14**. For visualization purposes only, we plot the simple least-squares lines and the estimated slopes for each group (β_{sess}) in **Supplementary Fig. S8**.

As noted, we now use planned contrasts to compare the effect between groups. Full results are attached below (**Supplementary Table 3.11** through **3.14**).

Supplementary Table 3.11. Long-term adjustment in stimulus-based strategy ($ERDS_{Stim}$) during the What-only task.

Model#	Block-wise $ERDS_{Stim} \sim group * session_perc + (1 + session_perc + block_in_sess subject) \#$				
Hypothesis#	There is significant effect of long-term adjustment in stimulus-based strategy				
Coeff.#	Estimate	SE	t-stat	DF	P-value
Intercept#	0.46524	0.03119	14.916	4096	4.9007e-49
group_amyg#	0.38501	0.044163	8.718	4096	4.0496e-18
group_VS#	0.21605	0.048054	4.496	4096	7.1158e-06
session_perc#	-0.076933	0.045091	-1.7062	4096	0.088053
group_amyg: session_perc#	0.043531	0.065166	0.668	4096	0.50417
group_VS: session_perc#	-0.54922	0.080032	-6.8625	4096	7.7832e-12
Planned contrasts#	Slope for each brain-lesioned group Difference in slope between amygdala and VS group				
session_perc + # group_amyg: session_perc#	-0.033402 (amyg slope)	0.047047	-0.70997	4096	0.47776
session_perc + # group_VS: session_perc#	-0.62615 (VS slope)	0.06612	-9.4699	4096	0

group_amyg: session_perc – group_VS: session_perc#	0.59275 (amyg vs. VS)	0.081149	7.3044	4096	3.3307e-13
---	--------------------------	----------	--------	------	------------

Supplementary Table 3.12. Long-term adjustment in action-based strategy (ERDS_{Action}) during What-only task.

Model#	Block-wise ERDS_{Act} ~ group* session_perc + (1+ session_perc + block_in_sess subject)#				
Hypothesis#	There is significant effect of long-term adjustment in action-based strategy				
Coeff.#	Estimate	SE	t-stat	DF	P-value
Intercept#	0.98149	0.005686	172.62	4096	0
group_amyg#	-0.058954	0.009191	-6.4141	4096	1.5771e-10
group_VS#	-0.016861	0.013944	-1.2092	4096	0.22666
session_perc#	-0.019254	0.042687	-0.45106	4096	0.65197
group_amyg: session_perc#	0.033012	0.06049	0.54574	4096	0.58527
group_VS: session_perc#	0.21104	0.068511	3.0803	4096	0.002081
Planned contrasts#	Slope for each brain-lesioned group, Difference in slope between amygdala and VS group				
session_perc + #	0.013757 (amyg slope)	0.042858	0.321	4096	0.748227
session_perc + #	0.191783 (VS slope)	0.053587	3.578898	4096	0.000349
group_amyg: session_perc – group_VS: session_perc#	-0.178026 (amyg vs. VS)	0.068618	-2.594448	4096	0.009508

Supplementary Table 3.13. Long-term adjustment in reaction time (RT) during What-only task.

Model#	Block-wise median RT ~ group*session_perc + (1+ session_perc + block_in_sess subject)#				
Hypothesis#	There is significant effect of long-term adjustment in action-based strategy				
Coeff.#	Estimate	SE	t-stat	DF	P-value
Intercept#	223.6	15.517	14.41	4096	5.8459e-46
group_amyg#	28.694	22.016	1.3033	4096	0.19254
group_VS#	-33.208	24.017	-1.3827	4096	0.16684
session_perc#	6.4037	4.7777	1.3403	4096	0.18022
group_amyg: session_perc#	-5.3131	6.7961	-0.78179	4096	0.43439
group_VS: session_perc#	24.258	7.9722	3.0429	4096	0.0023581
Planned contrasts#	Slope for each brain-lesioned group, Difference in slope between amygdala and VS group				
session_perc + #	1.0906 (amyg slope)	4.833257	0.225644	4096	0.82149
session_perc + #	30.662 (VS slope)	6.381877	4.804542	4096	1.6066e-06
group_amyg: session_perc – group_VS: session_perc#	-29.5714 (amyg vs. VS)	8.005544	-3.693865	4096	2.2375e-04

Supplementary Table 3.14. Long-term adjustment in the initial effective arbitration weight (Ω_0) during What-only task.

Model#	Session-wise Ω_0 ~ group*session_perc + (1+ session_perc subject)#				
Hypothesis#	There is significant effect of long-term adjustment in action-based strategy				
Coeff.#	Estimate	SE	t-stat	DF	P-value
Intercept#	0.89228	0.069903	12.764	237	9.5239e-29
group_amyg#	-0.263	0.09874	-2.6635	237	0.008263
group_VS#	-0.12909	0.10846	-1.1902	237	0.23517
session_perc#	-0.11273	0.061507	-1.8328	237	0.068085
group_amyg: session_perc#	0.13703	0.084658	1.6186	237	0.10686
group_VS: session_perc#	0.56793	0.13812	4.1117	237	5.4173e-05
Planned contrasts#	Slope for each brain-lesioned group, Difference in slope between amygdala and VS group				

session_perc + # group_amyg: session_perc#	0.024298 (amyg slope)	0.058171	0.41771	4096	0.67654
session_perc + # group_VS: session_perc#	0.4552 (VS slope)	0.12367	3.6806	4096	2.8809e-04
group_amyg: session_perc - group_VS: session_perc#	-0.4309 (amyg vs. VS)	0.13667	-3.1528	4096	0.0018258

Reflecting these results, we have now updated the text as follows (Line #576-612):

For consistency in stimulus-based strategy, we observed a long-term decrease in $ERDS_{Stim}$ in VS-lesioned monkeys (planned contrast for the slope of VS group: $b = -0.626$, $p = 4.94 \times 10^{-324}$; **Supplementary Fig. S8a**), to a significantly greater extent than control monkeys ($\beta_{VS:sess\%} = -0.549$, $p = 7.78 \times 10^{-12}$; **Supplementary Table S3.11**). There was no evidence for such adjustment in control ($\beta_{sess\%} = -0.077$, $p = .0881$) or amygdala-lesioned monkeys across time (planned contrast for the slope amygdala group: $b = -0.033$, $p = .478$). Specifically, despite their impaired stimulus-based learning, monkeys with VS lesions were able to increase their adoption of stimulus-based strategy over time. Consistently, these monkeys also decreased their adoption of action-based strategy as reflected in the positive slope of $ERDS_{Action}$ over time (planned contrast for the slope of VS group: $b = 0.192$, $p = 3.49 \times 10^{-4}$; **Supplementary Fig. S8b**), which was significantly greater compared to controls ($\beta_{VS:sess\%} = 0.211$, $p = .00208$; **Supplementary Table S3.12**). There was no evidence of such an effect in control monkeys ($\beta_{sess\%} = -0.019$, $p = .652$) or in monkeys with amygdala lesions (planned contrast for the slope amygdala group: $b = 0.014$, $p = .748$). Interestingly, consistent with previous results, the complementary changes in model adoption in VS-lesioned monkeys were also reflected in increased median RT over time in these monkeys (planned contrast for the slope of VS group: $b = 30.7$, $p = 1.61 \times 10^{-6}$; **Supplementary Fig. S8c**; **Supplementary Table S3.13**) to greater extent than controls ($\beta_{VS:sess\%} = 24.3$, $p = .00236$) or amygdala-lesioned monkeys (planned contrast for group difference in slopes: $b = -29.6$, $p = .00224$). This was accompanied by a long-term increase in the initial effective arbitration weights Ω_0 (toward stimulus-based system) **only in the VS-lesioned monkeys** (planned contrast for the slope of VS group: $b = 0.455$, $p = 2.88 \times 10^{-4}$; **Supplementary Fig. S8d**; **Supplementary Table S3.14**), which was significantly greater compared to both controls ($\beta_{VS:sess\%} = 0.568$, $p = 5.42 \times 10^{-5}$) and amygdala group (planned contrast for group difference in slopes: $b = -0.431$, $p = .00183$).

Lastly, the corresponding figure (**Supplementary Fig. S8** insets) have been updated as follows:

Figure S8. Contribution of the amygdala to behavioral adjustments over long timescales. (a, b) Time course of entropy of reward-dependent strategy on stimulus identity ($ERDS_{Stim}$, a) and performed action ($ERDS_{Action}$, b) for controls (black), amygdala- (red), and VS-lesioned (blue) monkeys during the What-only task. Number of blocks completed was normalized by each monkey into percentages (error bars indicate SEM across subjects). Straight lines indicate least-squares lines regressing ERDS on the fraction of sessions completed. Sub-panels to the right show regression coefficients for the proportion of sessions completed for each group (corresponding to the slope of the fitted lines but accounting for subject variability). Asterisks indicate significant coefficients from mixed-effects analyses. Asterisks between bar plots indicate significant group difference in slopes (*: $p < .05$, **: $p < .01$, ***: $p < .001$). Significant effects shown only. Full statistics are available in Supplementary Table S3.11-14. (c, d) Time course of median RT (c) and initial effective arbitration weight Ω_0 (d) within each block for the entirety of the What-only task. Conventions are the same as in panels a and b.

Group comparison of the initial arbitration weights:

Note that we made a direct group comparison on the paired difference measure ($\Omega_0 - \omega_0$), since commenting on presence/absence of significance on each of the measure (e.g., group effect on Ω_0 or ω_0 separately) does not entail a direct comparison, following a similar logic to the provided reference.

Line #471:

We note that amygdala- and VS-lesioned monkeys did not significantly differ in the initial Ω (effective ω) values (planned contrast in group difference in Ω_0 : $b = 0.0242$, $p = .657$; Supplementary Table S3.8; compare Ω of the first trial in Fig. 3b and Fig. 3c). However, by examining the initial arbitration weights (ω_0) before scaling by ρ , we found that amygdala-lesioned monkeys had significantly smaller ω_0 values compared to controls (mixed-effects analysis on Ω_0 ; $\beta_{amyg} = -0.194$, $p = .00419$) while VS-lesioned group did not ($\beta_{VS} = -0.119$, $p = .108$; Supplementary Table S3.9). More directly, amygdala group showed larger changes in ω_0 after scaling by ρ compared to VS group (group contrast in mixed-effects analysis on $\Omega_0 - \omega_0$: $b = 0.111$, $p = .0204$;

Supplementary Table S3.10). This means that larger values of ρ in amygdala-lesioned monkeys were offset by lower ω_0 values to yield Ω_0 comparable to VS-lesioned monkeys.

Supplementary Table 3.8 Comparison of initial effective arbitration weight (Ω_0) across the three groups during the What/Where task.

Model#	Session-wise $\Omega_0 \sim group + (1 + sess_perc subject)\#$				
Hypothesis #	Initial "effective" arbitration weight is not different between amygdala and VS groups.				
Coeff. #	Estimate	SE	t-stat	DF	P-value
Intercept#	0.44977	0.028853	15.589	529	2.3315e-45
group_amygdala#	-0.20808	0.045898	-4.5336	529	7.1792e-06
group_VS#	-0.23227	0.050245	-4.6228	529	4.7636e-06
Planned contrasts#	Difference between amygdala and VS groups				
group_amyg – group_VS#	0.024192	0.054463	0.44419	529	0.65709

Supplementary Table 3.9 Comparison of initial arbitration weight (ω_0) across the three groups during the What/Where task.

Model#	Session-wise $\omega_0 \sim group + (1 + sess_perc subject)\#$				
Hypothesis #	Initial "effective" arbitration weight is not different between amygdala and VS groups.				
Coeff. #	Estimate	SE	t-stat	DF	P-value
Intercept#	0.38545	0.042547	9.0594	529	2.53e-18
group_amygdala#	-0.19421	0.067522	-2.8762	529	0.0041873
group_VS#	-0.11905	0.073843	-1.6122	529	0.10751
Planned contrasts#	Difference between amygdala and VS groups				
group_amyg – group_VS#	-0.075155	0.079947	-0.94005	529	0.34762

Supplementary Table 3.10 Comparison of the changes in the initial arbitration weight (ω_0) across the three groups (after scaling by ρ) during the What/Where task.

Model#	Session-wise $(\Omega_0 - \omega_0) \sim group + (1 + sess_perc subject)\#$				
Hypothesis #	Changes to effective arbitration weight after re-scaling is larger in amygdala group compared to VS.				
Coeff. #	Estimate	SE	t-stat	DF	P-value
Intercept#	0.062704	0.025184	2.4898	529	0.013087
group_amygdala#	-0.012829	0.040057	-0.32027	529	0.74889
group_VS#	-0.12382	0.044075	-2.8094	529	0.0051467
Planned contrasts#	Difference between amygdala and VS groups				
group_amyg – group_VS#	0.111	0.047735	2.3252	529	0.020437

Please also report how the new permutation tests reported in the results were run, i.e. what was permuted and how the p values were computed.

[R3.2.6] We thank the reviewer for asking this question. We have now added a description of how the permutation test was run in the **Methods** as follows (Line #1161):

For testing the group difference in the ρ parameter (Eqs. 10-11) across task conditions, which was fitted for each subject and therefore lacks adequate sample size, we utilized two-sided permutation tests. Specifically, we conducted permutation tests on the ρ values of each subject from a given pair of tested groups across both tasks and generated a null distribution of test statistic (i.e., mean group difference in

ρ) by randomly shuffling the group assignment 10,000 times. The p-value was calculated as the proportion of permuted test statistics that were as or more extreme than the empirically observed test statistic.

Reviewer #4 (Remarks to the Author):

The authors have admitted that their dataset is not original, and that they here test a linear combination of two already existing models. Thus, the conceptual novelty is very limited.

Nevertheless, I thank the authors for performing a few additional analyses which have improved a bit their manuscript (model comparison + model simulation (Fig.S7)), but which nevertheless confirm that the data does not support the authors' conclusions, as they show that neither model correctly reproduces the animal data.

This manuscript should not be accepted for publication in such a high-impact journal given its strong limitations.

We thank the reviewer for their time in reading our work and for the valuable feedback provided during the first round of reviews, as well as for several additional constructive suggestions in this round. However, we were concerned by the accusatory tone in phrases such as “the authors have admitted...”. We invested considerable time and effort in addressing the reviewer’s earlier comments—dedicating 34 of the 71 pages in our original response letter to them—and believe the manuscript has significantly improved as a result. While the final decision regarding publication rests with the editors of the journal, we respectfully disagree with the reviewer’s overall assessment of our study for the following reasons.

First, regarding the originality of the dataset, we want to emphasize that we have never intended to obscure or downplay the fact that these datasets were previously published, contrary to the reviewer’s accusatory comment. From our initial submission, we clearly cited and referenced all original studies and discussed their findings in detail. We explicitly stated in the Methods section that the “experimental setup ... and some analyses of the data have been previously reported.” Nonetheless, we apologize if any of our wording was perceived as misleading and hope that our current and previous revisions have clarified this point—for example, the addition of **Supplementary Figure S1** summarizing the subjects, and our revision in **[R4.2.1]** below.

Second, re-analyzing existing datasets using new methods is a well-established and important scientific practice, essential for both re-evaluating prior conclusions and generating new insights. Therefore, we believe that the reuse of previously published data, when properly acknowledged—as we have done—should not be discouraged. This point is especially relevant for studies involving non-human primates with controlled lesions to specific brain areas. Such experiments are notoriously time-consuming and logistically challenging and expensive, yet they

are exceptionally valuable due to the close phylogenetic relationship between non-human primates and humans—offering a higher degree of translational relevance than rodent models. Discouraging the re-analysis of these datasets would, in our view, be a disservice to the scientific community, funding agencies, and the broader public, as it undermines the efficient and responsible use of rare and valuable scientific resources.

Third, we believe our study offers important novel insights by re-examining the data through the lens of arbitration and the interaction between stimulus- and action-based systems. This perspective is supported by comprehensive analyses of both control and lesioned animals, as well as model simulations that generate new predictions (e.g., **Fig. 5**) and provide a unified explanation for previously conflicting findings within the arbitration framework. For further discussion on this point, please see our response in **[R4.2.2]** below. Lastly, because obtaining rewards from the environment inherently requires action, our findings from amygdala-lesioned monkeys contribute a more nuanced understanding of the amygdala's role in learning and decision making.

Fourth, we are concerned that the reviewer appears to overlook a central contribution of our work—namely, the introduction of a model that captures the interaction and arbitration between two learning systems. Our model comparison results consistently demonstrate that the dynamic arbitration model provides a better fit to the choice data across all groups and tasks. As the reviewer notes in **[R4.2.5]**, we have rigorously validated this using cross-validation, even at the level of individual monkeys—an approach that is rarely undertaken. Importantly, the study includes 20 unique animals, and the consistency of results across individuals provides strong support for our claims.

Regarding the results in **Fig. S7**, we acknowledge the observed discrepancy between model and data but believe it is expected and within a reasonable range. The CDF plots reflect the full distribution of the empirical data, including potential outliers that are difficult for any model to capture. The discrepancy likely stems from our previous method of averaging simulated metrics over 100 runs per block, which reduced variability relative to the raw empirical data, where each choice occurred only once. To address this, we now compare the full set of simulated block-level metrics (prior to averaging) to the empirical data. Ultimately, we view this as an issue of visualization rather than a substantive weakness of the model. The model comparison results continue to clearly support the superior explanatory power of our proposed model and framework.

Please see our point-by-point responses below to your specific comments.

LIMITED ORIGINALITY

- The authors have clarified in the rebuttal that "the data used in our study have been previously published". However, there are still sentences in the manuscript that may mislead readers by making them think some data have been newly collected: "The remaining control and the four amygdala-lesioned monkeys were newly trained for this

study". The authors should more clearly state that this paper does not present any new data.

[R4.2.1] We thank the reviewer for catching this. By “this study” we had originally meant to refer to the later What/Where task, not the current manuscript. We have now fixed the given sentence as follows (Line #800, in **Methods**):

The remaining control and the four amygdala-lesioned monkeys were additionally trained for the subsequent study using the What/Where task¹⁸.

- Regarding conceptual originality, their new model is, as the authors state, consistent with the previous one, which was based on the exact same dataset. Hence, beside having the two models (one for each task) linearly combined into one, there is no conceptual novelty proposed in this paper.

[R4.2.2] We respectfully disagree with this assessment. Our model is not merely a linear combination of two existing models; rather, it introduces a dynamic mechanism that balances the influence of stimulus-based and action-based systems on choice behavior. To that end, we have devoted substantial effort to characterizing this interaction and its behavioral implications. The reviewer’s reference to a “linearly combined” model corresponds to the static omega model, which we have shown performs significantly worse than our dynamic model in fitting choice behavior. Thus, the dynamic arbitration between stimulus- and action-based learning under uncertainty is a novel and central aspect of our work. These features represent significant contributions to the literature, especially given that stimulus-based tasks inherently require actions, introducing uncertainty about the true source of reward.

Therefore, we believe that the main conceptual novelty lies in the specification of the arbitration mechanism and its verification through empirical data. More specifically, we delineate the types of reliability signals that the brain could use, confirm their empirical evidence in both control and lesioned monkeys, and also demonstrate that RT data is meaningfully predicted by the dynamic arbitration weight. In other words, we not only go beyond the specification of behavioral mechanisms which the intact controls operate on, but also pinpoint the separable influences of different subcortical structures on this process.

Lastly, we would like to remind the reviewer that our analyses demonstrating consistency with previous models were conducted specifically in response to the reviewer’s earlier request [see prior comment **R4.6**], which asked us to “explain why the same authors with the same data have previously interpreted amygdala lesions as ‘reduc[ing] choice consistency (sensitivity to value signals) for stimulus-based learning and increas[ing] sensitivity to negative feedback (α -) for action-based learning.” We therefore believe that this consistency with prior models should be viewed as a strength rather than a limitation, as it demonstrates that our more general framework not only generates new insights but also successfully accounts for earlier, more narrowly focused findings.

- *The best fitting model, a dynamic combination of both RL models, does not constitute a conceptual breakthrough for multiple reasons:*

- 1) Presented analyses lack compelling evidence that the dynamic part adds on significant explanatory power: the authors insist that the amygdala lesion is specifically associated with a lower ω_0 (ignoring the difference in alpha- and alpha_omega of Fig S9, S10), which corresponds to the baseline arbitration. The fact that this affects the arbitration weight dynamically in simulations does not imply that this dynamic mechanism is necessary to explain the behavioral patterns, compared to a static arbitration weight. Adding ω_0 in regression analyses of the RT (figure SN2) and showing simulations of the static model in posterior predictive checks (figure S7) to see if it behaves significantly worse than the dynamic model, could help support the proposed dynamic model as a real breakthrough.

[R4.2.3] We thank the reviewer for this detailed comment. To clarify, the RT regression analysis in **Supplementary Fig. SN2** has demonstrated that the trial-by-trial arbitration weight is a significant predictor of RT. Note that ω_0 is already included in the regressor as it corresponds to the initial ω value on the first trial. Nonetheless, to address the reviewer's concern, we have run an additional analysis comparing the goodness-of-fit (cross-validated R^2) of regression models that predict RT with either static or dynamic arbitration weight. As shown in the **Fig. R4.1** (attached below), the dynamic arbitration weight outperforms static arbitration weight in predicting RT. This is consistent with the model fitting results, which have shown that dynamic arbitration weight is superior in predicting choice behavior compared to the static models.

Relatedly, we would like to highlight that the model fits to choice behavior (**Fig. 2b** and **Fig. 3a**) provide compelling evidence that the dynamic arbitration model offers the best account of the data. We also emphasize that this study includes 20 unique subjects, and the consistency of results across all individuals further strengthens the validity of our findings.

To address the reviewer's comment regarding simulations of the static model, we have repeated the analysis presented in **Supplementary Fig. S7** using the static omega model. The corresponding statistics from two-sample Kolmogorov–Smirnov (K–S) tests, comparing the distributions of model-simulated and empirical metrics, are reported in **Table R4.2** below. This analysis shows that the dynamic arbitration model consistently produces distributions that are closer to the empirical data than those of the static model. These findings further support our conclusion that reliability-based dynamic arbitration provides a better account of the empirical data than a static mixture of two strategies.

We have now updated **Supplementary Fig. S7** (attached below) to reflect these results.

Figure R4.1 Plotted are mean 10-fold cross-validated R-squared measures of regression models predicting RT with static or dynamic arbitration weight, inferred by respective models. To directly compare between two types of arbitration weight, we used regression models with just a single predictor—either static or dynamic arbitration weight.

Block types → K-S test stats ↓	What-only (amyg , VS)	What	Where
Dynamic model: Distance to data	$D = .0742, p = 2.77e-09$ $D = .100, p = 2.09e-09$	$D = .0370, p = 0.136$ $D = .0322, p = 0.343$	$D = .0933, p = 4.48e-07$ $D = .0767, p = 9.64e-05$
Static model: Distance to data	$D = .102, p = 2.7e-17$ $D = .111, p = 2.11e-11$	$D = .0919, p = 1.21e-07$ $D = .0794, p = 4.57e-05$	$D = .222, p = 3.20e-38$ $D = .165, p = 2.48e-20$
Distance between two models	$D = .0359, p = 2.69e-86$ $D = .0123, p = 2.43e-06$	$D = .0622, p = 7.48e-166$ $D = .0537, p = 7.18e-107$	$D = .131, p < 4.97e-324$ $D = .102, p < 4.97e-324$

Table R4.2. Reported are the D-statistics and corresponding p-values of the two-sample Kolmogorov–Smirnov test, comparing the distributions of $\Delta ERDS = ERDS_{Stim} - ERDS_{Action.}$, for the amygdala- (red) and VS-lesioned (blue) monkeys for respective block types. The first two rows report the distance of model-simulated metrics from the empirical data. The last row compares the distributions of the metrics simulated by the two models (static or dynamic arbitration). Smaller D-statistics indicate closer distances (larger similarity) between the two distributions under comparison.

Figure S7. (a) Plotted are cumulative distribution functions (CDF) of empirical (magenta) and simulated (red) values of the relative strength of two strategies ($ERDS_{Stim} - ERDS_{Action}$ during each block) in amygdala-lesioned monkeys. Dynamic ω - ρ , Static ω , and one-system (stimulus-only) models are compared. Reported values next to each model are the test statistics (D-values) using the two-sample Kolmogorov-Smirnov test, comparing the distance between the distributions of data and the indicated model. Asterisks next to the D-values indicate significance ($p < .001$; ns = not significant). Asterisks between the model statistics indicate significant difference between the distributions of two models. Each block was simulated 100 times with the fitted parameters of each model. (b) CDF of empirical and simulated values of $ERDS_{Stim} - ERDS_{Action}$ for the What blocks during the What/Where task. Conventions are the same as in (a). (c) Same plot as in (b) but for the Where blocks during the What/Where task. Action-only model was simulated for the one-system model. (d-f) Same plots as in panels a-c but for VS-lesioned monkeys.

- 2) The ambiguity surrounding the lack of novelty in the data presented overshadows the explanatory potential of the proposed model. What predictions would this dynamic arbitration model do about neural circuitry and cognition in other tasks? What would be the experimental design to falsify a static model or other types of arbitration? Lines 708-732 the authors discuss known results about the role of amygdala in reversal tasks, and how the experimental data presented at the beginning of the paper fit in. But these data have already been published, and a theoretical paper presenting a conceptual novel and unifying model should present more simulations to support these verbal assumptions.

[R4.2.4] We would like to note that we have dedicated an entire figure (**Fig. 5**) specifically to address this point. Proposing new experimental designs to dissociate different models is beyond the scope of the current study, as varying task contexts may engage distinct behavioral mechanisms.

Regarding the reviewer’s mention of “other types of arbitration,” it is unclear what specific alternatives are being suggested. However, we have already tested several forms of reliability signals that could plausibly guide arbitration (see **Supplementary Fig. S4**). Additionally, we discuss in detail the potential neural circuits involved in arbitration, including pathways such as the prefrontal cortex–amygdala connection.

COMPUTATIONAL ANALYSES

- The authors have improved the model comparison by showing individual cross-validated log-likelihood and performing the cross-validation across all the experimental blocks.

[R4.2.5] We thank the reviewer for this positive feedback.

- New simulations of different models compared with the data make an important contribution (figure S7). They do not support the authors' conclusions, however, as they show quite clearly that neither model correctly reproduces the animal data. Furthermore, as mentioned above, the same simulations using the static arbitration model (with a common omega across the blocks) are needed to conclude that the dynamic arbitration aspect is a necessary ingredient.

[R4.2.6] We would like to emphasize that these simulation results are not new—they were included in our initial submission (**Supplementary Fig. S7g–i**). The only addition in the latest revision is an alternative visualization of the same results (**Supplementary Fig. S7a–f**). Moreover, as discussed in **[R4.2.3]** above, we have shown that the dynamic arbitration model consistently outperforms the static model in the lesioned groups. For completeness, we now also provide the same analysis for the control monkeys (see **Table R4.3** below), which likewise demonstrates that the dynamic arbitration model significantly outperforms the static model in this group as well.

Block types → K-S test stats ↓	What-only (controls)	What	Where
Dynamic model: Distance to data	$D = .0454, p = 0.00122$	$D = 0.125, p = 7.31e-41$	$D = 0.157, p = 2.40e-63$
Static model: Distance to data	$D = 0.0645, p = 6.36e-07$	$D = 0.244, p = 1.51e-155$	$D = 0.317, p = 2.23e-255$

Distance between two models	$D = 0.0227, p = 1.18e-37$	$D = 0.125, p < 4.97e-324$	$D = 0.162, p < 4.97e-324$
----------------------------	----------------------------	----------------------------

Table R4.3. Reported are the D-statistics and corresponding p-values of the two-sample Kolmogorov–Smirnov test, comparing the distributions of $\Delta ERDS = ERDS_{Sim} - ERDS_{Action}$ in the intact controls.

Round 3: RESPONSE LETTER TO THE REVIEWERS' COMMENTS AND SUMMARY OF REVISIONS

REVIEWERS' COMMENTS

Reviewer #3 (Remarks to the Author):

The authors have made a thorough and good faith effort to address the issues raised in my previous review and have massively improved the clarity of the stats reporting through the use of contrasts to assess differences between the two lesion groups.

The revised manuscript is an interesting and valuable piece of work which I believe has substantial novelty, through introducing and carefully providing evidence for a model of dynamic arbitration between stimulus and action-based learning, and the effects of VS and amygdala lesions on this.

Response: We thank the reviewer for positive evaluation of our work. We believe that the reviewer's suggestions have greatly improved the statistical rigor and the overall strength of the manuscript.

Minor point: I think there is a typo in supplementary table 3.3 as the model formula does not include group, but there are values detailed for coefficients group_amygdala and group_VS.

Response: We thank the reviewer for catching this. We have now included the term for the group effect. The formula in **Table S3.3** has been fixed as follows:
Block-wise $\Delta\psi \sim 1 + \text{group} + (1 + \text{sess_perc} + \text{block_in_sess} | \text{subject})$

Reviewer #4 (Remarks to the Author):

I would like to thank again the authors for all the efforts they made to significantly improve their manuscript. I do acknowledge, as I did before, that they have improved their manuscript by performing additional analyses. I acknowledge the further clarifications and analyses that they have provided at this new stage in response to Reviewers #2, #3 and to me.

Response: We thank the reviewer again for taking the time and effort to assess our manuscript very carefully. We are happy to hear that the reviewer's comments have been addressed.

"However, we were concerned by the accusatory tone in phrases such as "the authors have admitted...". [...] First, regarding the originality of the dataset, we want to emphasize that we have never intended to obscure or downplay the fact that these datasets were previously published, contrary to the reviewer's accusatory comment."

I thank again the authors for having now clarified their manuscript about the data. While I don't pretend that the previous formulations about the experimental data (e.g., "All remaining monkeys were not previously used in the studies mentioned above") had been intentional, I

think the authors had not made enough efforts to make it very clear initially that all the data had already been published before. Their answers to my previous and present comments speak for themselves:

Previous stage: "[R4.1] [...] In the process, we also identified an error regarding the monkeys used across experiments: only one of the control monkeys (monkey #2), and not four as previously stated, and all three VS- lesioned monkeys participated in both the What-only and What/Where tasks."

Present stage: "[R4.2.1] We thank the reviewer for catching this. By "this study" we had originally meant to refer to the later What/Where task, not the current manuscript. We have now fixed the given sentence [...]"

This is not a negligible issue since the formulations adopted in an initial submission leave first impressions to reviewers that may contribute to inflating their impression about the originality. This is why I think it is deontologically not acceptable.

Response: We thank the reviewer for this feedback. As we have not intended for this misunderstanding, we hope that the current revision further clarifies the point about re-use of previously published data.

Nevertheless, as for computational analyses, I would like to thank the authors for providing characterizations of the interaction between the two components of their computational model. I agree that this brings insights into the dynamic arbitration mechanism that balances the influence of stimulus-based and action-based systems on choice behavior, thus going beyond the static omega model.

The manuscript has now been further improved and can be published. I still consider that the originality is too limited for Nature Communications. But I leave this decision to the editor.

Response: We thank the reviewer for this valuable feedback. We are happy to hear that we have reached agreement about the additional insight that dynamic arbitration model brings, beyond the static model.